# Net primary production annual maxima in the North Atlantic projected to shift in the 21st century

**Jenny Hieronymus**[1], **Magnus Hieronymus**[1], **Matthias Gröger**[2], **Jörg Schwinger**[3], **Raffaele Bernadello**[4], **Etienne Tourigny**[4], **Valentina Sicardi**[4], **Itzel Ruvalcaba Baroni**[1], and **Klaus Wyser**[1]

[1]Swedish Meteorological and Hydrological Institute, SMHI, Norrköping, 601 76, Sweden
[2]Department of Physical Oceanography and Instrumentation, Leibniz Institute for Baltic Sea Research Warnemünde, Rostock, 18119, Germany
[3]Barcelona Supercomputing Center, BSC, Plaça d' Eusebi Güell, 1–3, 08034 Barcelona, Spain TS1
[4]Barcelona Supercomputing Center, BSC, Barcelona, 08034, Spain

**Correspondence:** Jenny Hieronymus (jenny.hieronymus@smhi.se)

**Abstract.** Shifts in the day of peak net primary production (NPP) were detected in different biogeochemical provinces of the North Atlantic (25–65° N). Most provinces displayed a shift toward earlier peak NPP, with the largest change points in the 21st century and in the northern parts of the domain. Furthermore, the occurrences of the first day with a mixed-layer depth (MLD) shallower than 40 m and the day of peak NPP are positively correlated over most of the domain. As was the case for the day of peak NPP, the largest change points for the day of MLD shallower than 40 m occur around or after the year 2000. Daily output from two fully coupled CMIP6 Earth system models, EC-Earth3-CC and NorESM2-LM, for the period 1750–2100 and under the SSP5-8.5 scenario, were used for the analysis. The ESM NPP data were compared with estimates derived from Carbon, Absorption and Fluorescence Euphotic-resolving (CAFE) satellite-based data. The ESMs showed significant differences from the CAFE model, though the timing of peak NPP was well captured for most provinces. The largest change points in the day of peak NPP occur earlier in EC-Earth3-CC than in NorESM2-LM. Although SSP5-8.5 is a scenario with very high warming, EC-Earth3-CC generates change points for most provinces in the early part of the 21st century, before the warming has deviated far from lower-emissions scenarios. NorESM2-LM displays the largest change points centered around the mid 21st century, with two out of eight provinces displaying the largest change point before the year 2050. The early timing of the detected shifts in some provinces in both ESMs suggests that similar shifts could already have been initiated or could start in the near future. This highlights the need for long-term monitoring campaigns in the North Atlantic.

## 1 Introduction

Net primary production (NPP) is the rate of photosynthetic carbon fixation minus cellular respiration. In the ocean, the majority of NPP is performed by microscopic planktonic phototrophs. Though the individual plankton are small, the total marine NPP is similar in size to its terrestrial counterpart, with an estimated size of marine NPP on the order of $50 \, \mathrm{Pg \, C \, yr^{-1}}$ (e.g., Kulk et al., 2020; Westberry et al., 2008; Silsbe et al., 2016; Field et al., 1998). Phytoplankton constitutes the basis of the food chain, and the carbon fixed through NPP provides the energy for higher trophic levels. Changes in NPP thus affect the entire ecosystem and ultimately fisheries and human food supply (Stock et al., 2017). In addition, NPP is the first step in the biological carbon pump, a set of processes by which carbon is exported from the surface to the deep ocean through the sinking of organic matter (Lutz et al., 2007). Understanding how the NPP and the subsequent export of organic carbon from the euphotic zone will change in the future climate is thus vital for evaluating the future uptake of atmospheric carbon dioxide (Honjo et al., 2014).

The North Atlantic is a region of particular importance for carbon sequestration in the deep ocean (Goris et al., 2018; Baker et al., 2022). This region contributes about 0.55–1.94 $Pg\,C\,yr^{-1}$ (Sanders et al., 2014) to the global export production, estimated to be 4–12 $Pg\,C\,yr^{-1}$ (DeVries and Weber, 2017). Moreover, cold water increases $CO_2$ solubility. Deep mixing and subduction in the subpolar North Atlantic thereby result in a net transport of carbon to depth, a combination of processes known as the solubility pump.

NPP is affected by climate variability through precipitation, wind patterns, temperature, and light and is thus projected to change with anthropogenic climate change (Laufkötter et al., 2015; Paerl et al., 1999; Myriokefalitakis et al., 2020). Though an increase in temperature may enhance the growth rate of phytoplankton and thereby the net primary production, global NPP is projected to decrease (Behrenfeld et al., 2006; Steinacher et al., 2010; Bopp et al., 2013); however, the uncertainty displayed in state-of-the-art Earth system models (ESMs) is very large (Kwiatkowski et al., 2020). A projected NPP decline is often explained as being caused by increased water column stability that decreases the amount of nutrients available for primary production (Behrenfeld et al., 2006; Steinacher et al., 2010), but processes such as retreat of sea ice and increased stratification in high latitudes reduce the light limitation, leading to NPP increases (Kwiatkowski et al., 2020). Efforts have been made to estimate how NPP has already changed in the historical satellite record, but the limited range of satellite time series makes such endeavors difficult. Estimates range from −2.1 % per decade over the period 1998–2015 (Gregg and Rousseaux, 2019) to no significant change (Kulk et al., 2020).

Several mechanisms have been hypothesized to explain the seasonal cycle of phytoplankton blooms. One often cited is the critical depth hypothesis (Sverdrup, 1953), which postulates that a bloom can occur when the mixed layer has shoaled to a critical depth where the light-limited gross production outweighs respiration. It does not, however, provide an explanation as to when a bloom starts and ends. A more recent hypothesis, termed the disturbance recovery theory, regarding the timing of blooms, was given by Behrenfeld (2010) (see also Behrenfeld and Boss, 2018). The hypothesis suggests a balance between growth and loss in terms of respiration, grazing, and disturbances to the physical environment, such as the depth of the mixed layer. Other hypotheses include that of Smyth et al. (2014), which relates seasonality to the shift between negative and positive net heat flux.

The study of the timing of recurring biological events is termed "phenology" and has become an important field of research during recent years owing to its dependence on climate (Chivers et al., 2020). Phenological indicators include seasonal length, the timing of the start and end of the bloom, and the timing of the annual maximum (e.g., Nissen and Vogt, 2021; Henson et al., 2013). The phenology of algal blooms can change along with climate change, with cascading effects into higher trophic levels up to fish and marine mammals. Changes in the phenology of phytoplankton blooms owing to climate change have already been observed in the North Sea with the Continuous Plankton Recorder (CPR) since 1960, with data displaying a significantly earlier onset of the spring bloom (Chivers et al., 2020).

A phenological change in phytoplankton blooms will affect zooplankton and larvae, as the timing of available food resources will change, an effect known as the match–mismatch hypothesis (Cushing, 1990; Durant et al., 2007). The suggested causes of phenological shifts range from bottom-up controls, including thermal stratification occurring earlier in the year allowing for an earlier bloom initiation, to top-down controls, resulting from changes in zooplankton grazing pressure (Yamaguchi et al., 2022).

Henson et al. (2013) used historical simulations from six ESMs covering the years 1985–2009 and a high-emissions future scenario (RCP8.5) to study changes in NPP phenology. They found a shift toward an earlier peak NPP for most areas around the globe. However, the monthly resolution of the CMIP5 data dampens the phenology signal considerably. In a more recent study, Henson et al. (2018) used higher-frequency model output to investigate the effect of the temporal resolution on the results of phytoplankton phenology. They found that, to detect long-term trends in bloom timing, temporal resolution of 20 d or less is required.

However, even though a 20 d resolution may be adequate to detect long-term trends, it is certainly not enough for detecting the timing of a rapid change in phenology in the course of global warming. In this paper, we use daily output from two ESMs that contributed to the 6th Coupled Model Intercomparison Project (Eyring et al., 2016) to investigate the evolution of oceanic net primary production and its phenology. The focus is a region 25–65° N in the North Atlantic during the period 1750–2100. We divide the domain into biogeochemical provinces (Longhurst et al., 1995) to see the evolution of NPP in different subregions across the domain. We then investigate the occurrence of change points in the time series of the day of peak NPP for the different provinces using change-point analysis.

To test how well the timing of mixed-layer shoaling relates to the timing of peak NPP in different North Atlantic regions, we also investigate the largest change points in the day of the mixed-layer shoaling above a certain limit (here arbitrarily taken to be 40 m). We further analyze the cross-correlation between the day of mixed-layer depth shallower than the limit and the day of peak NPP. The cross-correlation analysis is complementary to the change-point analysis. This analysis highlights when the timing of the mixed-layer shoaling and peak NPP leads and lags are covariant in the different provinces.

## 2 Method

Daily output of vertically integrated NPP has been produced using NorESM2-LM and EC-Earth3-CC for 100 years of pre-industrial control (piControl), historical (1850–2014), and the very high-emissions scenario SSP5-8.5 (2015–2100, Kriegler et al., 2017). The high-emissions scenario was chosen to generate a sort of upper end of the amount of change. Since daily data require a lot of resources, no lower-emissions scenarios were run. Note that, in EC-Earth3-CC, NPP is integrated over the entire water column, while it is integrated over the top 100 m in NorESM2-LM. All runs are forced with prescribed atmospheric $CO_2$ concentrations (concentration driven) in accordance with Meinshausen et al. (2020). The models are described in Sect. 2.1. Section 2.2 describes the observational data set, Sect. 2.3 describes the Longhurst provinces, and Sect. 2.4 provides an overview of the change-point analysis method used. The phenological indicator that we used is the day of peak NPP, which is calculated as the annual maximum of NPP. This is a well-defined metric that is robust unless for bimodal distributions with two peaks of similar size, which were not found in our data. The metric has previously been used in, for example, Nissen and Vogt (2021) and Henson et al. (2013).

The mixed-layer depths used for the analysis are calculated differently in the two ESMs. In EC-Earth3-CC, we have used the turbocline depth as a mixed layer depth proxy calculated with a turbulent mixing coefficient criterion of $5\,\mathrm{cm^2\,s^{-1}}$, while in NorESM2-LM, the mixed-layer depth has been calculated in accordance with the criterion of de Boyer Montégut et al. (2004) and with a density difference of $0.03\,\mathrm{kg\,m^{-3}}$.

### 2.1 Models

#### 2.1.1 EC-Earth3-CC

EC-Earth3-CC is an ESM developed by a European consortium of institutes and universities (Döscher et al., 2022). It is available in several different configurations. For this work, we used EC-Earth3-CC, which consists of the Integrated Forecast System (IFS) CY36R4 of the European Centre for Medium-Range Weather Forecasts (ECMWF) to simulate physics of the atmosphere and land surface, NEMO3.6 (Madec et al., 2015) for ocean physics, LPJ-Guess (Smith et al., 2014) for terrestrial vegetation, and PISCES (Aumont et al., 2015) for ocean biogeochemistry. In concentration-driven form, PISCES is fed a spatially uniform atmospheric $pCO_2$, while a $CO_2$ mapping occurs within IFS to account for regional heterogeneities.

PISCES is a mixed Monod-quota model simulating two different phytoplankton functional types: diatoms and nanophytoplankton; two size classes of zooplankton: micro and meso; and nutrients: nitrate, ammonium, phosphate, iron, and silicate. Iron and silicate are modeled using quotas in

phytoplankton and the other nutrients with fixed Redfield ratios (Redfield, 1958). Phytoplankton growth depends on nutrient concentrations in ambient water, light, and temperature. PISCES further simulates the carbon system, as well as dissolved and particulate organic matter. The integrated net primary productivity used for the analysis is integrated over the water column and also summed over the two different phytoplankton functional types.

PISCES has been used and validated in a number of settings (Ramirez-romero et al., 2020; Gutknecht et al., 2019; Kwiatkowski et al., 2018). Skyllas et al. (2019) showed a good agreement between EC-Earth3-CC and temperature, salinity, and nutrients and chlorophyll *a* observations in an offline ocean-only version of NEMO-PISCES, for a north–south (29–63° N) transect in the northwest Atlantic. Net primary production has not previously been validated for EC-Earth3-CC, although the air–sea $CO_2$ flux, which is strongly affected by net primary production, was compared with an observation-based climatology in Döscher et al. (2022). Their results showed stronger uptake of $CO_2$ than observations in the North Atlantic, which is thought to be caused by too active convection in the Labrador Sea.

#### 2.1.2 NorESM2-LM

The Norwegian Earth System Model NorESM2 (Seland et al., 2020; Tjiputra et al., 2020) is a fully coupled ESM based on the Community Earth System Model version 2 (CESM2, Danabasoglu et al., 2020) but employs a different ocean component (the Bergen Layered Ocean Model, BLOM) and a modified atmosphere model (CAM6-Nor). The land surface and terrestrial biogeochemistry is represented by the Community Land Model version 5 (CLM5). BLOM uses isopycnic coordinates in the vertical (below a bulk mixed layer represented by two non-isopycnic model layers on top), and it includes the iHAMOCC model to represent ocean biogeochemistry. BLOM is coupled to the sea–ice component CICE5, which is the same as in CESM2. The LM version of NorESM2 used in this study has an atmosphere–land resolution of 2° and a nominal ocean model resolution of 1°. iHAMOCC is derived from the HAMOCC model (Six and Maier-Reimer, 1996; Ilyina et al., 2013) and was adapted for use with isopycnic coordinates by Assman et al. (2010). HAMOCC includes a relatively simple NPZD ecosystem model with one phytoplankton functional type, one zooplankton functional type, and an implicit representation of calcifying and silicifying organisms. The model simulates the nutrients nitrate, phosphorus, and dissolved iron with phytoplankton nutrient uptake according to Redfield molar ratios. The model also simulates the carbon system and dissolved and particulate organic matter. The growth of phytoplankton is further affected by light and temperature.

NorESM2-LM has been validated with regard to biogeochemical variables including net primary production in Tjiputra et al. (2020). The results show a seasonal cycle of

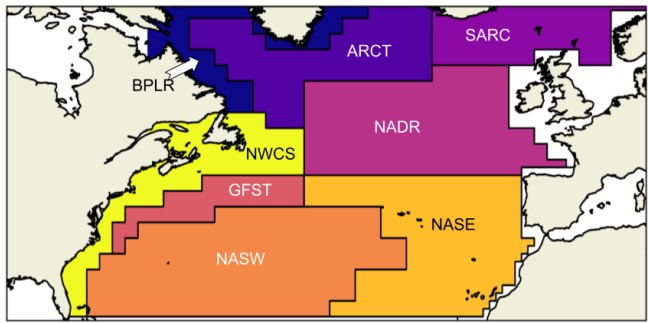

**Figure 1.** Study area and Longhurst provinces. BPLR: boreal polar, ARCT: Atlantic Arctic, SARC: Atlantic sub-Arctic, NADR: North Atlantic drift, GFST: Gulf Stream, NASW: northwest Atlantic subtropical gyre, NASE: northeast Atlantic subtropical gyre, NWCS: northwest Atlantic shelf.

marine NPP that is reasonably well captured in amplitude but with a too-low annual mean.

## 2.2 Satellite-based data – the CAFE model

Direct observational data records of net primary production are scarce, and to validate the two ESMs with respect to NPP, we chose to use data from a satellite-based approach. There are several different methods for deriving total water column NPP estimates from satellite data. Often, they are based on ocean color (Behrenfeld and Falkowski, 1997), carbon (Westberry et al., 2008), or absorption (Smyth et al., 2005).

In this work, we use data from the Carbon, Absorption and Fluorescence Euphotic-resolving (CAFE) model (Silsbe et al., 2016). The model utilizes satellite-derived properties and has been shown to compare well to in situ observations (Johnson and Bif, 2021). We utilize the MODIS-aqua (moderate resolution imaging spectroradiometer) data set from 2002 to 2021 here.

## 2.3 Longhurst provinces

The seasonality of NPP depends, among other things, on local physical ocean conditions. In modeling the terrestrial environment, the division into provinces of similar growth conditions, such as boreal forest or savanna, is well defined, while in the ocean, biological differences between regions exist but are more difficult to constrain (Sathyendranath et al., 1995). The division of the global ocean into biogeochemical provinces has been attempted several times (Longhurst et al., 1995; Sathyendranath et al., 1995) with the object of determining the global or regional net primary production. Longhurst et al. (1995) defined the static boundaries that we have used in this analysis. Although the boundaries are, in reality, shifting on seasonal and interannual time scales (Reygondeau et al., 2013), we chose to use the static boundaries, as we then are able to compare the same localities in the two models and in the CAFE data. The North

Atlantic is divided into the provinces shown in Fig. 1. Note that we have chosen not to include the entire Arctic basin, causing the Arctic provinces to be cut off in the north. The boreal polar province (BPLR) is defined by the southward flowing Labrador current that continues northward along the Greenland coastline. The Atlantic Arctic province (ARCT) is defined by strong stratification caused by large inflow of meltwater, while the Atlantic sub-Arctic (SARC) is characterized by poleward-flowing warm North Atlantic water. The Gulf Stream (GFST), North Atlantic drift (NADR) and northwest (NASW) and northeast (NASE) Atlantic subtropical TS2 gyres are governed by westerly winds and a Sverdrup-type circulation. We have also included the coastal province northwest Atlantic continental shelf (NWCS).

## 2.4 Change-point analysis

Change-point detection is a method to identify abrupt change in a time series. Formally, the problem is to find the best possible segmentation of a signal according to some chosen criterion. Depending on this criterion, one can look for changes in, for example, the mean, variance, or a spectral characteristic of a given signal. In climate science, the method has been used to detect shifts in a wide variety of quantities (Beaulieu et al., 2012), such as Atlantic Meridional Overturning Circulation (AMOC) strength (Smeed et al., 2018), coastal organic carbon sequestration (Watanabe et al., 2019), and cod stock (Möllmann et al., 2021). We have used change-point detection to identify rapid change in the calendar day of peak NPP. The calculations were performed using the Python package Ruptures (Truong et al., 2020).

In general, change-point detection requires a search method, a cost function, and a constraint on the number of change points to detect. Search methods can be either exact or approximate. Here, we use a version of the former, called optimal detection, as computational speed is not an issue. Moreover, we primarily use the kernel-based cost function and a constraint where we directly pick the desired number of change points. Many methods of change-point analysis focus on finding a predetermined number of shifts in a predefined quantity, such as the time series mean or variance (Truong et al., 2020). Another option is the pruned exact linear time (PELT) search method (Killick et al., 2012), which does not require the number of change points to be determined beforehand. Instead, one defines a penalty that is related to the amplitude of the change of interest. A small penalty generates many change points, which may arise due to intra-annual variability or noise, while a large penalty instead only gives the largest, if any, changes in the time series. By choosing a large enough penalty, the number of change points can in this way be tuned. In the process of doing this research, we tested both approaches.

Furthermore, instead of predefining the type of time series change, we have chosen to primarily use a kernel-based nonparametric cost function developed by Arlot et al. (2019),

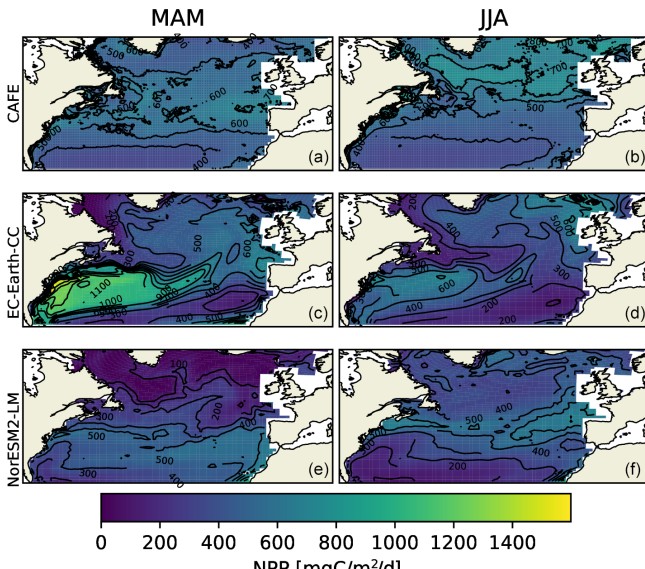

**Figure 2.** Vertically integrated seasonal mean NPP from the CAFE model (**a, b**), EC-Earth3-CC (**c, d**), and NorESM2-LM (**e, f**) for March, April, and May (MAM, **a**, **c**, and **e**) and for June, July, and August (JJA, **b**, **d**, and **f**). The difference between the contours is $100\,\mathrm{mg\,C\,m^{-2}\,d^{-1}}$.

called "the kernel-based cost function". This model can detect all types of changes in the probability distribution of the time series, including the mean, variance, and higher-order changes such as the skewness and kurtosis. The upside of this approach is that no large changes are missed. The downside is that the method does not provide information on which change point is related to what type of change. Therefore, we complement the method with analyses using the least absolute deviation (L1) cost function that detects changes in the median and the least squared deviation (L2) cost function that detects changes in the mean of the time series. Both of these are also available through the Ruptures package, and the search method used is the same as for the kernel-based cost function.

## 3 Results

### 3.1 ESMs vs. CAFE

We compared the daily ESM data with 8 d NPP estimates from the CAFE data (Silsbe et al., 2016). Seasonal mean NPP over the MODIS-aqua period 2003–2021, for March, April, and May (MAM) and June, July, and August (JJA), are shown in Fig. 2. Note that the 2003–2021 period modeled by the ESMs is not the same period as that in the observationally constrained CAFE model. The two ESMs are forced with greenhouse gas concentrations that are similar to those for the period, but the internal variabilities of the two ESM climate systems are not synchronized with nature or with each other. The comparison that can be done is thus strictly climatological.

Figure 2 shows large spatial differences between CAFE, EC-Earth3-CC, and NorESM2-LM data. Most notably, EC-Earth3-CC shows a very strong NPP in MAM over the Gulf Stream region. The high-resolution CAFE data show that the enhanced production occurs in the warm Gulf Stream eddies, while the low resolution of the ESMs gives a wider warm water transport. The NorESM2-LM results in the Gulf Stream region are closer to the CAFE data, although the production in the northern part of the domain is underestimated in both ESMs.

The 8 d moving average of the area mean seasonal cycle over the period 2003–2021 for the different provinces is shown in Fig. 3. Owing to the smaller area seen by satellites in winter, the CAFE data contain missing data over the winter months. To correctly compare the seasonal cycles, the ESM data were bounded to the north by the maximum latitude present in the CAFE data (Fig. S1 in the Supplement).

The sizes of the NPP annual maxima, as shown in Fig. 3, are well captured by both ESMs, with the notable exception of the Gulf Stream province (GFST) in EC-Earth3-CC. This strong GFST production in EC-Earth3-CC is clearly seen in Fig. 2. However, the CAFE data show a flatter and wider peak, indicating a longer growing season, which generates a higher mean NPP over the time period compared with both ESMs for all provinces except for GFST and the northwest Atlantic subtropical gyre (NASW) in EC-Earth3-CC (Table 1). It is also apparent from Fig. 3 that the timing of peak NPP differs between biogeochemical provinces and models (Table 1). CE1 In the CAFE data, the day of peak NPP occurs on day 164–178 (early to late June) in the northernmost provinces BPLR, ARCT, SARC, and NADR, while the subtropical gyres NASW and NASE, the Gulf Stream (GFST), and the northwest Atlantic shelf (NWCS) generate an earlier peak NPP, between day 114 (24 April) and day 130 (10 May). Similarly, in EC-Earth3-CC, the three Arctic provinces, BPLR, ARCT, and SARC, display the latest peak NPP, occurring from day 150 to day 166 (30 May to 15 June), while the peak NPP in the North Atlantic drift (NADR) occurs earlier compared with CAFE (day 124, 4 May). The earliest peak NPP occurs in the southeastern part of the domain, in NASE, on day 83 (24 March). As in the CAFE data, the earliest peak NPP in NorESM2-LM occurs in the northwest subtropical gyre (NASW) (26 April, day 116 compared with 24 April in CAFE), while the latest occurs in the continental shelf area, NWCS (day 186, 5 July). In NorESM, the three Arctic provinces display a day of peak NPP of 159 (8 June) for BPLR, 161 (10 June) for ARCT, and 176 (25 June) for SARC. The southeastern province NASE and the Gulf Stream province (GFST) have a day of peak NPP of 138 (18 May) and 148 (28 May), respectively. We note from Table 1 that the annual mean over this period is closer to CAFE in EC-Earth3-CC than in NorESM2-LM for all but one province (GFST), where the annual mean NPP is

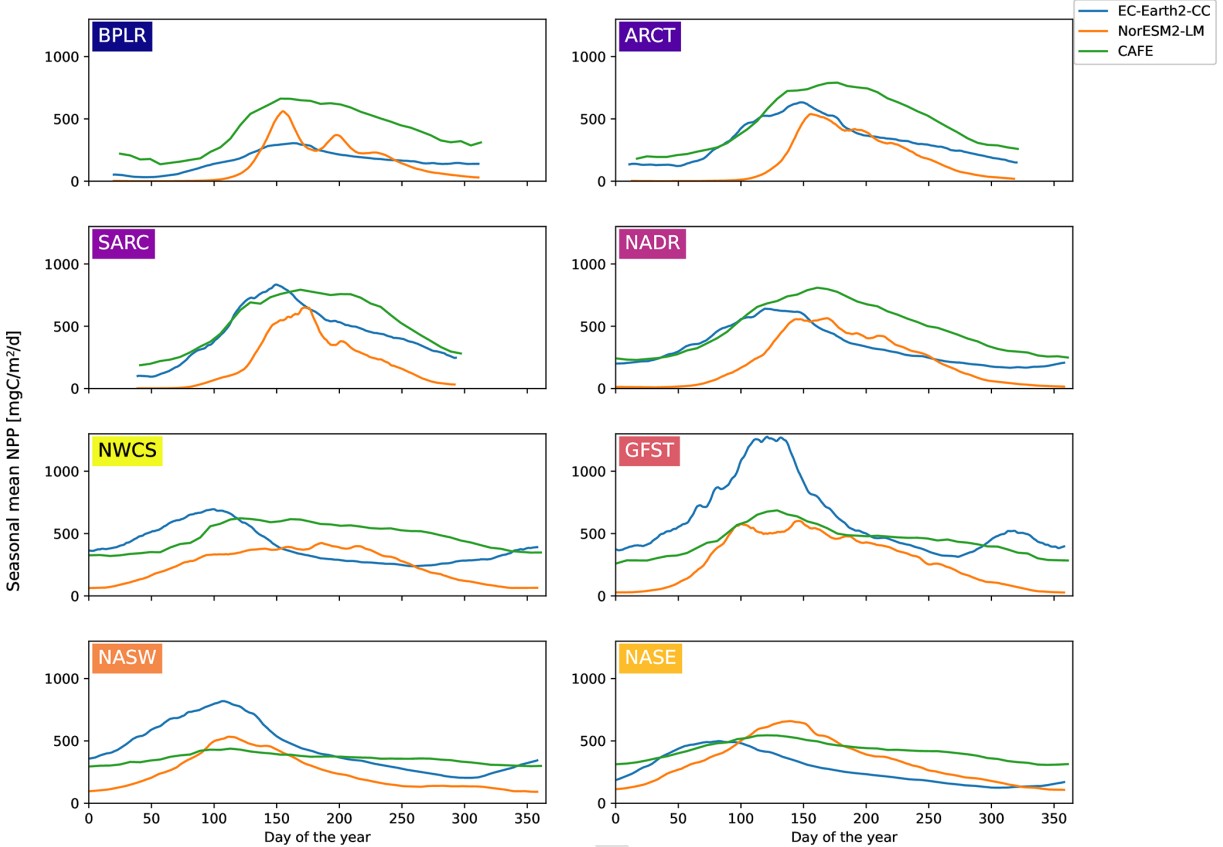

**Figure 3.** Seasonal cycles of vertically integrated NPP for CAFE (green), EC-Earth3-CC (blue), and NorESM2-LM (orange) for the different provinces shown in Fig. 1. The model data were masked by the maximum latitude present in the CAFE data to account for the smaller winter domain visible by satellites. A multi-year (2003–2021) average is shown. The ESM data are an 8 d moving average.

**Table 1.** Mean NPP and mean day of peak NPP over the time period 2003–2021 for the different provinces shown in Fig. 1. Also shown are the mean values averaged over the entire domain (Total). The ESM data were masked to the real-valued CAFE data.

| Province | CAFE | | EC-Earth3-CC | | NorESM2-LM | |
|---|---|---|---|---|---|---|
| | Mean NPP [$\mathrm{mg\,C\,m^{-2}\,d^{-1}}$] | Day of peak NPP | Mean NPP [$\mathrm{mg\,C\,m^{-2}\,d^{-1}}$] | Day of peak NPP | Mean NPP [$\mathrm{mg\,C\,m^{-2}\,d^{-1}}$] | Day of peak NPP |
| BPLR | 405 | 155 | 161 | 166 | 141 | 159 |
| ARCT | 470 | 179 | 321 | 152 | 160 | 161 |
| SARC | 525 | 171 | 442 | 150 | 210 | 176 |
| NADR | 472 | 163 | 332 | 124 | 203 | 172 |
| NWCS | 477 | 122 | 396 | 100 | 239 | 186 |
| GFST | 441 | 130 | 608 | 126 | 276 | 148 |
| NASW | 358 | 114 | 442 | 112 | 238 | 116 |
| NASE | 419 | 122 | 273 | 83 | 326 | 138 |
| Total | 424 | 161 | 371 | 121 | 242 | 153 |

38 % higher than CAFE in EC-Earth3-CC and 37 % lower than CAFE in NorESM2-LM. On the contrary, the day of peak NPP in this period is better captured by NorESM2-LM than EC-Earth3-CC in five out of eight provinces and is equally close to CAFE in one province (NASW). Here, the day of peak NPP is 2 d later than CAFE in NorESM2-LM and 2 d earlier in EC-Earth3-CC. Only in two provinces, the continental shelf NWCS and GFST, is the day of peak NPP closer to CAFE in EC-Earth3-CC.

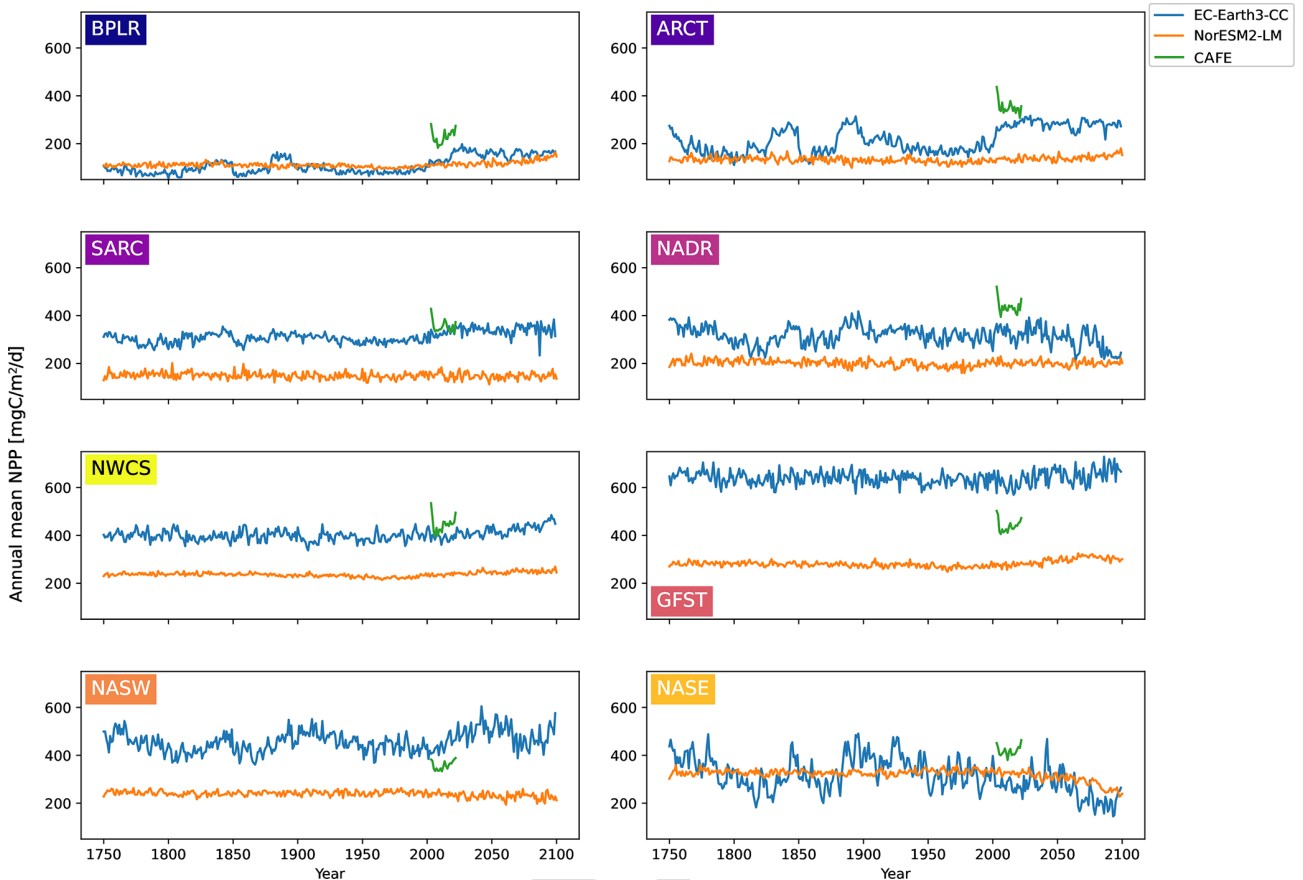

**Figure 4.** Time series of annual mean vertically integrated NPP for the different biogeochemical provinces for EC-Earth3-CC (blue), NorESM2-LM (orange), and CAFE (green).

## 3.2 Historical and future NPP

The time series of annual mean NPP for the different biogeochemical provinces from 100 years of piControl, historical, and SSP5-8.5 are shown in Fig. 4 for EC-Earth3-CC and NorESM2-LM. Also shown are the annual mean CAFE data for the period 2003–2021. The figure reveals a large interannual and multi-decadal variability in EC-Earth3-CC compared with NorESM2-LM.

For most provinces, EC-Earth3-CC generates higher annual mean NPP than NorESM2-LM, with the exception of the Arctic province, BPLR, and the south-eastern province, NASE. For BPLR, the mean for the entire period (not shown) is $110 \, \mathrm{mg \, C \, m^{-2} \, d^{-1}}$ for EC-Earth3-CC and $112 \, \mathrm{mg \, C \, m^{-2} \, d^{-1}}$ for NorESM2-LM, while for NASE, it is $314 \, \mathrm{mg \, C \, m^{-2} \, d^{-1}}$ for EC-Earth3-CC compared with $321 \, \mathrm{mg \, C \, m^{-2} \, d^{-1}}$ for NorESM2-LM. The largest difference between the two models is seen in the Gulf Stream (GFST), where EC-Earth3-CC generates a time series mean of $640 \, \mathrm{mg \, C \, m^{-2} \, d^{-1}}$, compared with $282 \, \mathrm{mg \, C \, m^{-2} \, d^{-1}}$ in NorESM2-LM. The highest NPP in NorESM2-LM is instead found in the southeastern province, NASE.

The standard deviation for the entire period is in EC-Earth3-CC between $23.8$ and $71.6 \, \mathrm{mg \, C \, m^{-2} \, d^{-1}}$ depending on the province (not shown). The largest standard deviation is found in the eastern subtropical gyre (NASE), and the lowest in the western continental shelf province, NWCS. In contrast, the standard deviation in NorESM2-LM is between $9.17$ and $22.0 \, \mathrm{mg \, C \, m^{-2} \, d^{-1}}$, similar to EC-Earth3-CC, with the largest found in NASE and the lowest in NWCS.

To find how the NPP and the timing of peak NPP has changed over the time series, we compared the last 30 year period of SSP5-8.5 (2070–2099, 2085s in the following) with the first 30 year period of the historical simulation (1850–1879, 1865s in the following). The results are summarized in Table 2. EC-Earth3-CC shows an increased NPP for most provinces, with the exception of the North Atlantic drift (NADR) and the south -eastern NASE. Here, the NPP is lower in the 2085s compared with the 1865s. In addition to those provinces, NorESM2-LM also displays decreased NPP for the western subtropical gyre (NASW) and subpolar (SARC) provinces. The day of the year of peak NPP decreases for all provinces except one in both EC-Earth3-CC (NASE) and NorESM2-LM (GFST).

**Table 2.** Mean NPP over the period 2070–2099 minus mean NPP over the period 1850–1889 together with the difference in the day of peak NPP for the same periods. Also shown in the corresponding value averaged over the entire domain ("Total").

| Province | EC-Earth3-CC 2070 to 2099 (2085s) mean minus 1850 to 1889 (1865s) mean | | NorESM2-LM 2070 to 2099 (2085s) mean minus 1850 to 1889 (1865s) mean | |
|---|---|---|---|---|
| | NPP [$\mathrm{mg\,C\,m^{-2}\,d^{-1}}$] | Day of peak NPP | NPP [$\mathrm{mg\,C\,m^{-2}\,d^{-1}}$] | Day of peak NPP |
| BPLR | 79.4 | −68.2 | 24.7 | −12.09 |
| ARCT | 125 | −25.8 | 15.4 | −20.8 |
| SARC | 48.6 | −8.84 | −2.25 | −18.0 |
| NADR | −24.1 | −7.71 | −3.19 | −10.1 |
| NWCS | 42.8 | −12.7 | 12.2 | −1.40 |
| GFST | 20.4 | −5.73 | 28.0 | 13.3 |
| NASW | 47.4 | −5.13 | −14.1 | −1.28 |
| NASE | −86.6 | 27.0 | −59.8 | −12.8 |
| Total | 12.9 | −3.91 | −12.4 | −7.48 |

To further find how the shift in phenology is distributed over the domain, the spatial distribution of the day of peak NPP averaged over the 1865s for the two ESMs is shown in Fig. 5. Also shown in this figure is the difference of the ESM
results averaged over the period 1985–2014 (2000s in the following) and the 2085s from the 1865s. In the 1865s, both ESMs displayed a pattern of later bloom in the northern parts compared with the rest of the domain. For EC-Earth3-CC, this is most notable in the Labrador Sea, while in NorESM2-
LM, the later bloom is also visible in the Gulf Stream and the northwest continental shelf area (NWCS).

The 2000s show small and scattered differences from the 1865s. In the 2085s, most of the domain experienced an earlier peak NPP but with some notable exceptions. Parts around
15 the Gulf Stream display a markedly later peak NPP in the NorESM2-LM data compared with the 1865s. This corresponds to an expansion of the pattern of late peak NPP in the same area seen in the 1865s.

In EC-Earth3-CC, the pattern of earlier peak NPP in the
20 final 30 years of SSP5-8.5 is widespread over the domain, although a notable feature is the much later bloom in the eastern subtropical gyre (NASE) (27 d on average, Table 2). The NPP in this province was greatly reduced in the 2085s compared with the 1865s ($-86.6\,\mathrm{mg\,C\,m^{-2}\,d^{-1}}$, Table 2), caused
by a strong reduction in winter surface nitrate concentration (not shown). The NPP seasonality in this area shifts from a clear spring peak to an extended period of weak NPP (not shown), with peak NPP therefore occurring later in the year. Also shown in Fig. 5 is the deviation of the 2085s from the
1865s mean divided by the standard deviation of the piControl in each grid cell, which gives a measure of the significance of the 2085s change. A two-sample Kolmogorov–Smirnov test (KS test, Python routine ks_2samp) was done to compare the distributions for the 1865s and the 2085s. Ar-
eas where these distributions were not significantly different at the 95 % level are marked in the figure. The results show

large significance in the northern parts of the domain, while the KS test generated no significant change on the 95th percentile for parts of the domain.

Taking the area average of each province allows us to look
at the mean change in the day of peak NPP as well as to identify change points. Figure 6 shows the time series of the day of peak NPP averaged over the area of each province together with the largest (cf. Fig. S2 for the largest change points using the PELT search method). In EC-Earth3-CC, the largest
change point occurs between 2002 (Arctic provinces BPLR and ARCT) and 2066 (eastern subtropical gyre, NASE) for all provinces except the western subtropical gyre (NASW), where the largest change point occurs in the year 1900. Note also that NASW is the province with the least change over the
time period (Table 2). In NorESM2-LM, the largest change point is in general located later, between 2010 (NASW) and 2082 (NASE). When increasing to two change points, the pattern of most change occurring after the year 2000 is maintained, with few change points occurring earlier (Fig. 6).
Also shown in the figure is the largest change point found by the L1 and L2 cost functions that indicates changes in the median and mean, respectively. The results show that the L1, L2, and kernel-based cost function gives almost the same results for almost all the provinces. The most discrepancy is
found in the western subtropical gyre (NASW) in EC-Earth3-CC, which is also the region displaying the least change.

Figure 7 shows the first day of the year at which the spatial mean mixed-layer depth (MLD) shoals to 40 m or shallower in each province. Similar to the results of the day of
65 peak NPP (Fig. 6), the day of MLD shallower than 40 m occurs progressively earlier over SSP5-8.5 for most provinces and for both EC-Earth3-CC and NorESM2-LM. The largest change point in the time series (Table 3) occurs between 1997 (Gulf Stream, GFST) and 2067 (western subtropical
gyre, NASW) for EC-Earth3-CC and between 2025 (Arctic province, ARCT and the North Atlantic drift, NADR) and

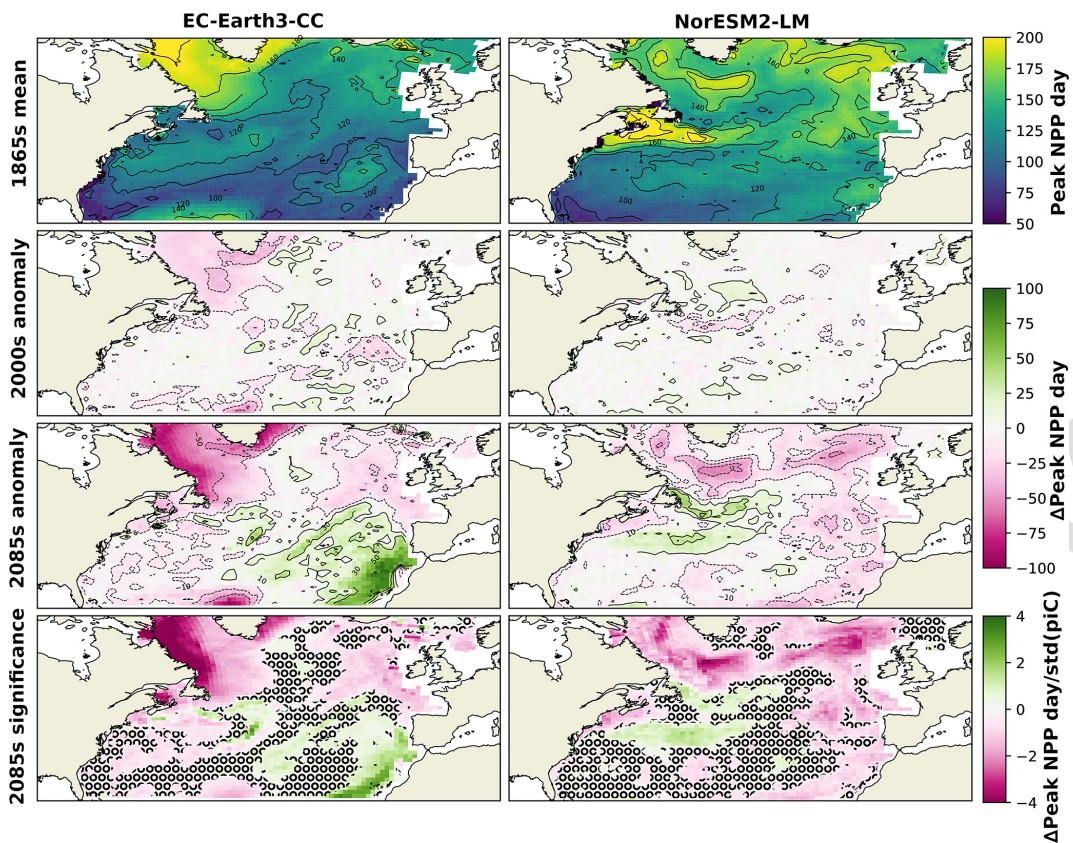

**Figure 5.** Mean day of peak NPP for NorESM2-LM (left) and EC-Earth3-CC (right) over the years 1850–1879 (top). The second panels show the mean over 1985–2014 minus the 1850–1879 mean. The third panels show the mean over 2070–2099 minus the 1850–1879 mean (bottom). The bottom panels show the results from the third panels normalized by the yearly standard deviation of the day of peak NPP in the respective piControl simulations, giving a view of how large the changes are compared with unforced variability. Grid cells that do not show significance on the 95th percentile are marked with a black ring pattern. The difference between the contour lines is 20 d.

**Table 3.** The table presents the largest change points of the day of peak NPP and the day of MLD<40 m time series for the different provinces shown in Fig. 1.

| Province | EC-Earth3-CC Largest change point [year] | | NorESM2-LM Largest change point [year] | |
|---|---|---|---|---|
| | Day of peak NPP | Day of MLD < 40 m | Day of peak NPP | Day of MLD < 40 m |
| BPLR | 2002 | 2001 | 2032 | 2031 |
| ARCT | 2002 | 2001 | 2050 | 2025 |
| SARC | 2036 | 2033 | 2049 | 2040 |
| NADR | 2017 | 2038 | 2061 | 2025 |
| NWCS | 2004 | 2056 | 2065 | 2092 |
| GFST | 2025 | 1997 | 2061 | 2069 |
| NASW | 1900 | 2067 | 2010 | 2031 |
| NASE | 2066 | 2064 | 2082 | 2028 |

2092 (western continental shelf, NWCS) for NorESM2-LM. Increasing to two change points, the pattern is consistent with most points located after the year 2000 (Fig. 7). Note that the choice of 40 m is arbitrary. We have tested for other cut-off depths, with similar results (Figs. S3–S5).

But how well do change points in the spatial mean day of peak NPP of the different provinces represent the separate grid points? The year during which the largest change point for every grid point occurs is shown in Fig. 8. The results broadly correspond to the results seen in the spatial

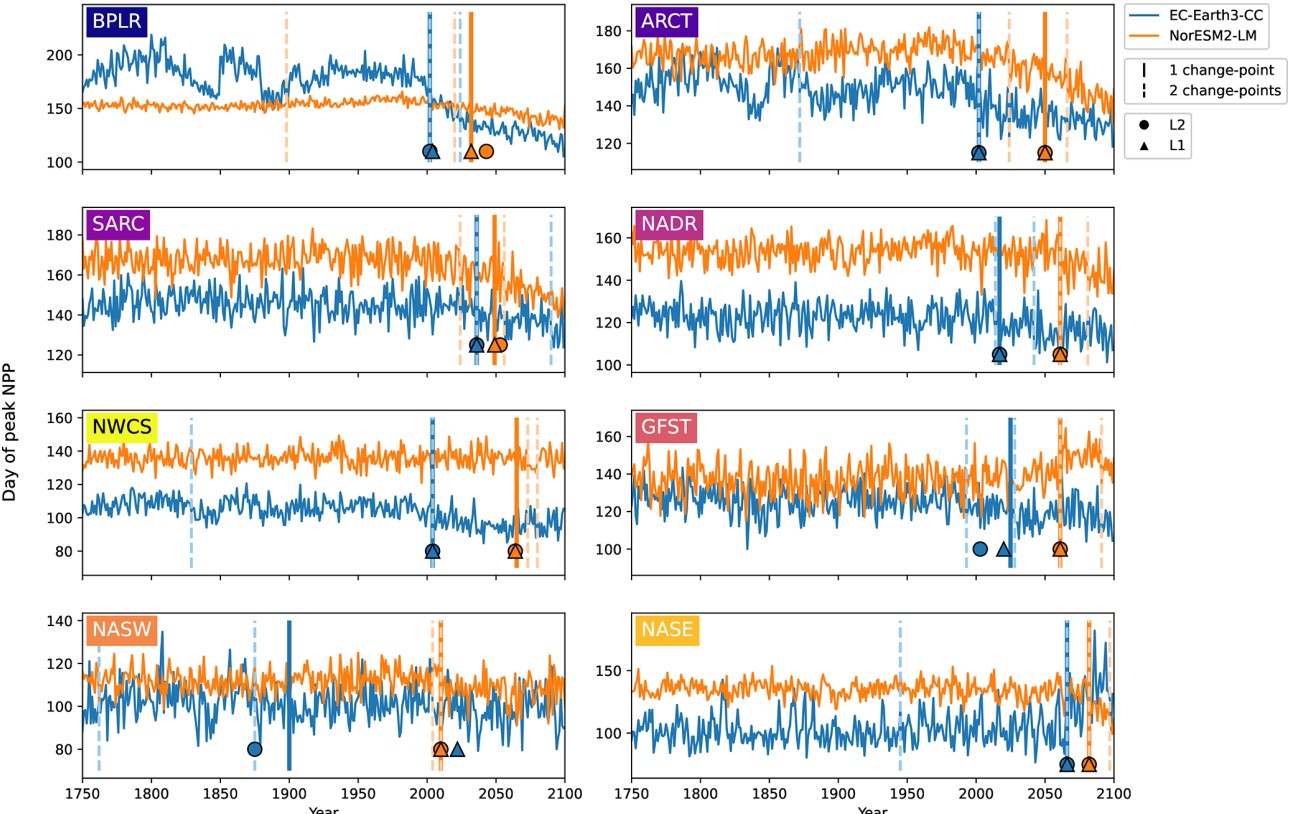

**Figure 6.** Day of peak NPP per province for EC-Earth3-CC (blue) and NorESM2-LM (orange). Note the differing *y* axes. The largest change points (calculated with a kernel-based cost function) in the time series are marked by the vertical lines. The single largest change point is marked by solid lines, and the two largest are marked with dashed lines. The center of the circles represents the largest change point in the time series that corresponds to a change in the mean (L2), while the center of the triangles represents the largest change point corresponding to a change in the median (L1).

mean time series, with the largest change points occurring after the year 2000. Few grid points display a change point earlier than that. Note that many of the grid cells displaying change points early in the time series correspond to cells
where the PELT search method could not find only one single change point (Fig. S6). This points to the fact that, in these grid points, little significant change occurs (cf. Fig. 5. bottom panel). Furthermore, EC-Earth3-CC displays an earlier largest change point for most grid points as compared with
NorESM2-LM. The northern part of the domain, where the euphotic zone is more vigorously coupled to the deep sea by vertical mixing, such as the Labrador Sea, northern North Atlantic, and sub-polar gyre, shows the earliest change point in the EC-Earth3-CC results close to the year 2000. The south-
eastern part of the domain displays the latest change point in both NorESM2-LM and EC-Earth3-CC.

To elucidate on the correlation between the day of MLD shallower than 40 m and the day of max NPP, the cross-correlation (MATLAB routine crosscorr) between the time
series shown in Figs. 6 and 7 has been plotted in Fig. 9. The figure shows a notable correlation, well above the 95 % confidence bound, between the two indices in most provinces for

both ESMs. The maximum correlation occurs for zero lag in most provinces, indicating, as expected, that peaks in these variables tend to occur within the same year. Note that the 25 strongest correlation for zero lag, at least in EC-Earth3-CC, is seen in the west wind provinces, GFST, NADR, NASW, and NASE, which have a Sverdrup-like circulation. These are also open ocean provinces where mixed layers can be expected to be less constrained by freshwater fluxes from land. 30 Furthermore, a striking feature is the strong negative correlations found in the northern provinces, BPLR and ARCT, in EC-Earth3-CC. Looking at Fig. 7, we find that the day of MLD shallower than 40 m, at least in the BPLR province, occurs so early in the year that it can hardly affect the day 35 of peak NPP, thus suggesting that the anti-correlation between these variables is owing to a hidden variable affecting both NPP and MLD. The similar correlation structure between BPLR and ARCT strongly suggests that the same is true about the ARCT province. 40

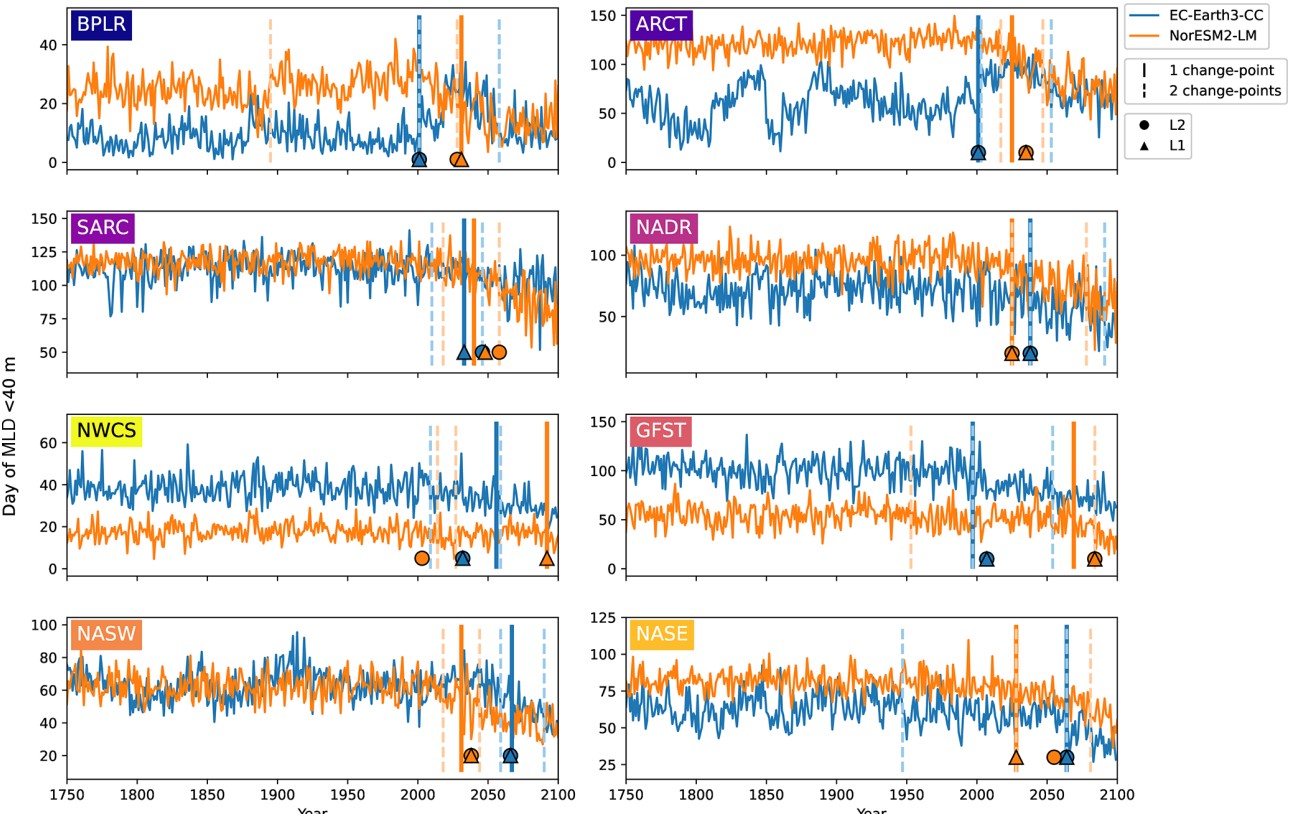

**Figure 7.** First day of the year when the mixed layer is 40 m or less for EC-Earth3-CC (blue) and NorESM2-LM (orange). Note the differing *y* axes. The largest change points (calculated with a kernel-based cost function) in the time series are marked by the vertical lines. The single largest change point is marked by solid lines, and the two largest are marked with dashed lines. The center of the circles represents the largest change point in the time series that corresponds to a change in the mean (L2), while the center of the triangles represents the largest change point corresponding to a change in the median (L1).

## 4   Discussion

The comparison between CAFE and the two ESMs showed large differences between the three data sets (Figs. 2 and 3). However, the annual mean NPP is of the same order of magnitude, and the day of peak NPP is well captured for most regions (Table 1). The regional difference in NPP is larger in the ESMs compared with the CAFE data, which is evident from the difference in annual mean between the provinces (Table 1). Peak NPP occurs latest in the year for the Arctic provinces (BPLR, ARCT, and SARC) in EC-Earth3-CC, which corresponds well to the CAFE data (although the peak NPP in NADR occurs later than for the BPLR in CAFE).

Most provinces display an increased NPP over SSP5-8.5 for EC-Earth3-CC, while for NorESM2-LM, four provinces showed an NPP increase and four displayed a decrease (Fig. 4, Table 2). Averaged over the entire domain, NPP in EC-Earth3-CC is slightly higher in the 2085s than in the 1865s and slightly lower in NorESM2-LM. The results are in line with the results of Tagliabue et al. (2021), which showed a 16 CMIP6 ESM mean NPP increase in the polar region, broadly corresponding to the Longhurst provinces

BPLR, ARCT, and SARC, where both EC-Earth3-CC and NorESM2-LM display increased NPP between the 2085s and the 1865s (though NorESM2-LM displays a slight decrease in the subpolar province SARC). Note, however, that most of the increase presented in Tagliuabue et al. (2021) seems to occur higher up in the Arctic than what is presented here. For the regions presented in Tagliabue et al. (2021) that can be broadly compared with the rest of our domain, the CMIP6 model mean displays a decline in NPP. Though only two out of five provinces in EC-Earth3-CC and three out of five provinces in NorESM2-LM display a decline, the decline is larger than the increase shown in the provinces displaying increased NPP (Table 2). Note also that the results presented in Tagliabue et al. relate to the reference period 1995–2014, which will impact the comparison to some degree. However, our results show little change before this period (Fig. 5), thus the difference might not be that significant.

The results showed that the most change in the day of peak NPP, as well as in the day of MLD shallower than 40 m, occurs after the beginning of the 21st century (Figs. 6–8), which is consistent with the results of Henson et al. (2009),

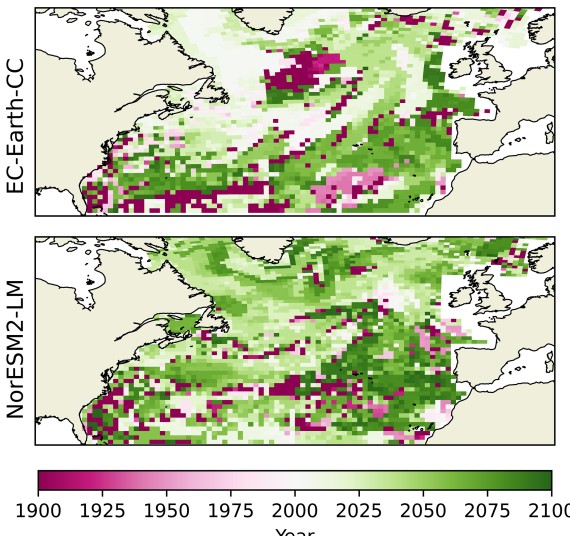

**Figure 8.** Year of change point of the day of maximum primary production for all grid spaces. Here, the change-point algorithm is set to look for only one change point.

who found no long-term trend in the subpolar North Atlantic toward earlier or later blooms in model data spanning 1959–2004. The earlier bloom displayed for most provinces (Table 2) under SSP5-8.5 is in agreement with Asch et al.
[5] (2019). They showed that blooms north of 40° N shifted earlier under RCP8.5 using 5-daily output from GFDL ESM2M including the biogeochemical model TOPAZ2.0. In contrast, Henson et al. (2018) reports, using an ocean-only model (MEDUSA-2.0, NEMO), a start of bloom shifting later in
[10] the year under RCP8.5 in most parts of the North Atlantic. However, both studies relate to surface chlorophyll and not to NPP, which makes the comparison problematic. Although surface chlorophyll has the benefit of being easily validated to observations, it is not, in a simple manner, connected to
[15] vertically integrated NPP. The chlorophyll maxima can be found below the surface (Cornec et al., 2021), and the relationship between the surface concentration and the subsurface profile differs between different localities (Sathyendranath et al., 1995). The seasonality of peak NPP is there-
[20] fore not necessarily directly relatable to the seasonality of surface chlorophyll. Moreover, our temporal resolution is higher, and both Henson et al. (2018) and Asch et al. (2019) use the start of bloom as well as length of bloom as a phenological indicator instead of the timing of the annual peak,
[25] which further complicates the comparison.

In EC-Earth3-CC, the largest change points in the day of peak NPP in many provinces occur already in the historical simulation, or in the early scenario simulation (Table 3), before the very high-emissions scenario SSP5-8.5 has started
[30] to diverge from the more moderate-emissions scenarios in terms of global mean surface temperature (O'Neill et al., 2016, Riahi et al., 2017). The effect of lower greenhouse gas

concentrations on the locations of the change points might therefore not be that large. These results point to significant phenological change that may have already started in this region and underline the need for long-term monitoring campaigns in this area. [35]

In NorESM2-LM, however, the largest change points in the day of peak NPP occur later, in the mid to late 21st century (Table 3). Only in three provinces (the western subtropical gyre, NASW, and the northern provinces, BPLR and [40] SARC) do the largest change points occur before 2050. For NorESM2-LM, a lower-emissions scenario might therefore generate even later change points for most provinces. Note that the same change points are generally found regardless [45] of which cost function (L1, L2, or kernel based) is chosen in both models. This indicates that the change points found are affecting multiple statistical moments.

The day of MLD shallower than 40 m displays a similar pattern of the largest change points generally occurring after the year 2000 (Table 3). We compared the day of peak [50] NPP with the day of MLD shallower than 40 m, and the cross-correlation showed the strongest correlation at zero lag. The fact that we saw significant correlations also with much longer lags likely reflected the low-frequency cycles [55] of the Atlantic multi-decadal variability that affects many physical parameters, such as Sea-Surface Temperature (SST) and MLD, on multi-decadal time scales (e.g., Börgel et al., 2020). Both this type of low-frequency variability and anthropogenic climate change could act as a hidden variable [60] that through, e.g., temperature and sea-ice changes, drives coherent changes in both NPP and MLD on a range of both positive and negative lags. Furthermore, we noted a strong anti-correlation from Fig. 9 between the MLD and NPP phenology for the Arctic provinces BPLR and ARCT in EC- [65] Earth3-CC. Given that both the provinces are far to the north and that SSP5-8.5 is a very strong warming scenario, we speculate that changes in sea ice could be behind the observed correlation structure. This is supported by the fact that EC-Earth3-CC has been shown to overestimate sea ice [70] concentrations in the Labrador Sea (Döscher et al., 2022). However, given that the timing of the MLD shallowing is unlikely to be important for the timing of peak NPP in these provinces, we did not investigate further.

In both models and in nature, NPP and its timing is de- [75] pendent on many other factors beyond the MLD, including light availability, nutrient concentrations, and temperature. MLD can similarly both be affected by and affect some of these factors. In light of this, MLD changes can both act as a driver of phenology changes in itself and as [80] a proxy for other drivers, which complicates the interpretation. The cross-correlation analysis therefore does not point to the validity of a certain bloom timing theory such as critical depth hypothesis or disturbance recovery theory (Behrenfeld, 2010), but it does highlight the covariance of NPP and [85] MLD phenology.

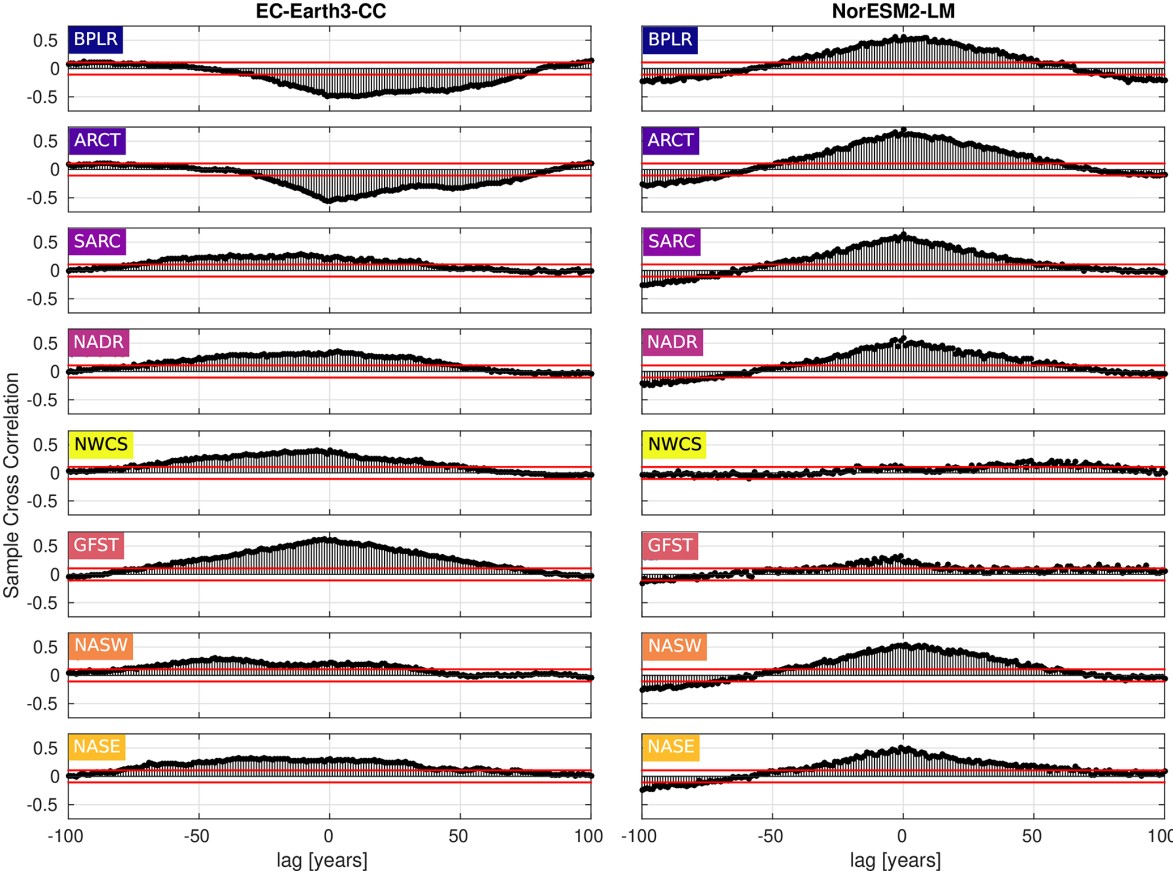

**Figure 9.** Cross-correlation between the day of peak NPP and the first day of mixed-layer depth (MLD) shallower than or equal to 40 m. Negative lag means that the day of peak NPP proceeds the first day of MLD shallower than 40 m, while the opposite holds for positive lag. The horizontal blue lines mark the 95 % confidence bounds.

The biogeochemical modules included in the Earth system models are by necessity simplistic, with PISCES simulating two phytoplankton functional types representing nanophytoplankton and diatoms, and HAMOCC only one. However, even with reduced complexity, interpretation is not straightforward. Compared with observations, community structure has been shown to affect NPP, and models containing a more dynamic phytoplankton community have a more nonlinear response to climate change owing to decreases in large cells and an increasing amount of regenerated production (Fu et al., 2016). Thus, more complex biogeochemical models may generate different results. The simpler biogeochemical model included in NorESM2-LM may be a factor in the lower variability seen in the NPP results (Fig. 4). Different phytoplankton functional types have been shown to react differently to climate change in the North Atlantic. Kléparski et al. (2023) found a decline in large flattened diatoms and an increase in biomass of elongated, slower-sinking diatoms and dinoflagellates in six CMIP6 ESMs, which could influence the carbon export in this region.

For this work, we used fully coupled Earth system models as opposed to ocean-only models, which are often used for similar work (e.g., Henson et al., 2018, 2009). The exchange of heat, momentum, and freshwater is more accurately treated in coupled models than in ocean-only models. This affects, for example, temperature and stratification. It has also been demonstrated that interactive coupling affects the variability of these variables (e.g., Bhatt et al., 1998, Barsugli and Battisti, 1998). The biogeochemical response is therefore expected to differ in the coupled vs. uncoupled case. Because of more consistent physics with respect to uncoupled models, we believe that using coupled models might constitute an important step forward in the larger effort of trying to understand what phenology changes might occur in the future.

The North Atlantic is a region of great importance for carbon sequestration through both the solubility pump and the biological pump (Baker et al., 2022). Our results show significant changes in the seasonality of peak NPP over large parts of the domain during the 21st century. These changes could, in turn, lead to trophic level decoupling and influence higher trophic levels (Yamaguchi et al., 2022) that are not yet modeled by the ESMs. In turn, this could impact the strength of the biological pump.

## 5 Summary and conclusions

In this work, we show that the seasonality of peak NPP in the North Atlantic (25–65° N) shifts earlier during the 21st century under SSP5-8.5 in two CMIP6 Earth system models. The largest change toward an earlier day of peak NPP occurs in the northern parts of the domain for both ESMs. We separated the domain into biogeochemical provinces in accordance with Longhurst et al. (1995) to account for different local conditions. EC-Earth3-CC displays change points for many provinces already in the historical simulation, while the largest change points in the NorESM2-LM data occur in the future scenario for all but one province (NASW), which is also the one displaying the least significant change. In EC-Earth3-CC, the largest change occurs in the biogeochemical province, BPLR, and in NorESM2-LM in the Arctic province, ARCT. Moreover, the changes in the day of peak NPP are far outside the range of the natural variability diagnosed from the piControl run in large parts of the domain. The changing seasonality may have an impact on fishery yields through the mismatch of fish spawning and available resources. Furthermore, carbon sequestration in this highly productive domain may be affected by changes in ecosystem structure, in turn affecting export production and the general efficiency of the biological pump.

A comparison with the satellite-based CAFE model showed that both ESMs display deviations from the CAFE data. At least for EC-Earth3-CC, this is especially true in the Gulf Stream region. NorESM2-LM is typically better at capturing the timing of peak NPP, and EC-Earth3-CC is closer in annual average NPP in most provinces. The multi-decadal variability is smaller in NorESM2-LM than in EC-Earth3-CC.

Cross-correlation analysis showed significant correlation between the day of MLD shallower than 40 m and the day of peak NPP in most provinces. The peak correlation occurs at zero lag, but correlations are significant at many both positive and negative lags. We ascribe the large range of correlated lags to forced and unforced low-frequency variability affecting both parameters; that is, the large range of correlated lags indicate that NPP is likely controlled by other factors, in addition to MLD, which affect both variables. We also found evidence that these variables covary on multi-decadal timescales, indicating that low-frequency internal as well as forced, climate variability affect these two parameters in similar ways.

We present results for two ESMs and for one future scenario. Including daily output of the two-dimensional variable NPP in standard CMIP runs would enable more thorough analysis of different models and scenarios. Since this variable is integrated over the water column, it diminishes the risk of missing out on deeper maxima, which could be the case when investigating surface variables, such as surface chlorophyll, although we acknowledge the differences between, and importance of, both variables. Furthermore, though analysis of the high-emissions scenario gives us an upper end estimate on the changes we can expect, more moderate-emissions scenarios and more models would generate a span of possible shifts. The largest shifts in EC-Earth3-CC occurring already in the historical simulation in many provinces indicates that seasonal shifts may have already started and highlights the importance of future work on including more models and scenarios in the analysis.

The results presented in this work point to a shift toward earlier peak NPP for large parts of the North Atlantic (25–65° N) in the 21st century, with most change occurring in the northern parts of the domain. Changes in primary production phenology may impact entire ecosystems through a mismatch between zooplankton spawning and available resources (Cushing, 1990). Expanding this analysis with additional scenarios and models, preferably including more complex ecosystem dynamics, would constitute important future work.

*Code availability.* The EC-Earth3-CC code is available from the EC-Earth development portal for members of the consortium. All code related to CMIP6 forcing is implemented in the component models. Model codes developed at ECMWF, including the atmosphere model IFS, are intellectual property of ECMWF and its member states. Permission to access the EC-Earth3-CC source code can be requested from the EC-Earth community via the EC-Earth website (http://www.ec-earth.org/, last access: 27 March 2024, EC-Earth consortium, 2024) and may be granted if a corresponding software license agreement is signed with ECMWF. The repository tag for the version of EC-Earth that is used in this work is 3.3.1. Currently, only European users can be granted access due to license limitations of the atmosphere model. The component models NEMO, LPJ-GUESS, TM5, and PISM are not limited by their licenses.

The NorESM code can be accessed via Zenodo at: Seland, Ø., Bentsen, M., Olivié, D., Toniazzo, T., Gjermundsen, A., Graff, L. S., Debernard, J. B., Gupta, A. K., He, Y., Kirkevåg, A., Schwinger, J., Tjiputra, J., Aas, K. S., Bethke, I., Fan, Y., Gao, S., Griesfeller, J., Grini, A., Guo, C., Ilicak, M., Karset, I. H. H., Landgren, O., Liakka, J., Moree, A., Moseid, K. O., Nummelin, A., Spensberger, C., Tang, H., Zhang, Z., Heinze, C., Iversen, T., and Schulz, M.: NorESM2 source code as used for CMIP6 simulations (includes additional experimental setups, extended model documentation, automated input data download, and restructuring of BLOM/iHAMOCC input data), Zenodo [code], https://doi.org/10.5281/zenodo.3905091, 2020.

*Data availability.* The NorESM2-LM and EC-Earth3-CC data used for the analyses can be downloaded from Zenodo using the link: https://doi.org/10.5281/zenodo.10390601 (Hieronymus et al., 2023). The shape file defining the Longhurst provinces can be found there: https://www.marineregions.org/sources.php#longhurst (last access: 5 February 2024, Flanders Marine Institute, 2009). The CAFE data is freely available through the Ocean Productivity site (http://sites.science.oregonstate.edu/ocean.productivity/, O'Malley, 2023). The python package Ruptures can be downloaded from https://centre-borelli.github.io/ruptures-docs/ (Truong et al., 2024).

*Supplement.* The supplement related to this article is available online at: https://doi.org/10.5194/bg-21-1-2024-supplement.

*Author contributions.* JH performed the EC-Earth3-CC model run, made the analysis with contributions from MH and drafted the manuscript. MG contributed to the research design. JS performed the NorESM2-LM run. ET, RB and VS made the EC-Earth3-CC setup and contributed to the EC-Earth3-CC model run. IRB contributed with discussions and paper writing. KW assisted in setting up and running EC-Earth3-CC. All co-authors contributed to the writing of the paper.

*Competing interests.* The contact author has declared that none of the authors has any competing interests.

*Disclaimer.* The contents of this article reflect only the authors' views – the European Commission and their executive agencies are not responsible for any use that may be made of the information it contains.

Publisher's note: Copernicus Publications remains neutral with regard to jurisdictional claims made in the text, published maps, institutional affiliations, or any other geographical representation in this paper. While Copernicus Publications makes every effort to include appropriate place names, the final responsibility lies with the authors.

*Acknowledgements.* The authors thank associate editor Stefano Ciavatta, reviewer Lee de Mora, and one anonymous reviewer for thorough reviews that have improved the quality of this work.

This project has received funding from the European Union's Horizon 2020 research and innovation programme under grant agreement no. 820989 (project COMFORT, Our common future ocean in the Earth system – quantifying coupled cycles of carbon, oxygen, and nutrients for determining and achieving safe operating spaces with respect to tipping points). Raffaele Bernadello acknowledges support from the European Union's Horizon 2020 research and innovation programme under Marie Skłodowska-Curie grant agreement no. GA 708063 (NetNPPAO).

*Financial support.* This research has been supported by the Horizon 2020 (grant no. 820989).

*Review statement.* This paper was edited by Stefano Ciavatta and reviewed by Lee de Mora and one anonymous referee.

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

**Remarks from the language copy-editor**

CE1    Please give an explanation of why this and the table needs to be changed and explain the difference. We have to ask the handling editor for approval. Thanks.

**Remarks from the typesetter**

TS1    Please confirm.

TS2    Please confirm.

TS3    Please provide date of last access (day month year).

TS4    Please confirm.

TS5    Please confirm.

TS6    Please confirm.

TS7    Please provide date of last access (day month year).