# Peer review of "Net primary production annual maxima in the North Atlantic projected to shift in the $21^{st}$ century."

_Biogeosciences, 2023_

## Referee Comment (RC2)

**North Atlantic patterns of primary production and phenology in two Earth System Models.**

**bg-2023-54**

Jenny Hieronymus et al.

**Reviewer summary:**

In this work, the authors use daily integrated net primary production data from two CMIP6 models and compare them against an observational dataset derived from satellite. They use change points analysis to highlight changes in the mean NPP and MLD time series. The model and observational data are compared but the comparisons are often subjective, leaving it up to the reader to decide by eye whether the models successfully reproduce the observed behaviour.

However, the authors do not describe the limitations of their work, leaving several key questions unasked or unanswered. In addition, the absence of a discussions section, means that results are often described but not fully interpreted or put into a wider context. This means that the overall conclusions

On the other hand, the language in this paper is excellent. It is very clearly written and there are very few spelling and typographical or grammar errors – at least that I have spotted.

This is a very interesting topic and there is a lot that could be done with this high temporal resolution model data. With a bit more effort, this could be a very exciting and impactful paper and I'd encourage the authors to keep going!

**Major points:**

Here are some questions that may encourage the authors to explore new ground. I don't expect all of these questions to be answered, but they may open up some new areas of interest.

Earth System Models s  tend to have relatively simple representations of the marine Ecosystem, relative to dedicated ocean models. PISCES has two phytoplankton functional types and NorESM2 has only one.  More complicated BGC models exist, such as BFM, Planktom, ERSEM and others, who have four or more phytoplankton classes. Furthermore, it is widely known that blooms are formed from cascades of multiples species. See for instance Kleparski 2021, which uses 24 species assemblages in CPR data to investigate the spatial distribution of north atlantic blooms in observations. How does the low PFT number of these models impact their ability to model blooms?

> Kléparski, L, Beaugrand, G, Edwards, M. Plankton biogeography in the North Atlantic Ocean and its adjacent seas: Species assemblages and environmental signatures. Ecol Evol. 2021; 11: 5135– 5149. https://doi.org/10.1002/ece3.7406

Similarly, marine BGC models are often run as standard-alone Ocean only runs. What do we gain by including the rest of the earth system when investigating phenology? Is there some feedback between the ocean and the atmosphere that improves bloom timing modelling?

The main technical criticism is: why only focus on these two models and this one scenario when all of CMIP6 is available? This work either needs a convincing argument as to why it uses this two model limit, or it needs to include other models from CMIP6. Similarly for the SSP5-8.5 scenario. A quick check of ESGF shows that while there is no daily primary production data, there are 380 datasets for daily Chlorophyll (chlos) from 12 different models with picontrol, historical, or Tier 1 ScenarioMIP

data. A tool like ESMValTool could help accumulate and prepare all the data, even if you don't actually use ESMValTool for the analysis. This would allow a comparison of the phenology (for chlorophyll) between several future scenarios and models.

You've located several change points for bloom timing and MLD. What are the differences (if any) between the period before this point and after this point? For instance, you could revisit fig 1, 2 or 4 comparing the "pre- change" to the post-change for these models.

On L41, you mention that NPP is affected by precipitation, wind patterns, temperature and light, but do not investigate any of these fields. You only investigate MLD as a proxy for reduced nutrient availability. I'm not fully convinced that the argument has been made that MLD shallowing is the cause of the bloom in these models. Daily data for several of these variables should be available in CMIP6 as well.

The paper is also missing a discussions section. This could include things like:

- How well this work fits into the current state of research for this field.
- What mechanisms makes one model better than the others? Is it purely their physical behaviour or is there something about the parameterisation of the marine BGC model?
- Can you estimate how many years of data would be needed to detect these change points in CAFE?
- Could you use longer data sets (ie Continuous Plankton Recorder data) to detect change points in these regions?
- A limitations of this method section

A more thorough investigation of phenology may also be interesting, not just the bloom initiation or maxima, but the intensity and duration of blooms may also be changing in the future. See for instance: https://www.sciencedirect.com/science/article/abs/pii/S1470160X11002160

In addition, other sources of observational data are available, and would strengthen the arguments around the presence of change points:

Floats: https://www.frontiersin.org/articles/10.3389/fmars.2020.00139/full

CPR (in North Sea): https://aslopubs.onlinelibrary.wiley.com/doi/10.1002/lno.11351

Bermuda time series: https://bats.bios.edu/

**Specific comments:**

**Title:** To get more impact, state your main result in the title. Peak Net Primary Production will shift earlier in the year over the 21st century. (or something like that).

**Abstract:**

L11: Similarly to the Title – if you open the abstract with the main result, it can be more eye catching. Try not to hide your exciting result at the bottom of the abstract!

L16: "The low spatial resolution of the earth system models can explain much of such difference": Can it? How so? I'm not convinced that this argument has been made.

L20: Using SSP585 as a forecast needs to be treated carefully, as this is not a "realistic" scenario. SSP585 is the scenario with enhanced fossil fuel usage – meaning that the rate of fossil fuel emissions accelerates beyond "business as usual" - something we have fortunately not seen in the previous 8 years since the end of the CMIP6 historical period.

**Introduction:**

L27: See Le Quéré et al for more up to date reference on the carbon budget.

> Le Quéré, C., et al : Global Carbon Budget 2018, Earth Syst. Sci. Data, 10, 2141–2194, https://doi.org/10.5194/essd-10-2141-2018, 2018.

L35 and elsewhere: CO2: 2 should be subscript

L41: Can you add some references for this? How is NPP impacted by precipitation in the North Atlantic?

L55: "Depending on the onset…" This sentence needs to be more explicit. Ie, If thermal stratification occurs, then spring bloom may start earlier…" or similar.

L57: One alternative theory about the causes of bloom timing changes is the switch from positive net heat flux to negative net heat flux:

> Smyth TJ, Allen I, Atkinson A, Bruun JT, Harmer RA, Pingree RD, et al. (2014) Ocean Net Heat Flux Influences Seasonal to Interannual Patterns of Plankton Abundance. PLoS ONE 9(6): e98709. https://doi.org/10.1371/journal.pone.009870

L72: What makes it unique? Why only these two models? Why not a full CMIP6 ensemble? What do you gain by using the years 1750-1850?

**Methods:**

L76: Just to confirm, are you are using the CMIP6 dataset, intpp, from ESGF? When you calculate the mean, are you taking the cell area weighted mean?

L87: Nemo should be NEMO

L89: pCO2: 2 should be subscript

L96: Primary production is indeed growth of phytoplankton, but elsewhere you talk about net primary production. Net PP usually does include some loss terms – otherwise you mean Gross Primary Production (GPP).

134: Modis should be MODIS

136: The Change point analysis method description section (sect. 2.3) does not sufficiently explain how the method works, and gives the impression that the authors have used the Ruptures package

as a "black box". Can you give more detail on how this method works here (or perhaps in an appendix)? Same for L1 & L2 methods.

**Results:**

L159 and figure 1: Is it sensible to compare depth-integrated Net Primary production to satellite (surface) Net Primary Production? Are these data comparable?

L169: Some statistical tools would help give an objective estimation of model data fit, something like some pattern statistics or even a linear regression?

L176 – If the mask is important, you may need to indicate it in this figure (or elsewhere).

L180: "Reasonable". Once again, an objective measure of goodness of fit may be useful here.

L186 – is a version of the model with higher resolution and better agreement with observations exists, then why not use data from that one?

Fig1: The MAM bloom in EC-Earth-CC seems unrealistically high, but I'm not convinced that either model is able to reproduce the observed behaviour over this time period. A statistical comparison would allow you to state how well these models reproduce observed behaviour. We can't expect ESMs to reproduce specific observations perfectly, but broad scale decadal means should be feasible.

Fig2. The coloured lined are the multi-year mean, but what does the lightly shaded area represent?

**Section 3.2:**

L193 – Try to avoid single sentence paragraphs like this.

L195: CAFE (uppercase)

Fig 3: Are you actually showing 8 day means on a figure that spans over three centuries? It looks like the shaded areas are the 8 day mean's annual minimum and maximum. Naively, from figure 2, I would expect the minimum value of NorESM2-LM to be around 50 mgC/m2/day, but it appears to be lower than that? Similarly, the range in EC-Earth-CC has a minimum around 200 in figure 2, but it looks closer to 150 in figure 3. I'm not convince that showing the range is useful here, and it's not clear to be me what it represents. Perhaps it may be easier to show the 5-95 percentile ranges (once again – weighted by area) instead, to avoid erroneously high or low values?

Figure 4 hides a lot of the important information, only showing a little of the earlier years underneath the later years. Perhaps you could instead show some decadal averages (or various change point regimes) as semitransparent bands?

208: Is the peak NPP the best metric for this? In the past, I've seen bloom timing calculated using the maximum in the first derivative, ie,when phytoplankton is growing the fastest, or when the chlorophyll concentration rises above the long term median: See for instance:

> Philippart, C.J.M., van Iperen, J.M., Cadée, G.C. et al. Long-term Field Observations on Seasonality in Chlorophyll-a Concentrations in a Shallow Coastal Marine Ecosystem, the Wadden Sea. Estuaries and Coasts 33, 286–294 (2010). https://doi.org/10.1007/s12237-009-9236-y ,

Marie-Fanny Racault, Corinne Le Quéré, Erik Buitenhuis, Shubha Sathyendranath, Trevor Platt, Phytoplankton phenology in the global ocean, Ecological Indicators, Volume 14, Issue 1, 2012, Pages 152-163, ISSN 1470-160X, https://doi.org/10.1016/j.ecolind.2011.07.010.

Figure 5: The pane labelled 1850 is the mean of 1850-1879. Perhaps these labels should be 1850-1879 mean, 1970-1999 anomaly, and 2070-2099 anomaly. I also think that a discrete colour scale would be useful here.  I'd also like to see the CAFÉ data peak NPP day.

L241: I'm not convinced how representative the mean over the whole region is here, especially after seeing how heterogenous the phenology behaviour is in fig 5! Perhaps it would be better to define sub regions within the domain and see how they behave (ie Labrador Sea, Gulf Stream, Southern NA, Central NA, Northern NA... etc.)?

L254: The fact that several methods agree that 2010 is a change point for EC-Earth-CC should give you confidence that this is a real change. However, there isn't the same agreement in NorESM2-LM. How would you interpret this? Are the changes perhaps more real in EC-Earth-CC than in NorESM2-LM? Can this shift at 2010 be seen in any observations? How do the phytoplankton bloom and the physical drivers of the bloom differ in EC-Earth-CC before and after 2010? The second EC-Earth-CC change point is around the year 2090 – how many years after the change are required to register the change?

Figure 6:  What's the value of using 8 change points? I can't see them mentioned anywhere in the text, I recommend removing them from this figure if they aren't discussed.

Figure 6: What does it look like if you apply this same method to CAFE? There isn't as much data but there is still a couple decades. Is that enough to detect changes?

L271 and figure 7: How do you interpret change points in the pre-industrial period? If they can occur using this method, then it's hard to justify that later change points are linked to climate change without additional analysis.

Figure 7: I really like this figure, but I think it could be more effective. Perhaps you could focus in on the recent past and the future regions by limiting the colour scale to (ie) the years 2000-2100? Do we really expect change points to occur earlier than say 1950?  If white means no change single point could be found, the land colour needs to be a different colour – light grey perhaps. (You also have some ocean points occurring over the land mask, so make sure you set land zorder to be higher than the pcolormesh here and elsewhere. Similarly, the contour lines are the same style, thickness and colour as the same surface, and this is confusing.

Fig 7 caption: rephrase "White spaces are areas where a single change point could not be found.**"**

L278: I find the Smythe theory about Net heat Flux (linked above) particularly compelling – even if it may not be applicable to the open ocean. Could heat (or temperature as there is a lot of CMIP6 SST data!) play a role in bloom timing?

L279: Please be cautious with using "more and less" to describe depths. As closer to the sea floor means larger values of depth, "40m or more" can mean: "deeper than 40m" or "shallower than 40m"!

L287: You have tested for the first day below 40m but comparing figures 6 and 8 makes it look like this threshold generally occurs after the peak NPP in EC-Earth-CC. The MLD lines average around day

140, while the peak NPP seems to be around 130-135.  How can MLD shallowing occur after the bloom peak if it is the main cause of the bloom initialization?

Fig 8:  This figure could be merged with figure 6, and it would drive the MLD discussion earlier in the results section.

**Conclusions:**

L300: Unresolved Eddies? How do you know this? I'm not convinced that you've demonstrated this conclusion.

L313: "1/11 day/s in NorESM2-LM/EC-Earth3-CC" should be: "1 day in NorESM2-LM and 11 days in EC-Earth3-CC"

L314: 31/33 should be "31 and 33"

L320: First mention of fish! Maybe put something in the introduction or the discussion sections.

L322: First mention of ecosystem structure! Maybe put something in the introduction or the discussion sections.

**Code Availabililty:**

L331: What about the Ruptures python package?

L340: This link did not work for me, nor could I find it using the zenodo search bar.

---

## Author Comment (AC1)

General comments:

The addressed question of this paper is very interesting and the applied statistical package to detect shifts in phenology seems to be adequate. However, I am not convinced that the strategy of using an area-based average gives meaningful results. As mentioned by the authors, the North Atlantic shows a very heterogenous regional pattern between 30°-60°N. Longhurst classified at least 4-5 distinct oceanic biogeographical provinces. One might expect each biogeographic province to exhibit different temporal behaviour; e.g. Figure 5 clearly shows this for NorESM (a dipole structure for 2070-1850 and only an 1 day change of the mean peak NPP day over the entire domain, as stated in line 219). From my perspective, the analysis of peak NPP day averaged over the entire domain is therefore of little informational value.

I recommend to repeat the analysis with the kernel based model for smaller domains (Longhurst provinces or regions aligned with common characteristics of the ESMs?). Then it might also be possible to identify local drivers of the change of peak NPP day. I assume that the intention of the investigation on the "Day of MLD <= 40m" was to identify one of these potential drivers. This MLD analysis came as quite a surprise and its findings should be mentioned in the abstract. Again, I'm not convinced that the approach of using an average value of MLD day for the entire domain is reasonable. The relatively high cross correlation values with negative lag (in Fig. 9) might be related to this averaging.

However, I appreciate this data analysis and therefore recommend publication of this manuscript with major revisions. Please see also my specific comments.

*Authors: We thank reviewer 1 for insightful comments and ideas that will greatly improve the manuscript.*

*The region is indeed heterogeneous and the change point analysis was therefore made for every grid point to ensure a robust pattern. However, we agree with the reviewer and will divide the region into Longhurst subregions.*

Specific comments:

L75: Please motivate the analysis of the change in MLD day already in the introduction.

*Authors: We will motivate the MLD analysis in the introduction as suggested.*

L96: There is a difference between "primary production" and "net primary production". The latter is the daily growth of phytoplankton minus respiratory demand. Please use net primary production throughout the manuscript.

*Authors: This will be corrected. The confusion arises from the fact that these models do not have that distinction (no explicit phytoplankton respiration).*

L101: Skyalls et al (2019) analysis did only cover a north-south transect east of 20°W from two cruises. Thus, the good agreement with observations is only shown for a very small part of the domain. Please add this information here.

*Authors: This information will be added.*

L119: Please add that phytoplankton growth is also a function of light and temperature in iHAMOCC.

*Authors: This information will be added.*

L 124: Please replace the subtitle "Observations". CAFE is a model based on satellite data, which strictly speaking are also only results of an algorithm (i.e. a model) and not observations. Same for the subtitle 3.1

*Authors: Agreed, the titles will be changed.*

L163 : typo, capitalize Gulf

*Authors: This will be corrected*

L171: typo, delete "the" in "For the the latter…"

*Authors: This will be corrected.*

L195: Statements about the multidecadal variability of an 18-year time series should be avoided. They are meaningless.

*Authors: We will remove such statements.*

L214ff: It is difficult to reconcile the results of Fig. 4, which shows a shift towards an earlier peak NPP day in NorESM at the end of the simulation and Fig. 5, which presents an extended area with a later peak NPP day in the Gulf stream region.  In L219, you give a shift of only 1 day for NorESM between 2070 and 1850. From my point of view, the weakness of area-based averaging is clearly evident here. Could you also please provide the significance of the results of Fig. 5?  Is ±10 days significant, especially for EC-Earth?  It might be useful to also show the peak NPP day for the CAFE data set as a longitude-latitude plot. (see also comment L259)

*Authors: We will add a plot showing the day of peak NPP in a longitude-latitude plot as suggested. We will also calculate the yearly standard deviation of the peak NPP day from the PI-control, so that these changes can be compared to the range of the natural variability. Indeed there is a strong spatial heterogeneity in the response and we hope the analysis broken down into Longhurst provinces will reveal interesting regional features. Thanks for the suggestion.*

L252: Please find different symbols for "l1" and "l2" for a better readability (e.g. capitalize L1 ?)

***Authors: This will be changed as suggested.***

L255: Could you please give an explanation why L1 and L2 do not identify the 2070 change point in NorESM? Would the results be more consistent if you decrease the penalty for L1 and L2 ?

***Authors: The kernel based model does not provide us with information on the nature of the change that occurs in connection with a change point. L1 and L2 give us change points related to a change in median and mean respectively. The change picked up by the kernel based model may therefore be related to a higher order change (e.g. skewness or kurtosis) in the probability distribution.***

L259: The finding of the kernel based model for EC-Earth is consistent with the findings of Fig. 5 (12 days change in peak NPP day compared to 11 ). However, the result for NorESM is quite different (10 days instead of 1). In addition, Fig.6 seems to give a much larger change for the end of the simulation than 1 or 10 days. Could you please elaborate this issue for NorESM? Furthermore, why didn't you already show the period 2010-2029 in Fig. 5? It seems to contain a strong change pattern. Why do you use 30-year averages for Fig. 5 and here you use a 50-year average?

***Authors: Thanks for noticing! In part this is likely due to different periods being used. However, it is likely also the result of us applying unfortunate averaging. In the case of Fig. 6 and the 50yr means, we first take the spatial mean of NPP and get a time series of daily NPP. After that we take the day of the year of max NPP.***

***In the case with the 30yr means, we have first calculated the temporal mean of daily NPP over a 30 year period in every grid point so that in each grid point we have 365 NPP values (Jan 1 is the temporal mean Jan 1 over 30 years and so on). After that we calculate the day of peak NPP in each grid point and obtain Fig. 5. Only after that we do the area averaging to get the 30yr mean values you are referring to. Thus Fig. 6 shows the spatial mean of the day of max NPP, while Fig 5. shows the day of max NPP of the mean seasonal cycle of NPP over the 30 yr period.***

***Of course, the order of operations matter when the max function is used. Since our focus is on the day of max NPP, we favor having only the mean of the day of max NNP in the revised manuscript. However, the difference between the two models in this respect is interesting and the implications of having the differences we see in NORESM might be worth investigating. One possible interpretation is that the larger signal toward the end of the simulation in NORESM in Fig. 5 than in Fig. 6 could be due to large interannual variability in NNP during those late years.***

***We will also include a running mean in Fig. 6 so that differences in the mean over different periods can more easily be distinguished.***

L273: Please give this motivation to look at the MLD already in the introduction

I have to admit that I'm not familiar with the used kernel based model. However, results shown for day of peak NPP in Fig.6 seem plausible: the decreasing penalty increases the numbers of change points that are found, but the change point with highest penalty is identical with one of the two changes points (there is an overlap of the pink line with one red lines in both models). I became confused when I looked at the results of "Day of MLD <= 40 m". There is no alignment between pink/red lines. Why does the kernel based model miss the largest change point with a decreasing penalty?

*Authors: We believe this is a sort of nonlinearity that materializes. Imagine that we have a dataset that changes in mean in two equal steps at times t1 and t2 from M1 to M2, so that each change is (M2-M1)/2. Having two change points you expect to find t1 and t2, but having only one, what is the expectation (t2-t1)/2 perhaps? However, admittingly, the routine we used for change point evaluation is not easily adaptable to make quantifications of the sort that could verify or refute this hypothesis, so the explanation is clearly hand-wavy.*

*As an antidote we will in the revised manuscript also try an alternative search method where the number of change points is a user input and see if the phenomenon persists.*

L292: Could you explain why both models have relatively high cross correlations for negative lag ( -3 to -1) years?

*Authors: Our interpretation is that the fact that there is strong correlation on many both positive and negative lags is owing to the changes seen in the two variables having the same drivers (i.e. global warming). Both mixed layer depth and NNP are directly affected by warming, so there is not just a simple NNP=f(MLD) but also at least one lurking variable in temperature. There are, of course, even more variables at play, but the reverse coupling that could exist i.e. NNP affecting light penetration and thus MLD, is not used in the models.*

L313: Here just to summarize that the change in peak NPP day is 1 day for NorESM is not convincing. For a different time period (1960-2010) versus (2010-2060) it was 10. It would be helpful to provide a little more discussion.

*Authors: This was discussed above (review comment relating to L259) and will be addressed in the revised version.*

L316 Please share your view on the results of the cross correlation analysis. Does it support the theory of Behrenfeld and Boss?

*Authors: The cross correlation between the day of peak NPP and the first day of MLD above 40 m indicates a high correlation between the two on an annual basis. Even though the correlation of course is not an indication of causality the results are in line with the Disturbance Recovery Hypothesis (Behrenfeld, 2010; Behrenfeld and Boss, 2014) as well as The Critical Depth Hypothesis of when a bloom has the potential to*

*start (see review by Behrenfeld and Boss, 2017). Some more discussion around this will be added to the manuscript.*

Figure captions:

Fig. 1 Give period for the data averaging; is it 2003-2021 as in Fig 2 or 2002-2021 as stated in L135? Did you also masked the model data as in Fig 2? Please indicate in the text if you always use masked ESM data for comparison with CAFE. How large is the difference between masked and unmasked ESM data ?

*Authors: The masking is only applied in Fig 2. However, as the November values are in fact influenced by low winter light we will apply the mask in the SON maps in Fig. 1 as well. Furthermore, we will add curves showing the unmasked data to Fig. 2 or the corresponding figure for the different subregions that will be implemented in the revised version.*

Fig. 3 The overlapping colour coding is difficult. Please find a better solution (e.g. instead of full time series show only mean/min/max for each ESM model.)

*Authors: We will instead use min and max levels for each ESM.*

Fig. 5 Please add significance level

*Authors: We plan to calculate the standard deviation over the PI-control run which will give us a good indication of the natural variability.*

Fig. 6 I assume that the centre of the symbols (circle and triangle) corresponds to the year of the change point. Please add this information. Please use the same names for l1 and l2 through the manuscript and in the figures. E.g. L1 and L2, not "model l1" or "model=l1"

*Authors: This will be changed as suggested.*

Fig. 7 I assume you used the highest penalty level (corresponding to the pink line). Please note. Please also change order of figures – usually EC-Earth is the top figure.

*Authors: The penalty in every grid point was tuned to, if possible, pick up only one change point. The level of the penalty is dependent on the time-series in every grid point. The order of the figures will be changed.*

Fig. 8 The figure caption is incomplete ; add …in the median (model l1). Or just write : same as Figure 6, but for first day of ….

*Authors: The caption will be expanded as suggested.*

*References:*

*Behrenfeld, M. J. (2010). Abandoning Sverdrup ' s Critical Depth Hypothesis on phytoplankton blooms. Ecology, 91(4), 977–989.*

*Behrenfeld, M. J., Boss, E. S. (2014). Resurrecting the ecological underpinnings of ocean plankton blooms. Ann Rev Mar Sci.;6:167-94. doi: 10.1146/annurev-marine-052913-021325. Epub 2013 Sep 25. PMID: 24079309.*

*Behrenfeld, M. J., & Boss, E. S. (2018). Student's tutorial on bloom hypotheses in the context of phytoplankton annual cycles. Global Change Biology, 24(1), 55–77. https://doi.org/10.1111/gcb.13858*

---

## Author Comment (AC2)

In this work, the authors use daily integrated net primary production data from two CMIP6 models and compare them against an observational dataset derived from satellite. They use change points analysis to highlight changes in the mean NPP and MLD time series. The model and observational data are compared but the comparisons are often subjective, leaving it up to the reader to decide by eye whether the models successfully reproduce the observed behaviour.

However, the authors do not describe the limitations of their work, leaving several key questions unasked or unanswered. In addition, the absence of a discussions section, means that results are often described but not fully interpreted or put into a wider context. This means that the overall conclusions

On the other hand, the language in this paper is excellent. It is very clearly written and there are very few spelling and typographical or grammar errors – at least that I have spotted. This is a very interesting topic and there is a lot that could be done with this high temporal resolution model data. With a bit more effort, this could be a very exciting and impactful paper and I'd encourage the authors to keep going!

***Authors: We thank the reviewer for their thorough and extensive review. The excellent suggestions will help us deepen the analysis and improve the results. Our responses in bold italics below.***

**Major points:**
Here are some questions that may encourage the authors to explore new ground. I don't expect all of these questions to be answered, but they may open up some new areas of interest.

Earth System Models s tend to have relatively simple representations of the marine Ecosystem, relative to dedicated ocean models. PISCES has two phytoplankton functional types and NorESM2 has only one. More complicated BGC models exist, such as BFM, Planktom, ERSEM and others, who have four or more phytoplankton classes. Furthermore, it is widely known that blooms are formed from cascades of multiples species. See for instance Kleparski 2021, which uses 24 species assemblages in CPR data to investigate the spatial distribution of north atlantic blooms in observations. How does the low PFT number of these models impact their ability to model blooms?

Kléparski, L, Beaugrand, G, Edwards, M. Plankton biogeography in the North Atlantic Ocean and its adjacent seas: Species assemblages and environmental signatures. Ecol Evol. 2021; 11: 5135– 5149. https://doi.org/10.1002/ece3.7406

***Authors: Earth system models by necessity include simple representations of the biogeochemical system. All models will include fewer PFTs than reality and the PFTs that are included are aggregates of many taxa. So how does the number of PFTs impact the vertically integrated NPP? It is possible that more PFTs would have an effect on the NPP and on the seasonal timing. However, more PFTs also introduce challenges as more variables means more variables to tune and validate. We will add some discussion about this issue in the revised version***

Similarly, marine BGC models are often run as standard-alone Ocean only runs. What do we gain by including the rest of the earth system when investigating phenology? Is there some feedback between the ocean and the atmosphere that improves bloom timing modelling?

*Authors: The physics of the coupled atmosphere ocean system differs from that of the ocean forced at the surface. In the former, the atmosphere and the ocean continuously affect each other and the temperature, currents, stratification are therefore different from the uncoupled system. In particular the air-sea exchanges of momentum, heat, and freshwater are profoundly different in coupled and uncoupled models, affecting especially SST, SSS and mixed layer depths.*

*Furthermore, it has been demonstrated that interactive coupling significantly influences the variability of thermal variables like heat exchange, temperature etc. (e.g. Bhatt et al. 1998, Barsugli and Battisti, 1998). The coupling effect is less constrained for precipitation (salinity does not impact local precipitation while precipitation impacts directly on salinity).*

*These differences affect the biogeochemistry and the seasonal pattern of primary production is therefore not expected to be equal in the coupled vs the uncoupled case. As the coupled runs in this case are concentration driven, atmospheric $pCO_2$ is however not different from the uncoupled version.*

The main technical criticism is: why only focus on these two models and this one scenario when all of CMIP6 is available? This work either needs a convincing argument as to why it uses this two model limit, or it needs to include other models from CMIP6. Similarly for the SSP5-8.5 scenario. A quick check of ESGF shows that while there is no daily primary production data, there are 380 datasets for daily Chlorophyll (chlos) from 12 different models with picontrol, historical, or Tier 1 ScenarioMIPdata. A tool like ESMValTool could help accumulate and prepare all the data, even if you don't actually use ESMValTool for the analysis. This would allow a comparison of the phenology (for chlorophyll) between several future scenarios and models.

*Authors: Daily surface chlorophyll (chlos) has the advantage that it is easily validated against satellite data and it is a common output from ESMs. However, NPP and surface chlorophyll are not connected in a simple manner. NPP is the rate of photosynthetic carbon fixation and it is integrated over the water column. It is thus a measure of the total water column production. This means that the primary production maxima can occur deeper down in the water column but still be picked up by the NPP variable (Richardson and Bendtsen, 2019). Furthermore, chlorophyll is not directly related to biomass or primary production and in PISCES chlorophyll in phytoplankton is modeled as a separate variable in accordance with Geider et al (1997). In that sense, primary production is more straightforward and more easily relatable to carbon dioxide fixation.*

*Moreover, the two runs used in the manuscript, in particular their output, were purposely made as part of the COMFORT project to investigate abrupt changes in ocean biogeochemistry. To this end we save not only high temporal resolution NPP, but also MLD, SST and some other biogeochemical variables. An important part of the work presented was to investigate whether NPP changes could be connected to changes in other variables. This would not be possible with data from the CMIP archive, as most of those variables will not be saved with high temporal resolution.*

*In the revised manuscript, we will include a motivation for the use of vertically integrated NPP.*

You've located several change points for bloom timing and MLD. What are the differences (if any) between the period before this point and after this point? For instance, you could revisit fig 1, 2 or 4 comparing the "pre- change" to the post-change for these models.

*Authors:  The largest change point in the time series marks the point in time of the largest change of the probability distribution. This could be a change in mean or median but could also be higher order changes like skewness or kurtosis. Since the kernel based model does not generate information on the type of change, the analysis was complemented with the L1 and L2 models that give change points in mean and median respectively. The alignment or non alignment of these different methods therefore give an indication of the type of change. From a more biogeochemical perspective it seems clear that the early change point is a change toward earlier peak NPP. We will add more discussion about this in the revised manuscript.*

On L41, you mention that NPP is affected by precipitation, wind patterns, temperature and light, but do not investigate any of these fields. You only investigate MLD as a proxy for reduced nutrient availability. I'm not fully convinced that the argument has been made that MLD shallowing is the cause of the bloom in these models. Daily data for several of these variables should be available in CMIP6 as well.

*Authors: We have looked at daily SST and MLD both absolute values and phenological indicators and their relation to NPP.  No clear cut correlations were found between SST and NPP or their phenology. Note also that MLD is directly dependent upon all the mentioned variables, so these different drivers should not be considered independent. We will add a deeper discussion about how different drivers affect NPP in the revised version.*

The paper is also missing a discussions section. This could include things like:

- How well this work fits into the current state of research for this field.
- What mechanisms makes one model better than the others? Is it purely their physical behaviour or is there something about the parameterisation of the marine BGC model?
- Can you estimate how many years of data would be needed to detect these change points in CAFE?

- Could you use longer data sets (ie Continuous Plankton Recorder data) to detect change points in these regions?
- A limitations of this method section

A more thorough investigation of phenology may also be interesting, not just the bloom initiation or maxima, but the intensity and duration of blooms may also be changing in the future. See for instance:
https://www.sciencedirect.com/science/article/abs/pii/S1470160X11002160

*Authors: The discussion section is currently put together with the results section in Section 3, Results and discussion. We will add a separate Discussions section in which we will add more discussion as indicated in the following responses.*

In addition, other sources of observational data are available, and would strengthen the arguments around the presence of change points:

Floats: https://www.frontiersin.org/articles/10.3389/fmars.2020.00139/full

CPR (in North Sea): https://aslopubs.onlinelibrary.wiley.com/doi/10.1002/lno.11351

Bermuda time series: https://bats.bios.edu/

*Authors: These are great references that we will add to the discussion. However, comparing vertically integrated NPP to observations is difficult. Most sources use phytoplankton biomass or chlorophyll as in the suggested references. This is also the reason why we choose to use the satellite based CAFE model for the comparison.*

*Note also that the presence of change points in the models is independent of the models ability to reproduce observations. What comparisons to observations can give us is some grasp of the model's skill. Although, even that is hard given that we compare historical simulations to our recorded history (i.e. one realization of all possible histories) and also given that the observations used are, in fact, also model dependent. However, the silver lining is that understanding and describing the behaviour of ESMs is interesting and important even in itself given their extensive use in science and their impact on policies. We will add some of these points to the introduction.*

**Specific comments:**

**Title:** To get more impact, state your main result in the title. Peak Net Primary Production will shift earlier in the year over the 21 st century. (or something like that).

*Authors: The title will be changed so that our main result is indicated there.*

**Abstract:**

L11: Similarly to the Title – if you open the abstract with the main result, it can be more eye catching. Try not to hide your exciting result at the bottom of the abstract!

*Authors: We will change this.*

L16: "The low spatial resolution of the earth system models can explain much of such difference": Can it? How so? I'm not convinced that this argument has been made.

*Authors: No, you're right. This has not been properly demonstrated. We will change this statement.*

L20: Using SSP585 as a forecast needs to be treated carefully, as this is not a "realistic" scenario. SSP585 is the scenario with enhanced fossil fuel usage – meaning that the rate of fossil fuel emissions accelerates beyond "business as usual" - something we have fortunately not seen in the previous 8 years since the end of the CMIP6 historical period.

*Authors: It is true that SSP5-8.5, in that sense, is not the most plausible scenario. However, this also gives us kind of an upper end on the size of change. Furthermore, one of the largest change points in the timeseries occurs in both models at the end of the historical simulation.*

**Introduction:**
L27: See Le Quéré et al for more up to date reference on the carbon budget.
Le Quéré, C., et al : Global Carbon Budget 2018, Earth Syst. Sci. Data, 10, 2141–2194, https://doi.org/10.5194/essd-10-2141-2018, 2018.

*Authors: This will be updated.*

L35 and elsewhere: CO2: 2 should be subscript

*Authors: This will be corrected.*

L41: Can you add some references for this? How is NPP impacted by precipitation in the North Atlantic?

*Authors: Increased precipitation and changed wind patterns affect the water column stratification. In this way, NPP is indirectly affected by these variables. Furthermore, precipitation can have a fertilizing effect by transporting nutrients to the surface waters (Myriokefalitakis et al., 2020). In particular, MLD is affected by precipitation. Freshwater added to the top of the ocean increases vertical stability through its effect on density. Fresher water needs to be cooled more to get the same density as saltier waters.*

L55: "Depending on the onset…" This sentence needs to be more explicit. Ie, If thermal stratification occurs, then spring bloom may start earlier…" or similar.

***Authors: We will change the sentence.***

L57: One alternative theory about the causes of bloom timing changes is the switch from positive net heat flux to negative net heat flux:
Smyth TJ, Allen I, Atkinson A, Bruun JT, Harmer RA, Pingree RD, et al. (2014) Ocean Net Heat Flux Influences Seasonal to Interannual Patterns of Plankton Abundance. PLoS ONE 9(6):e98709. https://doi.org/10.1371/journal.pone.009870

***Authors: We will add this reference***

L72: What makes it unique? Why only these two models? Why not a full CMIP6 ensemble? What do you gain by using the years 1750-1850?

***Authors: The Pi-Control is an important addition that underlines the uniqueness of the largest change-points. Furthermore, the benefit of using long time series is a more robust estimation in the tails of the probability distribution. Consequently, rare events such as extreme conditions are better reflected in the data set. The data used in this work was produced in the H2020 project COMFORT and the two models are the only ones that have saved daily vertically integrated primary production for the full 1750-2100 period.***

**Methods:**
L76: Just to confirm, are you are using the CMIP6 dataset, intpp, from ESGF? When you calculate the mean, are you taking the cell area weighted mean?

***Authors:***
***The EC-Earth-CC and NorESM2-LM historical and SSP5-8.5 used in this study were performed according to CMIP6 protocols (with additional daily output activated) but have not been published on ESGF. A minor update was made to the terrestrial vegetation model in EC-Earth-CC after the PiC was performed but is not something that is expected to affect the results.***
***The mean is the cell area weighted mean.***

L87: Nemo should be NEMO
***Authors: This will be corrected.***

L89: pCO2: 2 should be subscript
***Authors: This will be corrected.***

L96: Primary production is indeed growth of phytoplankton, but elsewhere you talk about net primary production. Net PP usually does include some loss terms – otherwise you mean Gross Primary Production (GPP).
***Authors: We mean NPP, that is GPP minus respiration. The models only generate NPP as they do not explicitly model respiration. This will be clarified.***

134: Modis should be MODIS

***Authors: This will be corrected.***

136: The Change point analysis method description section (sect. 2.3) does not sufficiently explain how the method works, and gives the impression that the authors have used the Ruptures packageas a "black box". Can you give more detail on how this method works here (or perhaps in an appendix)? Same for L1 & L2 methods.
***Authors: This method has been described in the cited references. However, we understand the criticism and will extend this section with a more thorough description.***

**Results:**
L159 and figure 1: Is it sensible to compare depth-integrated Net Primary production to satellite (surface) Net Primary Production? Are these data comparable?
***Authors: CAFE is a model that utilizes ocean color and other optical properties to estimate the vertically integrated NPP (Silsbe et al., 2016). These values are not just for the surface but for the entire water column in similarity with the ESM results. The vertically integrated properties is a strong argument for the method used in this manuscript.***

L169: Some statistical tools would help give an objective estimation of model data fit, something like some pattern statistics or even a linear regression?

***Authors: Given the short time series and the fact that these series are from different histories, we would argue that applying more statistical methods would not answer the relevant questions. Any method of comparison is only as good as the data admits, and this data admits very little. Put another way; say we calculate a spatial correlation between the CAFE data and our two models and find r1(EC-Earth-CAFE)<r2(NORESM-CAFE), what, of value, would that tell us?***

L176 – If the mask is important, you may need to indicate it in this figure (or elsewhere).
***Authors: We will add the unmasked data to the figure. So that the seasonal cycle for the entire region from the ESMs is visible.***

L180: "Reasonable". Once again, an objective measure of goodness of fit may be useful here.

***Authors: We will reformulate the sentence. We would, however, argue that it is easy to tell by eye that these distributions are quite different. Of course we can calculate the p-value of a two-sample Kolmogorov-Smirnov test and include that, but really it is evident that the data are drawn from different distributions.***

L186 – is a version of the model with higher resolution and better agreement with observations exists, then why not use data from that one?

***Authors: The model results were more similar to what is seen in EC-Earth-CC but not necessarily better. EC-Earth-CC displays a too early peak NPP and a too strong***

*decline after this peak. Model development of NorESM targeted this behavior resulting in a better timing of the peak. We will rewrite this to make this clearer. Note also that daily data from the higher resolution model version was not available.*

Fig1: The MAM bloom in EC-Earth-CC seems unrealistically high, but I'm not convinced that either model is able to reproduce the observed behaviour over this time period. A statistical comparison would allow you to state how well these models reproduce observed behaviour. We can't expect ESMs to reproduce specific observations perfectly, but broad scale decadal means should be feasible.

*Authors: By construction, neither of these models can replicate observed behavior over specific time periods as the internal variability of the two ESMs is not in sync with the observed one. An ocean only model forced with reanalysed atmospheric forcing could conceivably replicate observed behavior over specific time periods, coupled climate models on the other hand can be in completely different phases on NAO, ENSO, AMO and so on owing to the chaotic nature of unforced climate variability.*

Fig2. The coloured lined are the multi-year mean, but what does the lightly shaded area represent?
*Authors: The shaded areas show +- 1 standard deviation over the period 2003-2021. This information will be added to the figure caption.*

**Section 3.2:**
L193 – Try to avoid single sentence paragraphs like this.
*Authors: This will be changed.*

L195: CAFE (uppercase)
*Authors: This will be corrected.*

Fig 3: Are you actually showing 8 day means on a figure that spans over three centuries? It looks like the shaded areas are the 8 day mean's annual minimum and maximum. Naively, from figure 2, I would expect the minimum value of NorESM2-LM to be around 50 mgC/m2/day, but it appears to be lower than that? Similarly, the range in EC-Earth-CC has a minimum around 200 in figure 2, but it looks closer to 150 in figure 3. I'm not convince that showing the range is useful here, and it's not clear to be me what it represents. Perhaps it may be easier to show the 5-95 percentile ranges (once again – weighted by area) instead, to avoid erroneously high or low values?
*Authors: The shaded area shows the full area weighted mean time series of daily and 8 daily data for the ESMs and CAFE respectively.Whereas the lines shows the yearly means from the different models (i.e. the lines are a time filtered version of the full data set). We will add figures for the different subregions in the revised manuscript as suggested by both reviewers. As for the idea about the percentile ranges. Note that these are, non stationary, time series from single models, not distributions of outcomes from a model intercomparison experiment. That said, we don't understand what distribution those percentiles would show?*

Figure 4 hides a lot of the important information, only showing a little of the earlier years

underneath the later years. Perhaps you could instead show some decadal averages (or variouschange point regimes) as semitransparent bands?

**Authors: Yes, it is true that a lot of information is hidden in this figure. We will make 30 yr mean seasonal cycles instead.**

208: Is the peak NPP the best metric for this? In the past, I've seen bloom timing calculated using the maximum in the first derivative, ie,when phytoplankton is growing the fastest, or when the chlorophyll concentration rises above the long term median: See for instance: Philippart, C.J.M., van Iperen, J.M., Cadée, G.C. et al. Long-term Field Observations on Seasonality in Chlorophyll-a Concentrations in a Shallow Coastal Marine Ecosystem, the Wadden Sea. Estuaries and Coasts 33, 286–294 (2010).
https://doi.org/10.1007/s12237-009-
9236-y ,Marie-Fanny Racault, Corinne Le Quéré, Erik Buitenhuis, Shubha Sathyendranath, TrevorPlatt, Phytoplankton phenology in the global ocean, Ecological Indicators, Volume 14, Issue 1, 2012, Pages 152-163, ISSN 1470-160X,
https://doi.org/10.1016/j.ecolind.2011.07.010.

**Authors: We claim no optimality for this metric. There are different phenological indicators that could be used. The timing of the peak is a well defined property that has been used before (eg. Nissen and Vogt, 2021; Henson et al., 2013) while several different methodologies, giving different results, exist for the timing of the start (Thomalla et al., 2015). We have tried some other phenological indicators but found the day of peak NPP to be a more robust metric for this data set. Max is generally robust unless the distribution is bimodal, and two peaks have similar magnitude, which we did not see. First derivatives are typically spiky, concentration based cut-offs are a bit arbitrary, and different cut-offs would likely be needed for the two models.**

Figure 5: The pane labelled 1850 is the mean of 1850-1879. Perhaps these labels should be 1850-1879 mean, 1970-1999 anomaly, and 2070-2099 anomaly. I also think that a discrete colour scale would be useful here. I'd also like to see the CAFÉ data peak NPP day.

**Authors: We will change the labels and experiment with discrete color scales. We will also add CAFE day of peak NPP.**

L241: I'm not convinced how representative the mean over the whole region is here, especially after seeing how heterogenous the phenology behaviour is in fig 5! Perhaps it would be better to define sub regions within the domain and see how they behave (ie Labrador Sea, Gulf Stream, Southern NA, Central NA, Northern NA... etc.)?

**Authors: We agree and will define subregions as was also suggested by reviewer 1.**

L254: The fact that several methods agree that 2010 is a change point for EC-Earth-CC should give you confidence that this is a real change. However, there isn't the same agreement in NorESM2-LM. How would you interpret this? Are the changes perhaps more real in EC-Earth-CC than in NorESM2-LM? Can this shift at 2010 be seen in any observations? How do the phytoplankton bloom and the physical drivers of the bloom differ in EC-Earth-CC before and after 2010? The second EC-Earth-CC change point is around the year 2090 – how many years after the change are required to register the change?

*Authors: The kernel based model finds all changes in the probability distribution. This means that the exact nature of the change is not clear. We therefore complemented the analysis with the L1 and L2 method that finds changes in median and mean respectively. That the change points are not at the same location just means that the changes in mean, median and higher order changes do not occur at the same point in time. We think it is quite important to not overinterpret change points. In the sense that the statistics of the time series changes in some appreciable manner, all change points are real. In the sense that there is necessary a co-occurring change in the physical drivers, perhaps no change point is real. The observational time series is not long enough for it to make sense to look for change points. Moreover, the year 2010 in EC-Earth is not the same as 2010 in NORESM or in reality. What they have in common is similar levels of greenhouse gasses and aerosols in the atmosphere. Climate variability is not in phase in the two (or any other CMIP6) models nor is either model in phase with reality. Therefore, it is unlikely that 2010 has that significance. There is one possibility, however, 2010, was the year Eyjafjallajökull erupted. Perhaps this could have an imprint both in the models and in reality.*

*The algorithm generally always picks up change points at the end points of the time series. So the constraints of the kernel based method together with the pelt search algorithm used does not impose a minimum distance from the boundary for where change points can be picked up. So in a mathematical sense, change points can be found anywhere. However, our own, more subjective, constraints lead us to disregard change points at the end points of the time series. However, we have not decided on a specific distance from the end-point where change points should be dismissed. The latest change points found by the algorithm (except the end point) at year 2090, we subjectively judge to be far enough away from the boundary. The subjectivity may seem unwanted. However, a degree of subjectivity is no doubt inevitable when trying to translate a purely statistical result into a result that can have a biogeochemical meaning.*

Figure 6: What's the value of using 8 change points? I can't see them mentioned anywhere in the text, I recommend removing them from this figure if they aren't discussed.
*Authors: The reason for the inclusion of the 8 change points was to illustrate that decreasing the penalty gives change points all over the time series. We will reevaluate this after having remade the figures for subregions instead as suggested by both reviewers.*

Figure 6: What does it look like if you apply this same method to CAFE? There isn't as much data but there is still a couple decades. Is that enough to detect changes?
*Authors:  We can for sure get change points in the time series as we are able to obtain change points all over the 350 yr model time series for a low penalty. However, as the time series is very short it is difficult to say if the changes picked up are at all significant.*

L271 and figure 7: How do you interpret change points in the pre-industrial period? If they can occur using this method, then it's hard to justify that later change points are linked to climate change without additional analysis.

*Authors: It is generally speaking true that the change point analysis does not say anything about the cause of the change. Therefore, regardless of when a change point occurs you always need some additional arguments to tie it to e.g. climate change. Nevertheless, our main result is that the largest changes over this 350 yr timeseries occur in the late historical simulation or in the future scenario for the vast majority of this region as seen in Fig. 7. This is true in both models and their natural variability is not in sync, climate change is thus a very plausible candidate as the cause of these changes. We will add some more discussion around this in the revised manuscript.*

Figure 7: I really like this figure, but I think it could be more effective. Perhaps you could focus in on the recent past and the future regions by limiting the colour scale to (ie) the years 2000-2100? Do we really expect change points to occur earlier than say 1950? If white means no change single point could be found, the land colour needs to be a different colour – light grey perhaps. (You also have some ocean points occurring over the land mask, so make sure you set land zorder to be higher than the pcolormesh here and elsewhere. Similarly, the contour lines are the same style, thickness and colour as the same surface, and this is confusing.

*Authors: Change points are indeed occurring before 1950 in some grid points seen as blue colors in the figure. We thought this was interesting to show. We do however see the reviewers point that it is difficult to see variations in the green colors which constitutes the majority of the figure and we will change the time span of the colorbar. We will also change colors and line styles in accordance with the reviewers suggestions.*

Fig 7 caption: rephrase "White spaces are areas where a single change point could not be found."

*Authors: We will rephrase.*

L278: I find the Smythe theory about Net heat Flux (linked above) particularly compelling – even if it may not be applicable to the open ocean. Could heat (or temperature as there is a lot of CMIP6 SST data!) play a role in bloom timing?

*Authors: We have compared to SST but did not find any convincing correlations. We will add a discussion around this.*

L279: Please be cautious with using "more and less" to describe depths. As closer to the sea floor means larger values of depth, "40m or more" can mean: "deeper than 40m" or "shallower than 40m"!

*Authors: We will change this.*

L287: You have tested for the first day below 40m but comparing figures 6 and 8 makes it look like this threshold generally occurs after the peak NPP in EC-Earth-CC. The MLD lines average around day140, while the peak NPP seems to be around 130-135. How can MLD shallowing occur after the bloom peak if it is the main cause of the bloom initialization?

*Authors: In reality MLD shallowing in the model is gradual, but the choice of a 40 m threshold gives it a discrete date. In other words, we suggest that blooms may start when the MLD becomes sufficiently thin, not that 40 m is the magic number. The chosen number, 40 m, is just a proxy.*

Fig 8: This figure could be merged with figure 6, and it would drive the MLD discussion earlier in the results section.
***Authors: We will merge those. However, the figures will change as we will separate the analysis in different subregions as suggested by both reviewers.***

**Conclusions:**
L300: Unresolved Eddies? How do you know this? I'm not convinced that you've demonstrated this conclusion.
***Authors: It is true that this has not been properly demonstrated. Directly demonstrating this would amount to running also eddying versions of the models and compare the results, that we have not done. The sentence is intended to be more of a well founded hypothesis for future investigations. The idea is simply that the region is very much more dynamic in reality than in the models. The Gulf Stream meanders, rings are detached, etc. Such dynamics lead to spreading of water properties. In a long time-average like in fig.1 we therefore expect the CAFE data to have more spatially homogenized properties than the non-eddying models. We will rephrase and elaborate on this in the manuscript.***

L313: "1/11 day/s in NorESM2-LM/EC-Earth3-CC" should be: "1 day in NorESM2-LM and 11 days in EC-Earth3-CC"
***Authors: This will be changed.***

L314: 31/33 should be "31 and 33"
***Authors: This will be changed.***

L320: First mention of fish! Maybe put something in the introduction or the discussion sections.
***Authors: We will put something about this in the introduction and discussion.***

L322: First mention of ecosystem structure! Maybe put something in the introduction or the discussion sections.
***Authors: We will put something on this in the introduction and discussion.***

**Code Availabililty:**
L331: What about the Ruptures python package?
***Authors: A link to Ruptures will be included.***

L340: This link did not work for me, nor could I find it using the zenodo search bar.

***Authors: The link works but a space is missing after "Zenodo:" which makes it look like that word is supposed to be included in the link. We will change this.***

***References:***

***Bhatt, U. S., M. A. Alexander, D. S. Battisti, D. D. Houghton, and L. M. Keller (1998). Atmosphere–Ocean Interaction in the North Atlantic: Near-Surface Climate Variability.***

*J. Climate, 11, 1615–1632,*
*https://doi.org/10.1175/1520-0442(1998)011<1615:AOIITN>2.0.CO;2*

*Barsugli, J. J., and D. S. Battisti (1998). The Basic Effects of Atmosphere–Ocean Thermal Coupling on Midlatitude Variability. J. Atmos. Sci., 55, 477–493, https://doi.org/10.1175/1520-0469(1998)055<0477:TBEOAO>2.0.CO;2.*

*Geider, R. J., MacIntyre, H. L., and Kana, T. M. (1997). A dynamic model of phytoplankton growth and acclimation: responses of the balanced growth and Chlorophyll a : carbon ratio to light, nutrient limitation and temperature, Mar. Ecol.-Prog. Ser., 148, 187–200.*

*Myriokefalitakis, S., Gröger, M., Hieronymus, J., and Döscher, R. (2020): An explicit estimate of the atmospheric nutrient impact on global oceanic productivity, Ocean Sci., https://doi.org/10.5194/os-16-1183-2020*

*Nissen, C. and Vogt, M. (2021). Factors controlling the competition between Phaeocystis and diatoms in the Southern Ocean and implications for carbon export fluxes, Biogeosciences, 18, 251–283, https://doi.org/10.5194/bg-18-251-2021.*

*Richardson K, Bendtsen J (2019). Vertical distribution of phytoplankton and primary production in relation to nutricline depth in the open ocean. Mar Ecol Prog Ser 620:33-46. https://doi.org/10.3354/meps12960*

*Thomalla et al. (2015). High-resolution view of the spring bloom initiation and net community production in the Subantarctic Southern Ocean using glider data, ICES Journal of Marine Science, Volume 72, Issue 6, Pages 1999–2020, https://doi.org/10.1093/icesjms/fsv105*

---

## Author Response (AR1)

**Replies to reviewer 1**

General comments:

The addressed question of this paper is very interesting and the applied statistical package to detect shifts in phenology seems to be adequate. However, I am not convinced that the strategy of using an area-based average gives meaningful results. As mentioned by the authors, the North Atlantic shows a very heterogenous regional pattern between 30°-60°N. Longhurst classified at least 4-5 distinct oceanic biogeographical provinces. One might expect each biogeographic province to exhibit different temporal behaviour; e.g. Figure 5 clearly shows this for NorESM (a dipole structure for 2070-1850 and only an 1 day change of the mean peak NPP day over the entire domain, as stated in line 219). From my perspective, the analysis of peak NPP day averaged over the entire domain is therefore of little informational value.

I recommend to repeat the analysis with the kernel based model for smaller domains (Longhurst provinces or regions aligned with common characteristics of the ESMs?). Then it might also be possible to identify local drivers of the change of peak NPP day. I assume that the intention of the investigation on the "Day of MLD <= 40m" was to identify one of these potential drivers. This MLD analysis came as quite a surprise and its findings should be mentioned in the abstract. Again, I'm not convinced that the approach of using an average value of MLD day for the entire domain is reasonable. The relatively high cross correlation values with negative lag (in Fig. 9) might be related to this averaging.

However, I appreciate this data analysis and therefore recommend publication of this manuscript with major revisions. Please see also my specific comments.

*Authors: We thank reviewer 1 for insightful comments and ideas that have greatly improved the manuscript.*

*We have separated the region into Longhurst provinces which better captures the spatial heterogeneity.*

Specific comments:

L75: Please motivate the analysis of the change in MLD day already in the introduction.
*Authors: We have added a motivation for the MLD analysis in the introduction as suggested (lines 85-89).*

L96: There is a difference between "primary production" and "net primary production". The latter is the daily growth of phytoplankton minus respiratory demand. Please use net primary production throughout the manuscript.

*Authors: This has been corrected.*

L101: Skyalls et al (2019) analysis did only cover a north-south transect east of 20°W from two cruises. Thus, the good agreement with observations is only shown for a very small part of the domain. Please add this information here.

*Authors: This information has been added on line 126.*

L119: Please add that phytoplankton growth is also a function of light and temperature in iHAMOCC.

*Authors: This information has been added on line 144.*

L 124: Please replace the subtitle "Observations". CAFE is a model based on satellite data, which strictly speaking are also only results of an algorithm (i.e. a model) and not observations.  Same for the subtitle 3.1

*Authors: The titles have been changed.*

L163 : typo, capitalize Gulf

*Authors: This has been corrected*

L171: typo, delete "the" in "For the the latter…"

*Authors: The text has been removed.*

L195:  Statements about the multidecadal variability of an 18-year time series should be avoided. They are meaningless.

*Authors: We have removed such statements.*

L214ff: It is difficult to reconcile the results of Fig. 4, which shows a shift towards an earlier peak NPP day in NorESM at the end of the simulation and Fig. 5, which presents an extended area with a later peak NPP day in the Gulf stream region.  In L219, you give a shift of only 1 day for NorESM between 2070 and 1850. From my point of view, the weakness of area-based averaging is clearly evident here. Could you also please provide the significance of the results of Fig. 5?  Is ±10 days significant, especially for EC-Earth?  It might be useful to also show the peak NPP day for the CAFE data set as a longitude-latitude plot. (see also comment L259)

*Authors: We have separated the region into Longhurst provinces which better captures the spatial heterogeneity. Indeed the response is quite different between different provinces. The mean change between the last 30 yrs of SSP5-8.5 and the first 30 yrs of historical is summarized in Table. 2.*

*In the new version we added the day of peak NPP per region in Fig 3. and in Tab. 1. Given the large amount of figures, especially panels, with the new regions we thought*

*it was better not to add a whole new figure with full lat-long coverage on top of all the other new figures.*

*Regarding the significance we did add two new panels to Fig. 5 which shows the change normalized by the yearly standard deviation from the PI-control. It is readily evident that these results would be very unlikely to occur by chance.*

L252: Please find different symbols for "l1" and "l2" for a better readability (e.g. capitalize L1 ?)

*Authors: This has been changed as suggested.*

L255: Could you please give an explanation why L1 and L2 do not identify the 2070 change point in NorESM? Would the results be more consistent if you decrease the penalty for L1 and L2 ?

*Authors: The kernel based model does not provide us with information on the nature of the change that occurs in connection with a change point. L1 and L2 give us change points related to a change in median and mean respectively. The change picked up by the kernel based model may therefore be related to a higher order change (e.g. skewness or kurtosis) in the probability distribution.*

L259: The finding of the kernel based model for EC-Earth is consistent with the findings of Fig. 5 (12 days change in peak NPP day compared to 11 ). However, the result for NorESM is quite different (10 days instead of 1). In addition, Fig.6 seems to give a much larger change for the end of the simulation than 1 or 10 days. Could you please elaborate this issue for NorESM? Furthermore, why didn't you already show the period 2010-2029 in Fig. 5? It seems to contain a strong change pattern. Why do you use 30-year averages for Fig. 5 and here you use a 50-year average?

*Authors: This problem has been corrected. The results are now also summarized in Table. 2.*

L273: Please give this motivation to look at the MLD already in the introduction
*Authors: We have added a motivation for the MLD analysis to the introduction (lines 85-89)*

I have to admit that I'm not familiar with the used kernel based model. However, results shown for day of peak NPP in Fig.6 seem plausible: the decreasing penalty increases the numbers of change points that are found, but the change point with highest penalty is identical with one of the two changes points (there is an overlap of the pink line with one red lines in both models). I became confused when I looked at the results of "Day of MLD <= 40 m". There is no alignment between pink/red lines. Why does the kernel based model miss the largest change point with a decreasing penalty?

*Authors:*

*The largest changepoint using one set of constraints is not necessarily also the largest when another set of constraints are used. This has to do with the fact that changepoints, although defined as local properties, are dependent on an optimal segmentation, a global property. The problem of finding the changepoints is to find the optimal segmentation of a signal given some criterion. A useful criterion codes for example for, the type of change that is looked for and the number of changes. One may for example wish to find the best segmentation into two segments in terms of distance from the mean of some signal. One could then calculate the sum of the deviation from the mean for all possible segmentations by simply looping through the time series, and the smallest one would be the change point. However, if one instead had three changepoints, the set of possible segmentations is different,the optimal segmentation is by necessity also different (as you have more segments) and the largest change point could be, but is not necessarily, different.*

L292: Could you explain why both models have relatively high cross correlations for negative lag ( -3 to -1) years?

*Authors: Our interpretation is that the fact that there is strong correlation on many both positive and negative lags is owing to the changes seen in the two variables having the same drivers (i.e. global warming). Both mixed layer depth and NNP are directly affected by warming, so there is not just a simple NNP=f(MLD) but also at least one lurking variable in temperature. There are, of course, even more variables at play, but the reverse coupling that could exist i.e. NNP affecting light penetration and thus MLD, is not used in the models. We have added some discussion on this on lines 372-375.*

L313:  Here just to summarize that the change in peak NPP day is 1 day for NorESM is not convincing. For a different time period (1960-2010) versus (2010-2060) it was 10. It would be helpful to provide a little more discussion.

*Authors: This problem has been corrected (review comment relating to L259).*

L316 Please share your view on the results of the cross correlation analysis. Does it support the theory of Behrenfeld and Boss?

*Authors: The cross correlation between the day of peak NPP and the first day of MLD above 40 m indicates a high correlation between the two on an annual basis for most provinces. The new results do however also show provinces without notable correlation and even anti-correlation. Even though the correlation of course is not an indication of causality the results are in line with the Disturbance Recovery Hypothesis (Behrenfeld, 2010; Behrenfeld and Boss, 2014) as well as The Critical Depth Hypothesis of when a bloom has the potential to start (see review by Behrenfeld and Boss, 2017).*

*We have added discussion around the correlation and anticorrelation between day of peak NPP and day of MLD shallower than 40 m on lines 360-377.*

Figure captions:

Fig. 1 Give period for the data averaging; is it 2003-2021 as in Fig 2 or 2002-2021 as stated in L135? Did you also masked the model data as in Fig 2? Please indicate in the text if you always use masked ESM data for comparison with CAFE. How large is the difference between masked and unmasked ESM data ?

***Authors: The masking is now applied on both Fig. 2 and Fig 3. The maximum latitude of extant CAFE data is displayed in Fig. S1. The masking affects Nov, Dec, Jan and Feb values.***

Fig. 3 The overlapping colour coding is difficult. Please find a better solution (e.g. instead of full time series show only mean/min/max for each ESM model.)

***Authors: We have removed this figure and added instead a figure (Fig. 4) showing annual mean NPP for each Longhurst province.***

Fig. 5 Please add significance level

***Authors: We have added the 2070-2099 anomaly divided by the standard deviation over the PI-control run in the lower panels of Fig. 5.***

Fig. 6 I assume that the centre of the symbols (circle and triangle) corresponds to the year of the change point. Please add this information. Please use the same names for l1 and l2 through the manuscript and in the figures. E.g. L1 and L2, not "model l1" or "model=l1"

***Authors: This has been changed as suggested.***

Fig. 7 I assume you used the highest penalty level (corresponding to the pink line). Please note. Please also change order of figures – usually EC-Earth is the top figure.

***Authors: We have changed the search method to optimal detection where we predefined the amount of changepoints. The corresponding results using the Pelt search method and a penalty tuned to pick up only one change point is presented in the Supplementary material. The order of the figures has been changed.***

Fig. 8 The figure caption is incomplete ; add …in the median (model l1). Or just write : same as Figure 6, but for first day of ….

***Authors: The caption has been expanded as suggested.***

***References:***

*Behrenfeld, M. J. (2010). Abandoning Sverdrup ' s Critical Depth Hypothesis on phytoplankton blooms. Ecology, 91(4), 977–989.*

*Behrenfeld, M. J., Boss, E. S. (2014). Resurrecting the ecological underpinnings of ocean plankton blooms. Ann Rev Mar Sci.;6:167-94. doi: 10.1146/annurev-marine-052913-021325. Epub 2013 Sep 25. PMID: 24079309.*

*Behrenfeld, M. J., & Boss, E. S. (2018). Student's tutorial on bloom hypotheses in the context of phytoplankton annual cycles. Global Change Biology, 24(1), 55–77. https://doi.org/10.1111/gcb.13858*

**Replies to reviewer 2**

In this work, the authors use daily integrated net primary production data from two CMIP6 models and compare them against an observational dataset derived from satellite. They use change points analysis to highlight changes in the mean NPP and MLD time series. The model and observational data are compared but the comparisons are often subjective, leaving it up to the reader to decide by eye whether the models successfully reproduce the observed behaviour.

However, the authors do not describe the limitations of their work, leaving several key questions unasked or unanswered. In addition, the absence of a discussions section, means that results are often described but not fully interpreted or put into a wider context. This means that the overall conclusions

On the other hand, the language in this paper is excellent. It is very clearly written and there are very few spelling and typographical or grammar errors – at least that I have spotted. This is a very interesting topic and there is a lot that could be done with this high temporal resolution model data. With a bit more effort, this could be a very exciting and impactful paper and I'd encourage the authors to keep going!

*Authors: We thank the reviewer for their thorough and extensive review that has helped us to greatly improve the manuscript.*

**Major points:**
Here are some questions that may encourage the authors to explore new ground. I don't expect all of these questions to be answered, but they may open up some new areas of interest.

Earth System Models s tend to have relatively simple representations of the marine Ecosystem, relative to dedicated ocean models. PISCES has two phytoplankton functional types and NorESM2 has only one. More complicated BGC models exist, such as BFM, Planktom, ERSEM and others, who have four or more phytoplankton classes. Furthermore, it is widely known that blooms are formed from cascades of multiples species. See for instance Kleparski 2021, which uses 24 species assemblages in CPR data to investigate the spatial distribution of north atlantic blooms in observations. How does the low PFT number of these models impact their ability to model blooms?

Kléparski, L, Beaugrand, G, Edwards, M. Plankton biogeography in the North Atlantic Ocean and its adjacent seas: Species assemblages and environmental signatures. Ecol Evol. 2021; 11: 5135– 5149. https://doi.org/10.1002/ece3.7406

*Authors: The community structure of the biogeochemical model has been shown to affect the NPP with the more complex models generating a more non-linear response to climate change (Fu et al., 2016). We have added some discussion around this on lines: 365-370.*

Similarly, marine BGC models are often run as standard-alone Ocean only runs. What do we gain by including the rest of the earth system when investigating phenology? Is there some feedback between the ocean and the atmosphere that improves bloom timing modelling?

*Authors: The physics of the coupled atmosphere ocean system differs from that of the ocean forced at the surface. In the former, the atmosphere and the ocean continuously affect each other and the temperature, currents, stratification are therefore different from the uncoupled system. In particular the air-sea exchanges of momentum, heat, and freshwater are profoundly different in coupled and uncoupled models, affecting especially SST, SSS and mixed layer depths.*

*Furthermore, it has been demonstrated that interactive coupling significantly influences the variability of thermal variables like heat exchange, temperature etc. (e.g. Bhatt et al. 1998, Barsugli and Battisti, 1998). The coupling effect is less constrained for precipitation (salinity does not impact local precipitation while precipitation impacts directly on salinity).*

*These differences affect the biogeochemistry and the seasonal pattern of primary production is therefore not expected to be equal in the coupled vs the uncoupled case. As the coupled runs in this case are concentration driven, atmospheric $pCO_2$ is however not different from the uncoupled version.*

*We have added some discussion around this on lines:372-379*

The main technical criticism is: why only focus on these two models and this one scenario when all of CMIP6 is available? This work either needs a convincing argument as to why it uses this two model limit, or it needs to include other models from CMIP6. Similarly for the SSP5-8.5 scenario. A quick check of ESGF shows that while there is no daily primary production data, there are 380 datasets for daily Chlorophyll (chlos) from 12 different models with picontrol, historical, or Tier 1 ScenarioMIPdata. A tool like ESMValTool could help accumulate and prepare all the data, even if you don't actually use ESMValTool for the analysis. This would allow a comparison of the phenology (for chlorophyll) between several future scenarios and models.

*Authors:*
*We have added some discussion on this on lines:338-348.*

*Although daily surface chlorophyll (chlos) has the advantage that it is easily validated against satellite data and it is a common output from ESMs, it is not in a simple manner related to vertically integrated NPP. The vertical structure of chlorophyll differs between regions and peak NPP may occur below the surface layer (Sathyendranath et al., 1995, Richardson and Bendtsen, 2019). Therefore, peak*

*surface chlorophyll does not necessarily occur at the same time as peak vertically integrated chlorophyll or the NPP.*

*Furthermore, the model runs presented in this work were made within the H2020 project COMFORT and daily output of NPP as well MLD, SST and some biogeochemical variables are available. It is therefore possible to compare the model output of NPP to these variables which would not be possible using the CMIP archive as most of these variables are not saved at daily resolution.*

You've located several change points for bloom timing and MLD. What are the differences (if any) between the period before this point and after this point? For instance, you could revisit fig 1, 2 or 4 comparing the "pre- change" to the post-change for these models.

*Authors: the kernel method for change points used here is not simply picking up changes in a given moment of a distribution, like mean, variance or skewness. It can detect any type of change. This has obvious benefits in that the palette of changes that can be detected is much larger. However, it also makes interpretation more difficult. This ambiguity is the reasoning behind using the L1 and L2 models that focus on specific moments. Most often, the L1 and L2 models find the same change points as the kernel model, suggesting that the changes found are common to many statistical moments (line 356).*

*However, the information that we are most interested in is really more tied to physical and biogeochemical reasoning than to statistical moments. From this perspective, it is instructive to use a change point analysis mostly as a detection algorithm, but to leave the interpretation of the nature of the change to physical and biogeochemical reasoning. This is the essence of why we combine e.g. MLD analysis with the phenology, it also means that only some change points will be meaningful to us in this sense. Of course, also in the statistical sense the meaningfulness of a change point is down to arbitrary measures, like the size of a chosen penalty or one's favorite level of confidence.*

On L41, you mention that NPP is affected by precipitation, wind patterns, temperature and light, but do not investigate any of these fields. You only investigate MLD as a proxy for reduced nutrient availability. I'm not fully convinced that the argument has been made that MLD shallowing is the cause of the bloom in these models. Daily data for several of these variables should be available in CMIP6 as well.

*Authors: We have looked at daily SST and MLD both absolute values and phenological indicators and their relation to NPP.  No clear cut correlations were found between SST and NPP or their phenology. Note also that MLD is directly dependent upon all the mentioned variables, so these different drivers should not be considered independent. We have added discussion about how different drivers affect NPP in the revised version (From line 362).*

The paper is also missing a discussions section. This could include things like:

- How well this work fits into the current state of research for this field.
- What mechanisms makes one model better than the others? Is it purely their physical behaviour or is there something about the parameterisation of the marine BGC model?
- Can you estimate how many years of data would be needed to detect these change points in CAFE?
- Could you use longer data sets (ie Continuous Plankton Recorder data) to detect change points in these regions?
- A limitations of this method section

A more thorough investigation of phenology may also be interesting, not just the bloom initiation or maxima, but the intensity and duration of blooms may also be changing in the future. See for instance:
https://www.sciencedirect.com/science/article/abs/pii/S1470160X11002160

***Authors:  The discussion section was  put together with the results section in Section 3, Results and discussion.  In the new manuscript we separated them and added more discussion.***

***About the CAFE data and the length needed to detect these change points, it is important to realize that all change points that are down to natural variability will be different in CAFE and the two ESMs. So we should not expect, generally speaking, to find change points from an ESM in CAFE unless we can tie them to changes in a common forcing like greenhouse gases or aerosols. This is true regardless of the length of the CAFE time series.***

***However, it is also true that generally speaking that if a single time series, T, of length n has only two change points and both are in the latter half of the series (i.e. in T(n/2:n)), then the time series T(n/2:n) still could have two different change points. That is, running T(n/2:n) through the same change point algorithm as T does not necessarily give the same change points, because the optimal segmentation is a global property. In other words, the optimal segmentation of T and T(n/2:n) is not necessarily the same. Therefore, even for perfect replica time series, the time series has to have the same length to ensure that they have the same change points.***

In addition, other sources of observational data are available, and would strengthen the arguments around the presence of change points:

Floats: https://www.frontiersin.org/articles/10.3389/fmars.2020.00139/full

CPR (in North Sea): https://aslopubs.onlinelibrary.wiley.com/doi/10.1002/lno.11351

Bermuda time series: https://bats.bios.edu/

***Authors: These are great references. However, comparing vertically integrated NPP to observations is difficult. Most sources use phytoplankton biomass or chlorophyll as***

*in the suggested references. This is also the reason why we choose to use the satellite based CAFE model for the comparison.*

*Note also that the presence of change points in the models is independent of the models ability to reproduce observations. What comparisons to observations can give us is some grasp of the model's skill in terms of having a reasonable climatology. Although, even that is hard given that we compare historical simulations to our recorded history (i.e. one realization of all possible histories) and also given that the "observations" used are, in fact, also a model. However, the silver lining is that understanding and describing the behavior of ESMs is interesting and important even in itself given their extensive use in science and their impact on policies. We have added a comment on this (line 215).*

**Specific comments:**

**Title:** To get more impact, state your main result in the title. Peak Net Primary Production will shift earlier in the year over the 21 st century. (or something like that).

*Authors: The title is changed so that our main result is highlighted: Phenological shifts in the North Atlantic net primary production detected in the 21st century. Results from two Earth system models.*

**Abstract:**
L11: Similarly to the Title – if you open the abstract with the main result, it can be more eye catching. Try not to hide your exciting result at the bottom of the abstract!

*Authors: We have rewritten the abstract in order to showcase the main result higher up.*

L16: "The low spatial resolution of the earth system models can explain much of such difference": Can it? How so? I'm not convinced that this argument has been made.

*Authors: We have removed this statement.*

L20: Using SSP585 as a forecast needs to be treated carefully, as this is not a "realistic" scenario. SSP585 is the scenario with enhanced fossil fuel usage – meaning that the rate of fossil fuel emissions accelerates beyond "business as usual" - something we have fortunately not seen in the previous 8 years since the end of the CMIP6 historical period.

*Authors: It is true that SSP5-8.5, in that sense, is not the most plausible scenario. However, this also gives us kind of an upper end estimate on the size of change. Furthermore, many of the largest change points in the timeseries occur in both models at the end of the historical simulation (Line 351).*

**Introduction:**
L27: See Le Quéré et al for more up to date reference on the carbon budget.

Le Quéré, C., et al : Global Carbon Budget 2018, Earth Syst. Sci. Data, 10, 2141–2194, https://doi.org/10.5194/essd-10-2141-2018, 2018.

*Authors: This reference does not give explicit numbers for NPP.*

L35 and elsewhere: CO2: 2 should be subscript

*Authors: This has been corrected.*

L41: Can you add some references for this? How is NPP impacted by precipitation in the North Atlantic?

*Authors: Increased precipitation and changed wind patterns affect the water column stratification. In this way, NPP is indirectly affected by these variables. Furthermore, precipitation can have a fertilizing effect by transporting nutrients to the surface waters (Myriokefalitakis et al., 2020). In particular, MLD is affected by precipitation. Freshwater added to the top of the ocean increases vertical stability through its effect on density. Fresher water needs to be cooled more to get the same density as saltier waters.*

*We have added some references on line:43.*

L55: "Depending on the onset…" This sentence needs to be more explicit. Ie, If thermal stratification occurs, then spring bloom may start earlier…" or similar.

*Authors: The text has been changed starting from line: 54.*

L57: One alternative theory about the causes of bloom timing changes is the switch from positive net heat flux to negative net heat flux:
Smyth TJ, Allen I, Atkinson A, Bruun JT, Harmer RA, Pingree RD, et al. (2014) Ocean Net Heat Flux Influences Seasonal to Interannual Patterns of Plankton Abundance. PLoS ONE 9(6):e98709. https://doi.org/10.1371/journal.pone.009870

*Authors: We have added this reference (Line 60)*

L72: What makes it unique? Why only these two models? Why not a full CMIP6 ensemble? What do you gain by using the years 1750-1850?

*Authors: The Pi-Control is a very important addition that gives us a good gauge of the range of the natural variability, something that the forced changes can be compared against, see Fig. 5 bottom panel (new addition). Furthermore, the benefit of using long time series is a more robust estimation in the tails of the probability distribution. Consequently, rare events such as extreme conditions are better reflected in the data set. The data used in this work was produced in the H2020 project COMFORT and the two models are the only ones that have saved daily vertically integrated primary production for the full 1750-2100 period.*

**Methods:**

L76: Just to confirm, are you are using the CMIP6 dataset, intpp, from ESGF? When you calculate the mean, are you taking the cell area weighted mean?

*Authors:*
*The EC-Earth-CC and NorESM2-LM historical and SSP5-8.5 used in this study were performed according to CMIP6 protocols (with additional daily output activated) but have not been published on ESGF. A minor update was made to the terrestrial vegetation model in EC-Earth-CC after the PiC was performed but is not something that is expected to affect the results.*
*The mean is the cell area weighted mean.*

L87: Nemo should be NEMO
*Authors: This has been corrected.*

L89: pCO2: 2 should be subscript
*Authors: This has been corrected.*

L96: Primary production is indeed growth of phytoplankton, but elsewhere you talk about net primary production. Net PP usually does include some loss terms – otherwise you mean Gross Primary Production (GPP).
*Authors: We have changed to "net primary production".*

134: Modis should be MODIS
*Authors: This has been corrected.*

136: The Change point analysis method description section (sect. 2.3) does not sufficiently explain how the method works, and gives the impression that the authors have used the Ruptures packageas a "black box". Can you give more detail on how this method works here (or perhaps in an appendix)? Same for L1 & L2 methods.
*Authors: We have expanded this section.*

**Results:**

L159 and figure 1: Is it sensible to compare depth-integrated Net Primary production to satellite (surface) Net Primary Production? Are these data comparable?
*Authors: CAFE is a model that utilizes ocean color and other optical properties to estimate the vertically integrated NPP (Silsbe et al., 2016). These values are not just for the surface but for the entire water column in similarity with the ESM results. The vertically integrated properties is a strong argument for the method used in this manuscript. We clarified that it is total water column NPP on line 153.*

L169: Some statistical tools would help give an objective estimation of model data fit, something like some pattern statistics or even a linear regression?

*Authors:* **Given the short time series and the fact that these series are from different histories, we would argue that applying more statistical methods would not answer relevant questions. In essence, what we have here is not a well constrained model**

data fit problem. Any method of comparison is only as good as the data admits, and this data admits very little in terms of direct comparisons. Put another way; say we calculate a spatial correlation between the CAFE data and our two models and find that r(EC-Earth,CAFE)<r(NORESM,CAFE), or that a trend in some area in EC-Earth is more similar to that in CAFE than to that in NORESM, what, of value, would that tell us? The answer unfortunately is nothing of great value, as all such metrics would be strongly influenced by natural variability during the short CAFE period. This type of variability is not in temporal sync in these different models.  A comment on this has been added on line 213-215.

L176 – If the mask is important, you may need to indicate it in this figure (or elsewhere). *Authors: We have added the seasonal cycle of the maximum latitude of the Cafe data that has been used to mask the model data to the supplementary material (Fig. S1).*

L180: "Reasonable". Once again, an objective measure of goodness of fit may be useful here.

*Authors: We have removed this statement. We would, however, argue that it is easy to tell by eye that these distributions are quite different. Of course we can calculate the p-value of a two-sample Kolmogorov-Smirnov test and include that, but really it is evident that these data are drawn from different distributions. Moreover, given that they are sampling different histories it is not known what a good fit would be.*

L186 – is a version of the model with higher resolution and better agreement with observations exists, then why not use data from that one?

*Authors: The model results were more similar to what is seen in EC-Earth-CC but not necessarily better. EC-Earth-CC displays a too early peak NPP and a too strong decline after this peak. Model development of NorESM targeted this behavior resulting in a better timing of the peak. We have removed this statement in the new version that includes separate biogeochemical provinces.*

Fig1: The MAM bloom in EC-Earth-CC seems unrealistically high, but I'm not convinced that either model is able to reproduce the observed behaviour over this time period. A statistical comparison would allow you to state how well these models reproduce observed behaviour. We can't expect ESMs to reproduce specific observations perfectly, but broad scale decadal means should be feasible.

*Authors: By construction, neither of these models can replicate observed behavior over specific time periods as the internal variability of the two ESMs is not in sync with the observed one. An ocean only model forced with reanalysed atmospheric forcing could conceivably replicate observed behavior over specific time periods, coupled climate models on the other hand can be in completely different phases on NAO, ENSO, AMO and so on, owing to the chaotic nature of unforced climate variability. See comment on lines 213-215.*

Fig2. The coloured lined are the multi-year mean, but what does the lightly shaded area represent?
*Authors: (now Fig. 3) The shaded area has been removed as the figure that now includes eight different biogeochemical provinces would become too messy.*

**Section 3.2:**
L193 – Try to avoid single sentence paragraphs like this.
*Authors: This has been changed.*

L195: CAFE (uppercase)
*Authors: This has been corrected.*

Fig 3: Are you actually showing 8 day means on a figure that spans over three centuries? It looks like the shaded areas are the 8 day mean's annual minimum and maximum. Naively, from figure 2, I would expect the minimum value of NorESM2-LM to be around 50 mgC/m2/day, but it appears to be lower than that? Similarly, the range in EC-Earth-CC has a minimum around 200 in figure 2, but it looks closer to 150 in figure 3. I'm not convince that showing the range is useful here, and it's not clear to be me what it represents. Perhaps it may be easier to show the 5-95 percentile ranges (once again – weighted by area) instead, to avoid erroneously high or low values?
*Authors: (Now Fig. 4) We have removed the shading as the figure that now contains eight different biogeochemical provinces would get too messy. The lines show the annual means from the different models.*

Figure 4 hides a lot of the important information, only showing a little of the earlier years underneath the later years. Perhaps you could instead show some decadal averages (or variouschange point regimes) as semitransparent bands?
*Authors: We have removed this figure.*

208: Is the peak NPP the best metric for this? In the past, I've seen bloom timing calculated using the maximum in the first derivative, ie,when phytoplankton is growing the fastest, or when the chlorophyll concentration rises above the long term median: See for instance: Philippart, C.J.M., van Iperen, J.M., Cadée, G.C. et al. Long-term Field Observations on Seasonality in Chlorophyll-a Concentrations in a Shallow Coastal Marine Ecosystem, the Wadden Sea. Estuaries and Coasts 33, 286–294 (2010). https://doi.org/10.1007/s12237-009-9236-y ,Marie-Fanny Racault, Corinne Le Quéré, Erik Buitenhuis, Shubha Sathyendranath, TrevorPlatt, Phytoplankton phenology in the global ocean, Ecological Indicators, Volume 14, Issue 1, 2012, Pages 152-163, ISSN 1470-160X, https://doi.org/10.1016/j.ecolind.2011.07.010.
*Authors: We claim no optimality for this metric. There are different phenological indicators that could be used. The timing of the peak is a well defined property that has been used before (eg. Nissen and Vogt, 2021; Henson et al., 2013) while several different methodologies, giving different results, exist for the timing of the start (Thomalla et al., 2015). We have tried some other phenological indicators but found the day of peak NPP to be a more robust metric for this data set. Max is generally*

*robust unless the distribution is bimodal, and two peaks have similar magnitude, which we did not see. First derivatives are typically spiky, concentration based cut-offs are a bit arbitrary, and different cut-offs would likely be needed for the two models.*

*We have added some text on this on line 96.*

Figure 5: The pane labelled 1850 is the mean of 1850-1879. Perhaps these labels should be 1850-1879 mean, 1970-1999 anomaly, and 2070-2099 anomaly. I also think that a discrete colour scale would be useful here. I'd also like to see the CAFÉ data peak NPP day.
*Authors: We have changed the labels. We have added the mean day of peak NPP for the different provinces and for both CAFE and the ESMs to Table 1.*

L241: I'm not convinced how representative the mean over the whole region is here, especially after seeing how heterogenous the phenology behaviour is in fig 5! Perhaps it would be better to define sub regions within the domain and see how they behave (ie Labrador Sea, Gulf Stream, Southern NA, Central NA, Northern NA... etc.)?
*Authors: We have redone the analysis for Longhurst provinces.*

L254: The fact that several methods agree that 2010 is a change point for EC-Earth-CC should give you confidence that this is a real change. However, there isn't the same agreement in NorESM2-LM. How would you interpret this? Are the changes perhaps more real in EC-Earth-CC than in NorESM2-LM? Can this shift at 2010 be seen in any observations? How do the phytoplankton bloom and the physical drivers of the bloom differ in EC-Earth-CC before and after 2010? The second EC-Earth-CC change point is around the year 2090 – how many years after the change are required to register the change?

*Authors: The kernel based model finds all changes in the probability distribution. This means that the exact nature of the change is not clear. We therefore complemented the analysis with the L1 and L2 method that finds changes in median and mean respectively. That the change points are not at the same location just means that the changes in mean, median and higher order changes do not occur at the same point in time. We think it is quite important to not overinterpret change points. In the sense that the statistics of the time series changes in some appreciable manner, all change points are real. In the sense that there is necessary a co-occurring change in the physical drivers, perhaps no change point is real. The observational time series is not long enough for it to make sense to look for change points. Moreover, the year 2010 in EC-Earth3-CC is not the same as 2010 in NorESM2-LM or in reality. What they have in common is similar levels of greenhouse gasses and aerosols in the atmosphere. Climate variability is not in phase in the two (or any other CMIP6) models nor is either model in phase with reality. Therefore, it is unlikely that 2010 has that significance. Moreover, we note now with the regional analysis in place that a 2010 change point is not a common feature between models and regions.*

Figure 6: What's the value of using 8 change points? I can't see them mentioned anywhere in the text, I recommend removing them from this figure if they aren't discussed.

***Authors: We have removed those.***

Figure 6: What does it look like if you apply this same method to CAFE? There isn't as much data but there is still a couple decades. Is that enough to detect changes?
***Authors: We can for sure get change points in the time series as we are able to obtain change points all over the 350 yr model time series for a low penalty. However, as the time series is very much shorter and the change points depend on the global segmentation the results would not be comparable.***

L271 and figure 7: How do you interpret change points in the pre-industrial period? If they can occur using this method, then it's hard to justify that later change points are linked to climate change without additional analysis.
***Authors: It is generally speaking true that the change point analysis does not say anything about the cause of the change. Therefore, regardless of when a change point occurs you always need some additional arguments to tie it to e.g. climate change. Nevertheless, our main result is that the largest changes over this 350 yr timeseries occur in the late historical simulation or in the future scenario for the vast majority of this region as seen in Fig. 7. This is true in both models and their natural variability is not in sync, climate change is thus a very plausible candidate as the cause of these changes. Moreover, the bottom panel in Fig 5 now shows the change normalized by the yearly standard deviation from the PI-control experiments. From this it is very clear that these changes did not occur by chance.***

Figure 7: I really like this figure, but I think it could be more effective. Perhaps you could focus in on the recent past and the future regions by limiting the colour scale to (ie) the years 2000-2100? Do we really expect change points to occur earlier than say 1950? If white means no change single point could be found, the land colour needs to be a different colour – light grey perhaps. (You also have some ocean points occurring over the land mask, so make sure you set land zorder to be higher than the pcolormesh here and elsewhere. Similarly, the contour lines are the same style, thickness and colour as the same surface, and this is confusing.
***Authors: We have edited the figure. The color map is now exchanged for a diverging centering around the year 2000. We find that this gives a clearer view of the conclusion that the largest changepoint for most of the grid points occurs after the year 2000. We changed the land color. Furthermore, we changed the search method so that there are no nans in the data. A corresponding map containing the old search method is, however, included in the supplementary material (Fig. S6.). In this figure, previously white nans have been colored blue.***

Fig 7 caption: rephrase "White spaces are areas where a single change point could not be found."
***Authors: We have replaced this figure with a corresponding one using a different search method. The white spaces are not present in the new figure. A figure using Pelt search method has been added to the supplementary material (Fig. S6).***

L278: I find the Smythe theory about Net heat Flux (linked above) particularly compelling – even if it may not be applicable to the open ocean. Could heat (or temperature as there is a lot of CMIP6 SST data!) play a role in bloom timing?

*Authors: We have compared to SST but did not find any convincing correlations. We have added a discussion around the connection to other variables on L339. We have also added the Smyth et al reference (line 60). However, as we did not have daily output of net heat flux, this was not possible for us to examine.*

L279: Please be cautious with using "more and less" to describe depths. As closer to the sea floor means larger values of depth, "40m or more" can mean: "deeper than 40m" or "shallower than 40m"!

*Authors: We have changed this.*

L287: You have tested for the first day below 40m but comparing figures 6 and 8 makes it look like this threshold generally occurs after the peak NPP in EC-Earth-CC. The MLD lines average around day140, while the peak NPP seems to be around 130-135. How can MLD shallowing occur after the bloom peak if it is the main cause of the bloom initialization?

*Authors: In reality MLD shallowing in the model is gradual, but the choice of a 40 m threshold gives it a definitive date. In other words, we suggest that blooms may start when the MLD becomes sufficiently thin, not that 40 m is the magic number. The chosen number, 40 m, is just a proxy. Several other numbers have been tested with similar results (Supplementary material Figs. S3-S5).*

Fig 8: This figure could be merged with figure 6, and it would drive the MLD discussion earlier in the results section.

*Authors: We found it difficult to merge the figures as they both now contain data from all eight biogeochemical provinces. We have, however, switched the figures so that the MLD figure (now Fig. 7) comes right after the day of peak NPP (Fig. 6).*

**Conclusions:**

L300: Unresolved Eddies? How do you know this? I'm not convinced that you've demonstrated this conclusion.

The line is no longer in the manuscript.

L313: "1/11 day/s in NorESM2-LM/EC-Earth3-CC" should be: "1 day in NorESM2-LM and 11 days in EC-Earth3-CC"

*Authors: This has been changed. We have also included a table (Tab 2.) that summarizes the change in the different provinces.*

L314: 31/33 should be "31 and 33"

*Authors: This has been changed.*

L320: First mention of fish! Maybe put something in the introduction or the discussion sections.

*Authors: Fish is also mentioned in the introduction on lines and 63.*

L322: First mention of ecosystem structure! Maybe put something in the introduction or the

discussion sections.

*Authors: The effect of model community structure on the NPP is mentioned from line 380.*

**Code Availabililty:**
L331: What about the Ruptures python package?
*Authors: A link to Ruptures has been included.*

L340: This link did not work for me, nor could I find it using the zenodo search bar.

*Authors: We have rewritten.*

*References:*

*Bhatt, U. S., M. A. Alexander, D. S. Battisti, D. D. Houghton, and L. M. Keller (1998). Atmosphere–Ocean Interaction in the North Atlantic: Near-Surface Climate Variability. J. Climate, 11, 1615–1632,*
*https://doi.org/10.1175/1520-0442(1998)011<1615:AOIITN>2.0.CO;2*

*Barsugli, J. J., and D. S. Battisti (1998). The Basic Effects of Atmosphere–Ocean Thermal Coupling on Midlatitude Variability. J. Atmos. Sci., 55, 477–493,*
*https://doi.org/10.1175/1520-0469(1998)055<0477:TBEOAO>2.0.CO;2.*

*Fu, W., Randerson, J. T., & Keith Moore, J. (2016). Climate change impacts on net primary production (NPP) and export production (EP) regulated by increasing stratification and phytoplankton community structure in the CMIP5 models. Biogeosciences, 13(18), 5151–5170. https://doi.org/10.5194/bg-13-5151-2016*

*Geider, R. J., MacIntyre, H. L., and Kana, T. M. (1997). A dynamic model of phytoplankton growth and acclimation: responses of the balanced growth and Chlorophyll a : carbon ratio to light, nutrient limitation and temperature, Mar. Ecol.-Prog. Ser., 148, 187–200.*

*Myriokefalitakis, S., Gröger, M., Hieronymus, J., and Döscher, R. (2020): An explicit estimate of the atmospheric nutrient impact on global oceanic productivity, Ocean Sci., https://doi.org/10.5194/os-16-1183-2020*

*Nissen, C. and Vogt, M. (2021). Factors controlling the competition between Phaeocystis and diatoms in the Southern Ocean and implications for carbon export fluxes, Biogeosciences, 18, 251–283, https://doi.org/10.5194/bg-18-251-2021.*

*Richardson K, Bendtsen J (2019). Vertical distribution of phytoplankton and primary production in relation to nutricline depth in the open ocean. Mar Ecol Prog Ser 620:33-46. https://doi.org/10.3354/meps12960*

*Thomalla et al. (2015). High-resolution view of the spring bloom initiation and net community production in the Subantarctic Southern Ocean using glider data, ICES*

*Journal of Marine Science, Volume 72, Issue 6, Pages 1999–2020,*
*https://doi.org/10.1093/icesjms/fsv105*

---

## Referee Report (RR1)

Review to

**Phenological shifts in the North Atlantic net primary production detected in the 21st century. Results from two Earth system models.**

Jenny Hieronymus et al

General comments:

The authors did a great job in revising the analysis. By dividing the North Atlantic into Longhurst provinces, the results are much more consistent and meaningful. Interestingly, the most of the provinces show an increase of NPP until the end of the century except two regions in both models (disregarding the small changes in SARC and NADR in NorESM2-LM). This result was masked in the previous analysis where a declining trend in NPP was postulated for the entire domain over SSP8.5. This finding has to be included in the Summary and Conclusion section which still mentions an overall NPP decrease (L408). In addition, I do feel that many of the final findings are related to results from the EC-Earth, e.g. "phenological shifts occurring in the early 21st century " is not true for NorESM (in 6 out of 8 regions the changepoint is after 2048). Please critically review the entire manuscript to see if the final statements apply to both ESMs.

A general remark on the quality of the figures:

- the increment of contour lines should be specified for the subplots in the caption; e.g. in Fig 3 each of the SON panels has different increment

- contour lines in Fig.8 are horrible – delete or omit the entire Fig. (see specific comments)

- I plea for a,b,c notation in the figures for more readability

In general, I recommend the publication of the manuscript after my specific comments have been addressed.

Specific comments:

L25: Please correct: Net Primary Production (NPP) is the rate of photosynthetic carbon fixation minus cellular respiration

L82-83: "We divide the region into biogeochemical provinces (Longhurst et al., 1995) in order to see how localities with similar biogeochemical functioning differ across the region." This sentence is confusing. What do you mean by "localities"? Do your provinces really have a similar biogeochemical functioning? Delete "Furthermore".

L85-88: Please motivate here the purpose of MLD analysis and reorder the sentences - first: change point analysis for MLD as for peak NPP; second: all about cross-correlation and what we learn from it.

L94: typo: in section 2.4 is the change point analysis

L96: "maximum" instead of "max"

L97: "found in your data" ESM data or CAFE or all data sets?

L117: replace "external concentration in nutrients" by "nutrient concentrations of the ambient water"

L118: Please give the same information for both BGC modules. i.e. delete :"PISCES is suited for a wide range of spatial and temporal scales, including quasi-steady state simulations on the global scale."

And add for iHAMOCC, that iHAMOCC also simulates the carbon system, as well as dissolved and particulate organic matter"

L119-120: "Net primary production is the growth of phytoplankton thus the term excludes mortality, excretion and grazing." Why is this mentioned here? By definition, NPP excludes mortality, excretion and grazing. Don't mix it up with NCP = net community production. Delete sentence?

L163: Rephrase your sentence to e.g. : "The seasonality of NPP depends, among other things, on local physical conditions of the ocean" ?

L168: Longhurst defined the static boundaries – "made" is a strange word?

L171: You never use "coastal, westerlies and polar" – delete; The North Atlantic domain is divided in the provinces shown in Fig. 1.

L176 delete: The west wind regions;

L209ff: I recommend to show and discuss only MAM and JJA and omit SON. It shows a more or less a uniform pattern for the entire domain and complicates the data processing due the lack of data in CAFE in winter. SON gives no additional information. In addition, please find a better color scale. It is surprising, that your scale ends at 1000 but Fig.3 shows numbers higher than 1200. Please correct.

L226: Instead of using daily ESM data, use a 8-day running mean for the comparison to 8-day mean data from CAFE. Results in Fig.3 are difficult to compare. Please reorder the seasonal cycles by region instead of data sets: e.g. BPLR+ARCT for CAFE and both models, and so on. Adjust axes to maximum values. Make sure that all lines have the same starting point if you mask the ESM data with available CAFE data.

L264: make sure, that you don't use the word "region" for both, the entire North Atlantic and the provinces; use e.g. the words "domain" and "provinces" throughout the manuscript.

L265ff: could you improve the readability by shorten the name of the 3 periods: e.g. 1865s = 1850-1879, 2000s= 1985-2014, 2085s= 2070-2999? Then you can omit to write "period" or "early/late period"

Fig 5: Please use a standard statistical test (e.g. student's t-test) to determine the significance. With the given information, it is difficult/impossible to interpret the results. Please show results of EC-Earth on the left side as usual.

L277 " size of NPP" – delete "size of"

L284: you don't average over different provinces, rephrase.

L285: Fig. 6 shows …. together with the largest (…. sentence incomplete

L304: The posed question was reasonable for the previous analysis, but I cannot see the benefit when using Longhurst provinces. Isolines in Fig. 8 should be removed, if not the whole figure is omitted or transferred to the supplement. In the supplement you could also add the discussion on the difference between the PELT method and Fig.8 and why one has blanks and the other not.

L316: province averaged instead of area-averaged? Or just write: "between the time series in Fig 6 and 7" because it is clear how they were archived.
L320: Typo? NADW is not defined
L323: "Looking at Fig. 8 ….. you mean Fig 9 ?
L329: you never show the "size of peak NPP". delete "size of peak" or explain what you mean
L351: use "finding" instead of "observation"

L352: replace "then that…" with "when the warming is the strongest in the SSP5-8.5"

L392: replace "realistic physics" with "consistent physics"

L400-401: rephrase: you don't use Longhurst provinces to look at spatial averages, but to account for the different areal conditions

L408: As already mentioned above, the NPP increases for many provinces. Revise!

---

## Referee Report (RR2)

**Phenological shifts in the North Atlantic net primary production detected in the 21st century. Results from two Earth system models.**

Jenny Hieronymus, Magnus Hieronymus, Matthias Gröger, Jörg Schwinger, Raffaele Bernadello, Etienne Tourigny, Valentina Sicardi, Itzel Ruvalcaba Baroni, and Klaus Wyser

Submitted to Biogeoscience, bg-2023-54

Review by Dr. Lee de Mora, Plymouth Marine Laboratory, Plymouth, Devon.

**Summary**

In this work, the daily Net Primary production of two Earth System Models in the Northern Atlantic are described, compared against satellite data and analysed using change point and cross corelation analysis for several regions of the North Atlantic. The timing of the peak of the bloom will shift earlier in the year in the Northern parts of the North Atlantic. The models disagree for the Southern North Atlantic, but it is less of a shift than in the northern regions. The change point analysis highlights that several regions are likely to have pass the change already and that nearly all regions will cross the change point in the 21st century. However, it's not clear how significant the scale of the change point will be.

The text is well written, the underlying science is well introduced in a clear way, the results are presented and described accurately, I did not spot any spelling mistakes and the grammar is almost always fine. At times, the style is a little colloquial and would be improved if certain parts were written in a more formal style. There's a few run-on sentences which need to be pruned. There were many formatting issues, described below, and there are likely many more that I missed.

The figures are generally clear, but I suggest a few improvements below which I think would benefit the paper as a whole.

I found that there were a few discussion points that were hinted at in the abstract and introductions, but never made it to the final draft. I have a few questions below, a few suggestions and a few possible additions.

In my opinion, the biggest weakness of the paper as it currently stands is that the key results are not sufficiently well articulated. It's crucial that the revision of this paper focuses on its unique results and explains why they are important as clearly as possible.

I would also recommend a careful and thorough readthrough (including the references section) before re-submission. Sometimes it's better to share this task with a more distant co-author, as the lead author is often too close to the work to spot these issues.

As the list of changes below has become rather long, I recommend major corrections before reconsideration. However, I don't want to come across as being harsh. This is a good, well written paper, with good scientific content, it fits within the remit of the journal, and most of these changes should be resolvable with minimal effort.

**Specific Comments**

I expect a more forceful and direct tone in the title, abstract and the conclusions. At the moment, the abstract focuses on the methods, but it should effectively read: "This is our main result. This is why it

is important." Then once you've said that, only then you can describe the methods and models that were used to found out.

Similarly, the title could be a more direct and effective. Something like: "Net Primary Production Annual Maxima in the North Atlantic projected to shift in the 21$^{st}$ century" or something like that. (As an aside, I don't think you need to mention ESMs in the title – it's obvious that models were used if you're making projections of the future!)

I'm not convinced that either model captures the observations over the historical period. The model-data comparisons in figure 1 and table 1 are subjective. I'd like to see a robust statistical comparison of the model and data over the historical period. A graphical version like a pair of Taylor or Target diagrams would be a solid improvement. Alternatively but probably less effective, you could add pattern statistics (bias, deviation, and correlation) to Table 1. Please bear in mind that you have made a new and unique piece of work, and people in the future will be glad to have a robust statistical benchmark to cite that they can compare their model quality against.

It would be valuable to include the analysis of the mean of the whole North Atlantic region in some of these figures (ie Figure 3, Tables 1, 2 and 3 as well?) I understand the value of the individual regions, but a clear result for the whole region would be a good headline result.

I would be interested in seeing how different the phenologies of the various regions are on either side of the change point. Basically, a version of figure 3 which compares the climatological mean of ten (or some useful number of) years each side of the change point. This would be a clear and effective way of showing that there is indeed a real change between those two periods.

There is no daily net primary production data from CMIP6 on ESGF, but there is quite a lot of surface chlorophyll (chlos) and surface phytoplankton carbon (phycos). It would be interesting to place these two models against the rest of CMIP6 in the context of the phytoplankton carbon or chlorophyll. Are they typical or are they outliers? This would be a lot of additional work, so I leave it up to the authors to decide whether they can perform the additional analysis. If not, then maybe add it as a suggested extension in the discussion.

As I mentioned, several modelling centres have contributed daily surface chlorophyll and phytoplankton carbon to CMIP6, but no one has contributed daily NPP. Do you not want to make the case to include daily NPP (intpp) as a standard variable in future CMIP experiments? What do we gain from including NPP that we don't get from chlorophyll and carbon?

At the end of the discussion section, I found myself asking several follow up questions like these. I have listed these below with the label "L394".

**Typesetting & style comments**

There is a tendency for sentences to be too long and complex, which makes them harder to read and parse - particularly in the abstract. While they may be accurate, they take more effort to understand. For this reason, I personally have a strong preference for simpler shorter sentences. I have pointed out a couple in the abstract and made some suggestions on ways to shorten and split them. However, I'll leave it up to you from that point.

Please try to be consistent with hyphenation and capitalisation. Change-point, time-series, cross-correlation, North-west, PI-control are all written in several different ways throughout the text.

There are several places where the superscript is lost for both the degree symbol, units and centuries (especially in the title!). Please be more careful with subscript and superscripts.

There are a few places where the text is stretched: L157, L543, L691.

For references in the body of the text, there are a few places where the name in the text either doesn't match the reference, or a reference does not exist. Similarly, there is some variability in typography of the "et al.", sometimes the period is missing (L336) and sometimes there's no space before the year (L140).

In the reference section, there are a lot of inconsistencies:

- Several references with strange characters that need to be corrected. Ie L471, 474.
- Some DOI's are links in blue and some are not, ieL 462 vs L464.
- Most references have the author initials, but L475 includes author's first names.
- Some are missing DOI's ie L480.
- Some place the year at the start (L461) and some at the end (L466).
- Some times the journal name is in italics (L468) and sometimes it is not (L464).
- Some references include et al. (L494 & L574) but most do not. Some instead use "…" (L501, L556, L562 & L594)

To me, this suggests that the author is manually writing the bibliography – which just makes your life more difficult! If you are, I strongly recommend instead using some kind of reference management tool to keep track of the bibliography and ensure that references are done properly and consistently. (I use bibtex for latex).

Without being an expert on the North Atlantic, I find the 8 different region names to be confusing. I'm constantly having to refer to figure 1 to check what region is being discussed. It may be clearer to write where things are happening more descriptively than just relying on the four letter acronyms. For instance, in line 253: "The largest standard deviation is found in NASE and the lowest in NWCS" could be clearer as "The largest standard deviation is found in the southeast of our domain in NASE and the lowest is in the Northwest Atlantic Shelf (NWCS)." This is closer to how you have described figure 5 in lines 264-274, which is much clearer.

In the figures, I think there's scope for some additional consistency which would make it easier to interpret. For instance, you can use the same regional colours from figure 1 again in figure 3. Instead of the blue/orange colour scheme in figures 4, 6 and 7, you could use the regional colours again, but have a lighter one for EC-Earth and a darker one for NorESM2 (for instance). Alternatively, you could keep the same two line colours, but change the regional labels to match figure 1. Similarly, I see no reason for figure 9 to be different to figures 6 and 7. Finally, and this is purely subjective, but I'm not crazy about the blue and orange colours – I think a more aesthetic colour pairing could be found (https://colorbrewer2.org/ is a good resource for something like this).

As a added suggestion, daily data looks great in video format – much better than monthly and annual data. Have you considered including a supplementary video of the daily climatological mean for the two models and the observations? This would be a great resource to show when presenting this work in person. Alternatively, it could contribute to a great video abstract.

**Specific Points**

**Abstract:**

L14: This is a long sentence which could be clearer: "The majority of the region displays the largest change point in the day of peak NPP occurring after the year 2000 indicating a shift towards earlier peak NPP with the most change occurring in the northern parts of the domain."

Suggested change: "Most of the region has the largest change point in the day of peak NPP after the year 2000. This indicates a shift towards earlier peak NPP and the most change occurs in the northern parts of the domain."

L18: long sentence: "Furthermore, the occurrence of the first day with MLD shallower than 40 m shows positive correlation with the occurrence of the day of peak NPP for most of the domain and, similar to the day of peak NPP, displays the largest changepoints occurring around or after the year 2000."

Suggested change: "Furthermore, the occurrences of the first day with a MLD shallower than 40 m and the day of peak NPP are positively correlated over most of the domain. As was the case for day of peak NPP, the largest changepoints occur around or after the year 2000."

L20 and elsewhere: Is it **changepoint** or **change point** or **change-point**? Beaulieu (2012) uses change-point so maybe that is the best option?

**Introduction:**

L26: Turnover time is not defined and never referenced again. Why is it important?

L27: "Almost equals"? What is the land NPP value?

L27 – and elsewhere: I prefer Pg instead of Gt. Ton is not an SI unit and there are many definitions of tons/tonnes/imperial tons and it can be confusing for international readers. Also, /yr should be yr$^{-1}$

L28: "constitutes" isn't the right word. NPP is the act of fixing the carbon, while the phytoplankton themselves are the basis of the food chain.

L30-31: Add a reference for this.

L35: north -> North

L37: remove "here, ", but also consider simplifying this sentence.

L53: "The seasonal cycle of phytoplankton blooms has been explained with various theories" -> "Several mechanisms have been hypothesized to explain the seasonal cycle of phytoplankton blooms." (Suggestion)

L53-59. There's a bit of a confusion here about theories vs hypotheses. A theory is specifically a widely accepted and tested hypothesis (like gravity, evolution or similar). So by definition, these competing explanations can't all be theories! Also, I don't think that the critical depth hypothesis can be both a hypothesis and a theory. I recommend rephasing this paragraph so that these are called hypotheses, explanations or mechanisms, instead of theories.

L62: Please be more explicit with your definition of phenology. It's the core of the paper and its in the title. It deserves a full definition.

L76: " a maximum temporal resolution of not more than 20 days is required." -> "a temporal resolution of 20 days or less is required."

L79 "In this paper...": long sentence

L88: "highlights at which leads and lags" is there a missing word here? Consider simplifying this sentence

**Methods**

L91: Is the NPP integrated to the sea floor or to some other depth?

L92: "100 years pi-control"- > "100 years of Pre-industrial Control (piControl)" Note that there several spellings of PI-control here, you should choose one.

L92: "Kriegler et al., 2017" doesn't exist in the references.

L92: Can you justify why only one scenario and why you chose SSP5-8.5?

L94: Please check the submission guidelines for referencing sections. This should be: "The models are described in Sect. 2.1. Section 2.2 describes the observational data set and Sect. 2.3..." https://www.biogeosciences.net/submission.html

L96: "which is calculated as a simple max of NPP" -> "which is calculated as the annual maximum of the daily means NPP, in units of mgC $m^{-2}$ $d^{-1}$."

L96: Do you calculate the regional mean first and then the annual maximum? Or do you calculate first the spatial distribution of the annual maximum and then the regional means?

L100: comma after EC-Earth3-CC and NorESM2-LM.

L100 and L104: unit superscripts: cm2 should have a superscript $cm^2$, $s^{-1}$ and kg $m^{-3}$.

L104: I don't think that these two methods are compatible with each other. While I note that you never compare the two MLD datasets, do you expect the differences here to impact your conclusions? If not, please explain why.

L110: This reference is authored by "Gurvan Madec and the NEMO team", so perhaps should be an et al, or the NEMO collaboration or similar.

L114: please use colons: "PISCES is a mixed Monod-quota model simulating two different phytoplankton functional types: diatoms and nanophytoplankton, two size classes of zoo-plankton: micro and meso, and the nutrients: nitrate, ammonium, phosphate, iron and silicate."

L116: Add a reference to Redfield.

L124: remove comma after EC-Earth3.

L125: North-west -> Northwest.

L124 and elsewhere: "ocean only" -> "ocean-only"

L124: "Skyllas et al. (2019) validated EC-Earth3, in an offline ocean only NEMO-PISCES version, for a north-south (29-63oN) transect in the North-west Atlantic using cruise data of temperature, salinity and nutrients and chlorophyll-a and found a good agreement with observations."

Suggest re-writing this as: "Skyllas et al. (2019) showed a good agreement between EC-Earth3 and temperature, salinity and nutrients and chlorophyll-a observations in an offline ocean-only version of NEMO-PISCES, for a north-south (29-63°N) transect in the Northwest Atlantic."

L138: 2o -> 2°

L140: Assman et al.(2010). This should be Assmann, and add a space after et al.

L141: replace "with one phytoplankton and one zooplankton compartment and" with "with one phytoplankton functional type, one zooplankton functional type and"

L142: Is this nitrogen and phosphorus or nitrate and phosphate?

L158 and L168: move the weblink to a reference.

L159 please define MODIS.

L166: "Division of the global ocean into biogeochemical provinces has been done in a number of references" —> "The division of the global ocean into biogeochemical provinces has been attempted several times" (you can probably find more recent attempts as well.)

L176: Replace "and" after (NADR) with a comma.

L184: C-> Carbon

L188: "Generally speaking" is colloquial.

L191: "we directly pick the number of change points to find" -> "we directly pick the desired number of change points"

L191: Remove "in fact,"... This should all be facts, lol.

L193-194: extra space between "to  instead" and "that   is"?

L200: remove "in the following"

L201: "all sorts" is colloquial.

L206: ruptures was previously capitalised: Ruptures.

L207: You haven't described the cross-correlation method yet.

**Results**

L211: "for March" -> "for the March"

L213: "the internal variability of the climate system as modelled by the two ESMs is not in sync with that in reality or with each other." -> "the internal variabilities of the two ESMs climate systems are not synchronised with nature or each other."

L222: Is the yellow blob of overestimated NPP in the Spring EC-Earth3 related to the Döscher et al (2022) result that the model has too much active convection in the Labrador Sea?

L240: "In general, EC-Earth3-CC is closer to the CAFE data in size but NorESM2-LM is closer in timing." Can you back this up with some kind of objective statement?

L246: New paragraphs is missing a blank line.

L247 and elsewhere: Please fix the units.

L257-258 and elsewhere: yr-> year. Please use "year" in prose and use "yr" when it is a unit.

L265 & 309: Fig -> Fig. (Maybe elsewhere too)

L296: What is going on in EC-Earth3-CC ARTC in figure 7? There's such a wide range of behaviour there, it's hard to see why the charge point is placed where it is. Is there any way to quantify the quality of the fit – because this looks like a poor fit!

L315: I don't fully understand the value of the cross correlation analysis. Surely it's obvious that the current MLD is going to have the most impact on the current year's NPP? Maybe there's more to it here, but I think if you must include this analysis in your final paper, it needs to be explained with a bit more precision, details and specificity for those of us that don't use it on a regular basis.

L324: "lurking variable" could use a bit more explanation.

**Discussions**

L329: "the size of peak NPP was well captured by the ESMs". I'm not sure that this case has been made. You could just as easily argue from Figure 1 and Table 1 that neither model captures the observations very well. I'd like to see a robust statistical comparison of the model and data over the historical period. A graphical version like a pair of Taylor or Target diagrams or add the bias, deviation, and correlation of the annual time series to Table 1.  (I repeat this in the general comments above).

L336: north -> North

L345: Several modelling centres have contributed daily surface chlorophyll to CMIP6, but no one has contributed daily NPP. Do you want to make the case to include daily NPP (intpp) as a standard variable in future CMIP experiments?

L351: Remove "A noteworthy observation is that"

L352 and elsewhere: 21st should be 21$^{st}$

L355: IPCC 2022 is not an appropriate reference here. It's 3000+ pages long! At the very least, please cite an individual chapter. Also, this citation is for WG2, and the information you cite is more likely to be in WG1. You should probably instead cite O'Neill 2016 https://gmd.copernicus.org/articles/9/3461/2016/ and Riahi 2017 https://www.sciencedirect.com/science/article/pii/S0959378016300681.

L372: "NPP and its timing is, of course, both in the models and in reality dependent on many other factors in addition to the MLD. Some examples are light availability, nutrient concentrations and temperature." -> "In both models and in nature,  NPP and its timing is dependent on many other factors beyond the MLD, include light availability, nutrient concentrations and temperature."

L374: remove "it is clear that"

L379 & L387:  earth -> Earth

L379-385: Is there any indication of a difference between PISCES's phytoplankton functional types in terms of how climate change will impact their bloom onset phenology? Either here, in the monthly data, or in literature?

L391-393: While I more or less agree with this sentiment, I think it's a bit of a reach. You may need more to back up this statement. I'm not convinced that understanding bloom phenology will be better served by ESMs with only one PFT than an ocean-only model with multiple PFTs. This could be my personal bias talking – but I think there's definitely scope for both approaches.

**Additional discussions to consider:**

L394: The abstract concludes with "This highlights the need for long term monitoring campaigns in the North Atlantic.", but this is never discussed or mentioned again. Please add some discussion around this idea.

L394: I'd also like to see a discussion around the consequences of the changes these models have projected. As you mention in L35-L39, "the north Atlantic is a region of particular importance for carbon sequestration in the deep ocean." How will the changing phenology impact Carbon sequestration and deep mixing? Which regions are the most important and how will they change? Is the drop in NPP likely to affect higher trophic levels? How does this interact with biodiversity and marine policy?

L394: Several of the figures are not explicitly mentioned in the discussion. Please add links to the relevant figures where you discuss them, and make sure that all figures (except maybe fig 1) are mentioned in the discussion.

L394: What are the limitations of change point analysis? What does it mean when we pass a change point? How should this be interpreted? Is it like a regime change?

L394: Figures 2 and 3 shows that neither model is amazing at representing the historical behaviour. How much can we trust the projections of models that fail to capture historical observations? (I realise that we have no other tools available – but allow me to play devils advocate here!)

L394: Can you discuss the long-term projection of NPP in figure 4? In most regions (except NASE), it looks like both models project a rise in NPP? Can you compare these two models against the CMIP6 mean, for instance from https://www.frontiersin.org/articles/10.3389/fclim.2021.738224/full, which shows a decline in North Atlantic NPP in both CMIP5 and CMIP6 multi-model means.

L394: Similarly, considering that SSP5-8.5 is the most extreme climate change scenario (where future fossil fuel emission grows to 5x current values!), it barely impacts the NPP in figure 4. Is it possible that these two modelled ecosystems are particularly insensitive to climate change? Are they suitable for this type of analysis or are more flexible BGC models neccesairy for projecting the impact of climate change on the marine ecosystem?

L394: I'd like to see some discussion about the suitability of the SSP5-8.5 scenario. It has extremely high fossil fuel emissions and subsequent warming, which is likely to move these change points earlier in the simulation than you would see in other scenarios.

L394: NorESM2-LM has a particularly low (but feasible) Effective Climate Sensitivity or 2.54K (https://gmd.copernicus.org/articles/13/6165/2020/), while EC-Earth3 has an ECS of 4.3K

([https://gmd.copernicus.org/articles/13/3465/2020/](https://gmd.copernicus.org/articles/13/3465/2020/)). Does the difference in their sensitivity to carbon impact the overall conclusions? For instance, is it possible that the surface waters of EC-Earth3 will warm more than NorESM2-LM, and this may shift the locations of habitats in these models.

**Conclusions:**

L402: This is the first mention of the growing season. Please add a description of these around your figure 2 results in lines 220.

L416-419: Please explain why this is important?

**Figure captions:**

L655: Seasonal mean vertically integrated NPP-> "Vertically integrated seasonal mean NPP"

L686: There are several issues with white space here. Including an extra space in (L1) and (L2), here and in the caption for figure 7.

L694: remove period from ".Figure 7".

L698: No period at the end.

Figure 2:

- This colour bar should be made with a pointed end at the maximum value (use extend = 'max' in matplotlib), to indicate that the highest values shown are beyond the end of the scale.
- It's a shame you don't include DJF, as it looks like both models really struggle to capture the behaviour then in figure 3.

Figures 1, 3 and 9:

- It would be nice if the line colours matched the colours in the map (and the region labels in figure 4). See my comments earlier.

Figure 6 & 7:

- While I understand why all figures share the same y axis, perhaps this figure would be better served by each region showing it's own bespoke range so that the change point is easier to see. If you have to, you can move the region label above the axes.
- Please add ticks to the x axes of the top 6 panes.
- Move the legend outside the figure – ideally below the main figure.
- Can you also the solid and dashed lines to the legend.
- Replace the sidewards pointing triangle with a vertical pointing triangle.
- Is there any observational data for this that can be added? If you can't find anything, WOA has monthly MLD: [https://www.ncei.noaa.gov/access/world-ocean-atlas-2018/bin/woa18.pl?parameter=M](https://www.ncei.noaa.gov/access/world-ocean-atlas-2018/bin/woa18.pl?parameter=M)

- In the past, I've seen change point analysis that included a trend line either side of the change point. Is there any reason why this was not included here?

Figure 8

- You don't need the thick black contours here, just the colour scale might be more readable. As it stands, the figure really emphasizes the regions where the first change point is before the year 2000.

Figure 9:

- Can you remake this plot with the same style as figure 6 and 7?
- Can you highlight the times when you're within the 95% confidence bands, perhaps by making the line or dots thicker, or making them thinner when you're outside the confidence bands?

**Supplementary data**

There's no readme to describe the contents of each file.

**Supplementary Model**

The directory structure is not straightforward to understand. The structure described in the Readme doesn't match up with the directories in the zip. Please make the leading directory names more explicit.

---

## Referee Report (RR3)

**Response to revised "Net primary production annual maxima in the North Atlantic projected to shift in the 21st century."**

Jenny Hieronymus, Magnus Hieronymus, Matthias Gröger, Jörg Schwinger, Raffaele Bernadello, Etienne Tourigny, Valentina Sicardi, Itzel Ruvalcaba Baroni, and Klaus Wyser

Submitted to Biogeoscience, BG-2023-54

Review by Dr. Lee de Mora, Plymouth Marine Laboratory, Plymouth, UK

Thanks for your hard work. The abstract, introduction are all significantly improved, with a much clearer flow and logic. The methods and results sections have been streamlined and the overall tone of the text makes it easier to follow. The revised figures look great and support your conclusions. I'm also glad that you are now using consistent colouring for the regions in your figures. It does really help! As does adding regular reminders of the region names in the text.

Also, in addition to the changes to the manuscript, thanks for your patience in answering my questions in the Author's response. This has helped me understand the work and clarified the overall paper.

I'm happy for this paper to accepted subject to a small number of very minor changes described below.

This is a great work and congratulations.

Lee de Mora

**Minor comments:**

Please note that the line numbers here refer to the tracked changes document:
https://editor.copernicus.org/index.php?_mdl=msover_md&_jrl=11&_lcm=oc104lcm105m&_acm=get_authors_tracked_changes_file&_ms=110119&id=2268736&salt=1650201345432405649

L163 – Please add a sentence justifying the use of SSP5-8.5 (as you have in your response to the previous review.)

L243: Move the link to a reference.

Results section: It looks strange to me to refer to a decade as 2085s? Typically, the decade 2080-2089 would be the "2080s", not 2085s.

L412, L415: When you link to a software routine, it's best to use a monospace font like `Courier New` or something, or the command: \texttt{ks_2samp} in latex. ie `ks_2samp, crosscorr`

L637: "That is, the large range of correlated lags indicate that the whole story is not simply that MLD acts as a control on NPP." – this is a bit strangely worded, can you re-phrase it please? Maybe something like: "The large range of correlated lags indicate that that NPP is likely to be controlled by other factors, in addition to MLD." Or something like that?

Finally, the conclusion sections needs a closing statement of some kind.

Bibliography:

- DOI's sometimes have an http link and sometimes don't?
- L729:  Extra spaces around "Sverdrup ' s"
- Sometimes author lists are separated by "and" sometimes by "&"
- Sometimes journals are abbreviated and sometimes they're fully named
  - ie: L 764: Geoscientific Model Development but L767: Geosci. Model Dev
- L898: Elsewhere Authors initials have periods: , ie: Smyth **T.J.**

As I said in my previous review, a citation manager should be able to automate this next time. It will save time in the long run. However, BG's typeset editor will likely notice (and hopefully correct) other issues here too.

---

## Author Response (AR2)

Review to

**Phenological shifts in the North Atlantic net primary production detected in the 21st century. Results from two Earth system models.**

Jenny Hieronymus et al.

General comments:

The authors did a great job in revising the analysis. By dividing the North Atlantic into Longhurst provinces, the results are much more consistent and meaningful. Interestingly, the most of the provinces show an increase of NPP until the end of the century except two regions in both models (disregarding the small changes in SARC and NADR in NorESM2-LM). This result was masked in the previous analysis where a declining trend in NPP was postulated for the entire domain over SSP8.5. This finding has to be included in the Summary and Conclusion section which still mentions an overall NPP decrease (L408). In addition, I do feel that many of the final findings are related to results from the EC-Earth, e.g. "phenological shifts occurring in the early 21st century " is not true for NorESM (in 6 out of 8 regions the changepoint is after 2048). Please critically review the entire manuscript to see if the final statements apply to both ESMs.

**Authors: We have reviewed the manuscript and made sure that final statements apply to both ESMs.**

A general remark on the quality of the figures:

● the increment of contour lines should be specified for the subplots in the caption; e.g. in Fig 3 each of the SON panels has different increment
● contour lines in Fig.8 are horrible – delete or omit the entire Fig. (see specific comments)
● I plea for a,b,c notation in the figures for more readability

> We tested making the figures with that notation during our revisions, but found that it became less obvious what the different panels contained and also that the figure caption became much less readable. We therefore think this is the least worst option for our set of panels.

**Authors: We have specified the contour increment. We have removed the contour lines in Fig 8.**

In general, I recommend the publication of the manuscript after my specific comments have been addressed.

**Authors: We thank the reviewer for the thorough review that has greatly improved the manuscript.**

Specific comments:

L25: Please correct: Net Primary Production (NPP) is the rate of photosynthetic carbon fixation minus cellular respiration

**Authors: This has been corrected.**

L82-83: "We divide the region into biogeochemical provinces (Longhurst et al., 1995) in order to see how localities with similar biogeochemical functioning differ across the region." This sentence is confusing. What do you mean by "localities"? Do your provinces really have a similar biogeochemical functioning? Delete "Furthermore".

**Authors: This has been changed.**

L85-88: Please motivate here the purpose of MLD analysis and reorder the sentences - first: change point analysis for MLD as for peak NPP; second: all about cross-correlation and what we learn from it.

**Authors: We have rewritten as suggested.**

L94: typo: in section 2.4 is the change point analysis L96: "maximum" instead of "max"

**Authors: This has been corrected**

L97: "found in your data" ESM data or CAFE or all data sets?
**Authors: We were referring to the ESM data, but the statement is true also for the CAFE data.**

L117: replace "external concentration in nutrients" by "nutrient concentrations of the ambient water"

**Authors: This has been corrected.**

L118: Please give the same information for both BGC modules. i.e. delete :"PISCES is suited for a wide range of spatial and temporal scales, including quasi-steady state simulations on the global scale."
And add for iHAMOCC, that iHAMOCC also simulates the carbon system, as well as dissolved and particulate organic matter"
**Authors: This has been corrected.**

L119-120: "Net primary production is the growth of phytoplankton thus the term excludes mortality, excretion and grazing." Why is this mentioned here? By definition, NPP excludes mortality, excretion and grazing. Don't mix it up with NCP = net community production. Delete sentence?

**Authors: The line has been removed.**

L163: Rephrase your sentence to e.g. : "The seasonality of NPP depends, among other things, on local physical conditions of the ocean" ?

**Authors: This has been changed.**

L168: Longhurst defined the static boundaries – "made" is a strange word?

**Authors: This has been changed.**

L171: You never use "coastal, westerlies and polar" – delete; The North Atlantic domain is divided in the provinces shown in Fig. 1.

**Authors: This has been changed.**

L176 delete: The west wind regions;

**Authors: We have deleted this.**

L209ff: I recommend to show and discuss only MAM and JJA and omit SON. It shows a more or less a uniform pattern for the entire domain and complicates the data processing due the lack of data in CAFE in winter. SON gives no additional information. In addition, please find a better color scale. It is surprising, that your scale ends at 1000 but Fig.3 shows numbers higher than 1200. Please correct.

**Authors: We have removed the SON panel and changed the colorscale.**

L226: Instead of using daily ESM data, use a 8-day running mean for the comparison to 8-day mean data from CAFE. Results in Fig.3 are difficult to compare. Please reorder the seasonal cycles by region instead of data sets: e.g. BPLR+ARCT for CAFE and both models, and so on. Adjust axes to maximum values. Make sure that all lines have the same starting point if you mask the ESM data with available CAFE data.

**Authors: We have done this (Fig. 3).**

L264: make sure, that you don't use the word "region" for both, the entire North Atlantic and the provinces; use e.g. the words "domain" and "provinces" throughout the manuscript.

**Authors: This has been done.**

L265ff: could you improve the readability by shorten the name of the 3 periods: e.g. 1865s = 1850-1879, 2000s= 1985-2014, 2085s= 2070-2999? Then you can omit to write "period" or "early/late period"

**Authors: We have changed the names of the periods as suggested.**

Fig 5: Please use a standard statistical test (e.g. student's t-test) to determine the significance. With the given information, it is difficult/impossible to interpret the results. Please show results of EC-Earth on the left side as usual.

**Authors: We have performed a Kolmogorov-Smirnov test and marked the significance to the 95th percentile in the bottom panels in Fig. 5. However, we do think that the pattern of change divided by the standard deviation of the piControl adds to the analysis as we obtain a measure of how far we are from the natural variability of the piControl.**

L277 " size of NPP" – delete "size of"

**Authors: This has been done.**

L284: you don't average over different provinces, rephrase.

**Authors: We have rephrased.**

L285: Fig. 6 shows …. together with the largest (…. sentence incomplete

**Authors: This has been corrected.**

L304: The posed question was reasonable for the previous analysis, but I cannot see the benefit when using Longhurst provinces. Isolines in Fig. 8 should be removed, if not the whole figure is omitted or transferred to the supplement. In the supplement you could also add the discussion on the difference between the PELT method and Fig.8 and why one has blanks and the other not.

**Authors: We have removed the isolines in the figure. We do, however, think that the figure adds to the analysis. Mostly because there can be considerable variations also within the Longhurst provinces and also because it gives an indication of the suitability of these regions for the ESM data. An interesting avenue of future research would be to let an objective algorithm find provinces and then compare those to the Longhurst ones.**

L316: province averaged instead of area-averaged? Or just write: "between the time series in Fig 6 and 7" because it is clear how they were archived.
**Authors: "area-averaged" has been removed.**

L320: Typo? NADW is not defined
**Authors: This has been changed to NASW.**

L323: "Looking at Fig. 8 ….. you mean Fig 9 ?
**Authors: We mean Fig 7. This has been corrected.**

L329: you never show the "size of peak NPP". delete "size of peak" or explain what you mean
**Authors: We have revised this statement.**

L351: use "finding" instead of "observation"
**Authors: This has been changed.**

L352: replace "then that…" with "when the warming is the strongest in the SSP5-8.5"
**Authors: This has been changed.**

L392: replace "realistic physics" with "consistent physics"
**Authors: This has been changed.**

L400-401: rephrase: you don't use Longhurst provinces to look at spatial averages, but to account for the different areal conditions
**Authors: This has been rephrased.**

L408: As already mentioned above, the NPP increases for many provinces. Revise!

**Authors: This has been revised.**

**Phenological shifts in the North Atlantic net primary production detected in the 21st century. Results from two Earth system models.**

Jenny Hieronymus, Magnus Hieronymus, Matthias Gröger, Jörg Schwinger, Raffaele Bernadello, Etienne Tourigny, Valentina Sicardi, Itzel Ruvalcaba Baroni, and Klaus Wyser

Submitted to Biogeoscience, bg-2023-54

Review by Dr. Lee de Mora, Plymouth Marine Laboratory, Plymouth, Devon.

**Summary**

In this work, the daily Net Primary production of two Earth System Models in the Northern Atlantic are described, compared against satellite data and analysed using change point and cross corelation analysis for several regions of the North Atlantic. The timing of the peak of the bloom will shift earlier in the year in the Northern parts of the North Atlantic. The models disagree for the Southern North Atlantic, but it is less of a shift than in the northern regions. The change point analysis highlights that several regions are likely to have pass the change already and that nearly all regions will cross the change point in the 21st century. However, it's not clear how significant the scale of the change point will be.

The text is well written, the underlying science is well introduced in a clear way, the results are presented and described accurately, I did not spot any spelling mistakes and the grammar is almost always fine. At times, the style is a little colloquial and would be improved if certain parts were written in a more formal style. There's a few run-on sentences which need to be pruned. There were many formatting issues, described below, and there are likely many more that I missed.

The figures are generally clear, but I suggest a few improvements below which I think would benefit the paper as a whole.

I found that there were a few discussion points that were hinted at in the abstract and introductions, but never made it to the final draft. I have a few questions below, a few suggestions and a few possible additions.

In my opinion, the biggest weakness of the paper as it currently stands is that the key results are not sufficiently well articulated. It's crucial that the revision of this paper focuses on its unique results and explains why they are important as clearly as possible.

I would also recommend a careful and thorough readthrough (including the references section) before re-submission. Sometimes it's better to share this task with a more distant co-author, as the lead author is often too close to the work to spot these issues.

As the list of changes below has become rather long, I recommend major corrections before reconsideration. However, I don't want to come across as being harsh. This is a good, well written paper, with good scientific content, it fits within the remit of the journal, and most of these changes should be resolvable with minimal effort.

**Authors: We thank Dr. Lee De Mora for the ambitious, in depth review that has led to a greatly improved manuscript.**

**Specific Comments**

I expect a more forceful and direct tone in the title, abstract and the conclusions. At the moment, the abstract focuses on the methods, but it should effectively read: "This is our main result. This is why it

is important." Then once you've said that, only then you can describe the methods and models that were used to found out.

Similarly, the title could be a more direct and effective. Something like: "Net Primary Production Annual Maxima in the North Atlantic projected to shift in the 21$^{st}$ century" or something like that. (As an aside, I don't think you need to mention ESMs in the title – it's obvious that models were used if you're making projections of the future!)

**Authors: The Abstract and Conclusions has been rewritten as suggested. The title has been changed to that suggested by the reviewer.**

I'm not convinced that either model captures the observations over the historical period. The model-data comparisons in figure 1 and table 1 are subjective. I'd like to see a robust statistical comparison of the model and data over the historical period. A graphical version like a pair of Taylor or Target diagrams would be a solid improvement. Alternatively but probably less effective, you could add pattern statistics (bias, deviation, and correlation) to Table 1. Please bear in mind that you have made a new and unique piece of work, and people in the future will be glad to have a robust statistical benchmark to cite that they can compare their model quality against.

**Authors: We went through this also a bit in the first review, but we think it needs to be iterated why we don't believe this idea makes sense. Firstly, we are not comparing two models to a set of observations. We are comparing two earth system models to a different kind of model that utilizes satellite observations to infer phytoplankton biomass. These are not observations of phytoplankton biomass. This means of course that the CAFE model may have sizable errors, just as the ESMs might have. Moreover, while the CAFE model by construction models our recent history, the two earth system models do not. These are free running coupled models. The thing they have in common is that they are driven by similar greenhouse gas and aerosol forcing as our history. The climate variability is, however, not in sync. All relevant climate indices like NAO, AMO or AMOC will be out of phase in these three different models. Therefore a conventional Taylor diagram looking at temporal correlations would be meaningless. An early bloom in one model should not imply an early bloom in another during the same year, and so on. The only similarities we can hope for are climatological in nature. Long averages should hopefully be similar, but it is hard to say given the shortness of the CAFE model's time series, if that is indeed the case with our different models. One could in principle do a Taylor diagram with spatial correlations, but given that the temporal unsynced variability will also affect the spatial correlations, such a plot would be nearly impossible to interpret. Two twenty year periods could have rather different spatial patterns owing to different phases in the AMO for example. In conclusion, our unwillingness to put more advanced statistical methods into this comparison is because it would be a lot like looking at correlations between random numbers.**

It would be valuable to include the analysis of the mean of the whole North Atlantic region in some of these figures (ie Figure 3, Tables 1, 2 and 3 as well?) I understand the value of the individual regions, but a clear result for the whole region would be a good headline result.

**Authors: We have added values for the whole region in Tables 1 and 2.**

I would be interested in seeing how different the phenologies of the various regions are on either side of the change point. Basically, a version of figure 3 which compares the climatological mean of ten (or some useful number of) years each side of the change point. This would be a clear and effective way of showing that there is indeed a real change between those two periods.

**Authors:**

**We disagree with this approach. The change points found are real, in the only sense that change points can be real, that is, they are found by some algorithm to satisfy some objective, albeit arbitrary, condition imposed by us to define change points. One might think change points that are change**

**points according to multiple such definitions to be in some sense more real than others, which is one of the reasons for including the L1 and L2 method. The question you pose has more to do with whether they can be identified rather subjectively by a human eye, than whether they are real. We think such a figure could be useful only in very clear cut cases to illustrate some change that has indeed occurred, but for that we think Fig. 5 is a better illustration.**

There is no daily net primary production data from CMIP6 on ESGF, but there is quite a lot of surface chlorophyll (chlos) and surface phytoplankton carbon (phycos). It would be interesting to place these two models against the rest of CMIP6 in the context of the phytoplankton carbon or chlorophyll. Are they typical or are they outliers? This would be a lot of additional work, so I leave it up to the authors to decide whether they can perform the additional analysis. If not, then maybe add it as a suggested extension in the discussion.

**Authors: An article comparing these metrics across CMIP6 models would certainly be useful, but it is clearly material enough for an article in its own right. We don't think such an analysis really belongs here. We have added a line on the necessity of more models and scenarios in the conclusions. (Lines 470-473)**

As I mentioned, several modelling centres have contributed daily surface chlorophyll and phytoplankton carbon to CMIP6, but no one has contributed daily NPP. Do you not want to make the case to include daily NPP (intpp) as a standard variable in future CMIP experiments? What do we gain from including NPP that we don't get from chlorophyll and carbon?
**Authors: We have added a line on this in the Conclusions. (Lines 466-469)**

At the end of the discussion section, I found myself asking several follow up questions like these. I have listed these below with the label "L394".

**Typesetting & style comments**

There is a tendency for sentences to be too long and complex, which makes them harder to read and parse - particularly in the abstract. While they may be accurate, they take more effort to understand. For this reason, I personally have a strong preference for simpler shorter sentences. I have pointed out a couple in the abstract and made some suggestions on ways to shorten and split them.
However, I'll leave it up to you from that point.
**Authors:We have shortened long sentences.**

Please try to be consistent with hyphenation and capitalisation. Change-point, time-series, cross-correlation, North-west, PI-control are all written in several different ways throughout the text.
**Authors: This has been corrected.**

There are several places where the superscript is lost for both the degree symbol, units and centuries (especially in the title!). Please be more careful with subscript and superscripts.

**Authors: This has been corrected.**

There are a few places where the text is stretched: L157, L543, L691.

**Authors: This has been addressed.**

For references in the body of the text, there are a few places where the name in the text either doesn't match the reference, or a reference does not exist. Similarly, there is some variability in typography of the "et al.", sometimes the period is missing (L336) and sometimes there's no space before the year (L140).

**Authors: We have corrected this.**

In the reference section, there are a lot of inconsistencies:

● Several references with strange characters that need to be corrected. Ie L471, 474.
● Some DOI's are links in blue and some are not, ieL 462 vs L464.
● Most references have the author initials, but L475 includes author's first names.
● Some are missing DOI's ie L480.
● Some place the year at the start (L461) and some at the end (L466).
● Some times the journal name is in italics (L468) and sometimes it is not (L464).
● Some references include et al. (L494 & L574) but most do not. Some instead use "…" (L501, L556, L562 & L594)

To me, this suggests that the author is manually writing the bibliography – which just makes your life more difficult! If you are, I strongly recommend instead using some kind of reference management tool to keep track of the bibliography and ensure that references are done properly and consistently. (I use bibtex for latex).

**Authors: We have revised the references.**

Without being an expert on the North Atlantic, I find the 8 different region names to be confusing. I'm constantly having to refer to figure 1 to check what region is being discussed. It may be clearer to write where things are happening more descriptively than just relying on the four letter acronyms. For instance, in line 253: "The largest standard deviation is found in NASE and the lowest in NWCS" could be clearer as "The largest standard deviation is found in the southeast of our domain in NASE and the lowest is in the Northwest Atlantic Shelf (NWCS)." This is closer to how you have described figure 5 in lines 264-274, which is much clearer.

**Authors: We have expanded the text in accordance with the reviewers suggestion.**

In the figures, I think there's scope for some additional consistency which would make it easier to interpret. For instance, you can use the same regional colours from figure 1 again in figure 3. Instead of the blue/orange colour scheme in figures 4, 6 and 7, you could use the regional colours again, but have a lighter one for EC-Earth and a darker one for NorESM2 (for instance). Alternatively, you could keep the same two line colours, but change the regional labels to match figure 1. Similarly, I see no reason for figure 9 to be different to figures 6 and 7. Finally, and this is purely subjective, but I'm not crazy about the blue and orange colours – I think a more aesthetic colour pairing could be found (https://colorbrewer2.org/ is a good resource for something like this).

**Authors: We have labelled the different plots with province names in the same color as in Fig. 1. We do think the blue/orange/green works though and will keep them. Different shadings would not work as we use shading to discern between one and two change-points in Figs. 6 and 7.**

As a added suggestion, daily data looks great in video format – much better than monthly and annual data. Have you considered including a supplementary video of the daily climatological mean for the two models and the observations? This would be a great resource to show when presenting this work in person. Alternatively, it could contribute to a great video abstract.

**Authors: Thanks for the suggestion. The timeframe of this review does not allow us to explore this option currently, but we will keep it in mind for future presentations of this work**

**Specific Points**

**Abstract:**

L14: This is a long sentence which could be clearer: "The majority of the region displays the largest change point in the day of peak NPP occurring after the year 2000 indicating a shift towards earlier peak NPP with the most change occurring in the northern parts of the domain."

Suggested change: "Most of the region has the largest change point in the day of peak NPP after the year 2000. This indicates a shift towards earlier peak NPP and the most change occurs in the northern parts of the domain."

**Authors: The sentence has been changed as suggested.**

L18: long sentence: "Furthermore, the occurrence of the first day with MLD shallower than 40 m shows positive correlation with the occurrence of the day of peak NPP for most of the domain and, similar to the day of peak NPP, displays the largest changepoints occurring around or after the year 2000."

Suggested change: "Furthermore, the occurrences of the first day with a MLD shallower than 40 m and the day of peak NPP are positively correlated over most of the domain. As was the case for the day of peak NPP, the largest changepoints occur around or after the year 2000."

**Authors: The sentence has been changed as suggested.**

L20 and elsewhere: Is it **changepoint** or **change point** or **change-point**? Beaulieu (2012) uses change-point so maybe that is the best option?

**Authors: It is now "change-point" throughout.**

**Introduction:**

L26: Turnover time is not defined and never referenced again. Why is it important?

**Authors: Turnover time has been removed from the text.**

L27: "Almost equals"? What is the land NPP value?

**Authors: There is a lot of uncertainty in these estimations. Field et al. (1998) calculates marine NPP to 48.5 and the terrestrial NPP to 56.4 Pg yr$^{-1}$. We have added this reference and changed to "similar in size".**

L27 – and elsewhere: I prefer Pg instead of Gt. Ton is not an SI unit and there are many definitions of tons/tonnes/imperial tons and it can be confusing for international readers. Also, /yr should be yr$^{-1}$

**Authors: This has been changed.**

L28: "constitutes" isn't the right word. NPP is the act of fixing the carbon, while the phytoplankton themselves are the basis of the food chain.

**Authors: The sentence has been changed.**

L30-31: Add a reference for this.

**Authors: Reference has been added.**

L35: north -> North

**Authors: This has been corrected.**

L37: remove "here, ", but also consider simplifying this sentence.

**Authors: The sentence has been simplified.**

L53: "The seasonal cycle of phytoplankton blooms has been explained with various theories" -> "Several mechanisms have been hypothesized to explain the seasonal cycle of phytoplankton blooms." (Suggestion)

**Authors: The sentence has been changed as suggested.**

L53-59. There's a bit of a confusion here about theories vs hypotheses. A theory is specifically a widely accepted and tested hypothesis (like gravity, evolution or similar). So by definition, these competing explanations can't all be theories! Also, I don't think that the critical depth hypothesis can be both a hypothesis and a theory. I recommend rephasing this paragraph so that these are called hypotheses, explanations or mechanisms, instead of theories.

**Authors: We have changed to "hypothesis".**

L62: Please be more explicit with your definition of phenology. It's the core of the paper and its in the title. It deserves a full definition.

**Authors: We have added this to the introduction (Lines:64-67)**

L76: " a maximum temporal resolution of not more than 20 days is required." -> "a temporal resolution of 20 days or less is required."

**Authors: This has been changed as suggested.**

L79 "In this paper...": long sentence

**Authors: We have changed the sentence.**

L88: "highlights at which leads and lags" is there a missing word here? Consider simplifying this sentence

**Authors: A misplaced comma resulted in a strange sentence. We have also divided the sentence in two.**

**Methods**

L91: Is the NPP integrated to the sea floor or to some other depth?

**Authors: In EC-Earth3-CC NPP is integrated to the sea floor while in NorESM2-LM, NPP is integrated over the top 100 meters. We have added this information on lines 101-102.**

L92: "100 years pi-control"- > "100 years of Pre-industrial Control (piControl)" Note that there several spellings of PI-control here, you should choose one.

**Authors: This has been changed as suggested and to "piControl" throughout.**

L92: "Kriegler et al., 2017" doesn't exist in the references.

**Authors: The reference has been added.**

L92: Can you justify why only one scenario and why you chose SSP5-8.5?

**Authors: The reason is mainly that this gives us a sort of upper boundary on potential shifts. It is also interesting to note that most of the change points, especially in EC-Earth3-CC are located before SSP5-8.5 warming deviates too much from lower SSPs, as stated on lines 382-385. The reason for using only one is mainly that these kinds of runs, where daily data is saved, require a lot of resources and SSP5-8.5 was therefore the only scenario run within the H2020 project COMFORT.**

L94: Please check the submission guidelines for referencing sections. This should be: "The models are described in Sect. 2.1. Section 2.2 describes the observational data set and Sect. 2.3..." https://www.biogeosciences.net/submission.html
**Authors: This has been changed.**

L96: "which is calculated as a simple max of NPP" -> "which is calculated as the annual maximum of the daily means NPP, in units of mgC m$^{-2}$ d$^{-1}$."

**Authors: We have changed the sentence although since the focus is on the "day of peak NPP" we did not see the reason to include the NPP units.**

L96: Do you calculate the regional mean first and then the annual maximum? Or do you calculate first the spatial distribution of the annual maximum and then the regional means?

**Authors: First we calculate the day of max NPP in every grid point and then we average over the area of each province.**

L100: comma after EC-Earth3-CC and NorESM2-LM.

**Authors: Commas have been added.**

L100 and L104: unit superscripts: cm2 should have a superscript cm$^2$, s$^{-1}$ and kg m$^{-3}$.

**Authors: This has been corrected.**

L104: I don't think that these two methods are compatible with each other. While I note that you never compare the two MLD datasets, do you expect the differences here to impact your conclusions? If not, please explain why.

**Authors: Both methods are classical ways of finding the mixed layer depth. The turbulent mixing coefficient in NEMO is typically orders of magnitude larger in the surface mixed layer than in the stratified layers below. Both methods thus essentially find the depth where stratification starts. In a perfect world we would have used exactly the same criterion, but mixed layer depths are computed online at each timestep. An offline diagnostic using the same criterion, but not done at full temporal resolution would surely be worse. The two methods are thus different proxies for the same thing. Note that apart from differences in these criterion there are plenty of other differences that can affect the mixed layer depth, like for example the vertical resolution.**

**If you change the MLD criteria a little the MLDs obviously also change a little. However, the connection between MLD and other variables are robust to such changes. Note, for example, that we have tried many different thresholds for the MLD with similar results. Thus, there is no reason to expect neither the two methods to be none-compatible nor to expect the use of different criteria to affect the conclusions.**

L110: This reference is authored by "Gurvan Madec and the NEMO team", so perhaps should be an et al, or the NEMO collaboration or similar.

**Authors: We added "et al."**

L114: please use colons: "PISCES is a mixed Monod-quota model simulating two different phytoplankton functional types: diatoms and nanophytoplankton, two size classes of zoo- plankton: micro and meso, and the nutrients: nitrate, ammonium, phosphate, iron and silicate."

**Authors: Colons have been added.**

L116: Add a reference to Redfield.

**Authors: Reference has been added.**

L124: remove comma after EC-Earth3.

**Authors: This has been corrected.**

L125: North-west -> Northwest.

**Authors: This has been corrected.**
L124 and elsewhere: "ocean only" -> "ocean-only"
**Authors: This has been corrected.**

L124: "Skyllas et al. (2019) validated EC-Earth3, in an offline ocean only NEMO-PISCES version, for a north-south (29-63oN) transect in the North-west Atlantic using cruise data of temperature, salinity and nutrients and chlorophyll-a and found a good agreement with observations."

Suggest re-writing this as: "Skyllas et al. (2019) showed a good agreement between EC-Earth3 and temperature, salinity and nutrients and chlorophyll-a observations in an offline ocean-only version of NEMO-PISCES, for a north-south (29-63°N) transect in the Northwest Atlantic."

**Authors: The sentence has been changed as suggested.**

L138: 2o -> 2°

**Authors: This has been corrected.**

L140: Assman et al.(2010). This should be Assmann, and add a space after et al.

**Authors: This has been corrected.**

L141: replace "with one phytoplankton and one zooplankton compartment and" with "with one phytoplankton functional type, one zooplankton functional type and"

**Authors: This has been changed as suggested.**

L142: Is this nitrogen and phosphorus or nitrate and phosphate?

**Authors:** It should be nitrate. This has been corrected.

L158 and L168: move the weblink to a reference.

**Authors: We have moved the weblink to Data availability.**

L159 please define MODIS.

**Authors: This has been done.**

L166: "Division of the global ocean into biogeochemical provinces has been done in a number of references" –> "The division of the global ocean into biogeochemical provinces has been attempted several times" (you can probably find more recent attempts as well.)

**Authors: The sentence has been changed as suggested.**

L176: Replace "and" after (NADR) with a comma.

**Authors: This has been done.**

L184: C-> Carbon
**Authors: This has been done.**

L188: "Generally speaking" is colloquial.

**Authors: This has been changed to; In general,**

L191: "we directly pick the number of change points to find" -> "we directly pick the desired number of change points"

**Authors: This has been changed as suggested.**

L191: Remove "in fact,"... This should all be facts, lol.

**Authors: This has been done.**

L193-194: extra space between "to instead" and "that is"? L200:
remove "in the following"

**Authors: This has been done.**

L201: "all sorts" is colloquial.

**Authors: This has been change to "all types"**

L206: ruptures was previously capitalised: Ruptures.

**Authors: This has been changed.**

L207: You haven't described the cross-correlation method yet.
**Authors: We have used matlabs crosscorr routine. We have added a reference to this. As the cross correlation is a standard statistical method we see no reason to provide a deeper description here.**

**ResultsAuthors: We have used matlabs crosscorr routine. We have added a reference to this. As the cross correlation is a standard statistical method we see no reason to provide a deeper description here.**

L211: "for March" -> "for the March"

**Authors: This has been changed.**

L213: "the internal variability of the climate system as modelled by the two ESMs is not in sync with that in reality or with each other." -> "the internal variabilities of the two ESMs climate systems are not synchronised with nature or each other."

**Authors: This has been changed.**

L222: Is the yellow blob of overestimated NPP in the Spring EC-Earth3 related to the Döscher et al (2022) result that the model has too much active convection in the Labrador Sea?

**Authors: It is possible, although speculative. Convection in the Labrador sea does affect the AMOC and at least to some degree the Gulf Stream. However, the Gulf Stream is mostly a wind driven affair, a consequence of Sverdrup balance and lateral (Munk) or bottom (Stommel) friction. However, the blob could also have other causes, like large riverine nutrient fluxes or large lateral tracer spreading in the area. With the experiments and CAFE data we have at our disposal there is no way to prove such a connection. In fact, it is not even a given that NPP is overestimated in EC-Earth in this area during MAM. Perhaps the other two models underestimate the NPP, or perhaps NPP has considerable multidecadal variability.**

L240: "In general, EC-Earth3-CC is closer to the CAFE data in size but NorESM2-LM is closer in timing." Can you back this up with some kind of objective statement?

**Authors: We have removed this statement and added some text on lines:247-252.**

L246: New paragraphs is missing a blank line. L247 and elsewhere: Please fix the units.
**Authors: This has been done.**

L257-258 and elsewhere: yr-> year. Please use "year" in prose and use "yr" when it is a unit.
**Authors: This has been changed.**
L265 & 309: Fig -> Fig. (Maybe elsewhere too)
**Authors: This has been corrected.**
L296: What is going on in EC-Earth3-CC ARTC in figure 7? There's such a wide range of behaviour there, it's hard to see why the charge point is placed where it is. Is there any way to quantify the quality of the fit – because this looks like a poor fit!
**Authors: Not sure why you think so, but a common misconception about change points is that many think of them as purely local attributes, while in fact they signify, in some sense, optimal segmentations of a time series. That is, they depend on the global properties of the time series.**

**Here it looks to us like the time before and after the change point have very different means, which seems to be what the algorithm captures. Note also that it is not so much about a fit as it is about how much less optimal the second most optimal changepoint is compared to the most optimal and so on. In time series with a lot of variability and large jumps it is often harder to find these optimal segmentations by eye.**

L315: I don't fully understand the value of the cross correlation analysis. Surely it's obvious that the current MLD is going to have the most impact on the current year's NPP? Maybe there's more to it here, but I think if you must include this analysis in your final paper, it needs to be explained with a bit more precision, details and specificity for those of us that don't use it on a regular basis.
**Authors: The basic idea with the lagged correlations is to test the assumption that it is obvious that the timing of MLD shoaling is the most important. We find this to not always be the case. Had it been a simple story of the current MLD controlling day of peak NPP, one would have seen a big peak at lag zero and nothing else. We find instead strong correlations at a large range of lag and lead times. This we interpret as a lurking variable. That is, instead of simply having MLD controlling NPP, we have more hidden variables controlling both MLD and NPP. This hidden "variable" is climate change that for example through temperature, salinity and sea ice changes affect both MLD and NPP and gives rise to covariance over much longer than yearly periods.**

L324: "lurking variable" could use a bit more explanation.
**Authors: We have changed the sentence in accordance to the reply above on lines 343, 401-403.**

**Discussions**

L329: "the size of peak NPP was well captured by the ESMs". I'm not sure that this case has been made. You could just as easily argue from Figure 1 and Table 1 that neither model captures the observations very well. I'd like to see a robust statistical comparison of the model and data over the historical period. A graphical version like a pair of Taylor or Target diagrams or add the bias, deviation, and correlation of the annual time series to Table 1. (I repeat this in the general comments above).
**Authors: We have revised this statement on lines:348-352. For the issue of deeper statistical comparison with CAFE we refer to the answer to the reviewers general comment above.**

L336: north -> North
**Authors: This has been corrected.**

L345: Several modelling centres have contributed daily surface chlorophyll to CMIP6, but no one has contributed daily NPP. Do you want to make the case to include daily NPP (intpp) as a standard variable in future CMIP experiments?
**Authors: As answered above: We have added a line on this in the conclusions (line: 467)**

L351: Remove "A noteworthy observation is that" L352 and elsewhere: 21st should be 21$^{st}$
**Authors: This has been done.**

L355: IPCC 2022 is not an appropriate reference here. It's 3000+ pages long! At the very least, please cite an individual chapter. Also, this citation is for WG2, and the information you cite is more likely to be in WG1. You should probably instead cite O'Neill 2016
https://gmd.copernicus.org/articles/9/3461/2016/ and Riahi 2017
https://www.sciencedirect.com/science/article/pii/S0959378016300681.
**Authors: We have referenced the above instead.**

L372: "NPP and its timing is, of course, both in the models and in reality dependent on many other factors in addition to the MLD. Some examples are light availability, nutrient concentrations and temperature." -> "In both models and in nature, NPP and its timing is dependent on many other factors beyond the MLD, include light availability, nutrient concentrations and temperature."
**Authors: This has been changed as suggested.**

L374: remove "it is clear that"
**Authors: This has been done.**

L379 & L387: earth -> Earth
**Authors: This has been done.**

L379-385: Is there any indication of a difference between PISCES's phytoplankton functional types in terms of how climate change will impact their bloom onset phenology? Either here, in the monthly data, or in literature?
**Authors: We have not checked this as we do not have any daily data of the two functional types.We do expect that different biogeochemical models as well as different functional types will respond differently to the same physical forcing due to the differences in their response to temperature, mixed layer depth etc. In a recent article Kléparski et al. (2023) investigated the climate change response of different functional types in six CMIP6 ESMs. Their main result is that the seasonal duration as well as the biomass of the diatom bloom will decline and the biomass of the slower sinking dinoflagellates will increase with possible impacts to carbon sequestration in the deep ocean. We have added some text to the discussion (lines:423-426)**

L391-393: While I more or less agree with this sentiment, I think it's a bit of a reach. You may need more to back up this statement. I'm not convinced that understanding bloom phenology will be better served by ESMs with only one PFT than an ocean-only model with multiple PFTs. This could be my personal bias talking – but I think there's definitely scope for both approaches.
**Authors: Nowhere in that section is it suggested that ESMs with one PFT are superior to ocean only models with multiple PFTs. Nor do we advise against using either more or less comprehensive models. Rather, we think that a considerable range of different models are useful to tackle questions of the nature addressed here. Complexity usually comes at the cost of interpretability and computational time; what's optimal depends on one's means as well as analytical strengths and presumably several other things. However, keeping the ocean model constant we think few would argue against our assertion that coupled effects could be important, and that coupled simulations are more physically realistic (boundary conditions are more inline with those of the real atmosphere-ocean system) than uncoupled ones.**

**Additional discussions to consider:**

L394: The abstract concludes with "This highlights the need for long term monitoring campaigns in the North Atlantic.", but this is never discussed or mentioned again. Please add some discussion around this idea.
**Authors: We have added this in the discussion (lines:387-388 ).**

L394: I'd also like to see a discussion around the consequences of the changes these models have projected. As you mention in L35-L39, "the north Atlantic is a region of particular importance for carbon sequestration in the deep ocean." How will the changing phenology impact Carbon sequestration and deep mixing? Which regions are the most important and how will they change? Is the drop in NPP likely to affect higher trophic levels? How does this interact with biodiversity and marine policy?
**Authors: As we state in the introduction, changed seasonality may impact entire ecosystems through trophic level decoupling. This could have an impact on carbon sequestration through a reduced strength of the biological pump.  We have added some text on this on lines:436-440.**

L394: Several of the figures are not explicitly mentioned in the discussion. Please add links to the relevant figures where you discuss them, and make sure that all figures (except maybe fig 1) are mentioned in the discussion.
**Authors: All figures are now referenced in the discussion.**

L394: What are the limitations of change point analysis? What does it mean when we pass a change point? How should this be interpreted? Is it like a regime change?
**Authors: The main limitations or perhaps better pitfall, we would say is that it is easy to overinterpret statistical measures as if they also by necessity have a physical or biogeochemical meaning. This is not exclusive to changepoints. In fact, by far the debate has been most lively about statistical significance and its interpretation, which in some fields (fortunately seldom our**

own) is very often wrongly interpreted. All the same mistakes done with statistical significance can also be done with change points. The interpretation we find most valuable is that they give in some, well defined albeit arbitrary, sense an optimal segmentation of one's data. Change points can mark regime shifts, although we think of regime shifts as a more physical or biogeochemical than a statistical trait. However, how a regime shift is defined is purely a lexical semantics question, of little concern for the current study.

L394: Figures 2 and 3 shows that neither model is amazing at representing the historical behaviour. How much can we trust the projections of models that fail to capture historical observations? (I realise that we have no other tools available – but allow me to play devils advocate here!)
Authors: Since the earth system models are not synchronized with real world internal climate variability, they can not be expected to reproduce the historical patterns correctly. Furthermore, the models are not eddy permitting which will generate discrepancies. The indications the models give about future behaviour are therefore taken to be indications based on future climate change effects.

L394: Can you discuss the long-term projection of NPP in figure 4? In most regions (except NASE), it looks like both models project a rise in NPP? Can you compare these two models against the CMIP6 mean, for instance from https://www.frontiersin.org/articles/10.3389/fclim.2021.738224/full, which shows a decline in North Atlantic NPP in both CMIP5 and CMIP6 multi-model means.
Authors: We have added the total change averaged over the entire domain between the final 30 yrs of SSP5-8.5 and the first 30 yrs of historical in Tab. 2. We have also added some discussion on lines:356-366.

L394: Similarly, considering that SSP5-8.5 is the most extreme climate change scenario (where future fossil fuel emission grows to 5x current values!), it barely impacts the NPP in figure 4. Is it possible that these two modelled ecosystems are particularly insensitive to climate change? Are they suitable for this type of analysis or are more flexible BGC models neccesairy for projecting the impact of
climate change on the marine ecosystem?
Authors: As noted further down. The ECS of our two models pretty much span the likely range (66% percent of the probability range) assessed by the IPCC for the ECS. So in that sense, at least, they are not clear outliers in a physical modelling sense. Moreover, the assessment that the extreme climate scenario barely impacts NPP is subjective. That is, there is no good answer to the question of what the expected range of NPP changes for a scenario like SSP5-8.5 should be. Thus, there is no real baseline to which our projected changes can be seen as large or small. The paper referenced above also shows that there are large variations in NPP in the CMIP6 models. In light of this, we don't see why a different, more flexible, model type should be called for. That is not to say that models do need to be improved in many different ways. They certainly do, but the need of a different model type does not seem to follow from the presented arguments.

More importantly, the many questions of realism posed: similarities to observations, feasibility of SSP5-8.5 and differences between CMIP6 models, are in many ways good questions. However, they are not the main target questions of our research. The questions we pose about phenological changes under strong climate change in our models, we find to be interesting basic science regardless of the degree to which these model simulations will capture our future changes. That is, the value of our results is not so intimately tied to the likelihood of e.g. SSP5-8.5 coming to pass as one might think.

L394: I'd like to see some discussion about the suitability of the SSP5-8.5 scenario. It has extremely high fossil fuel emissions and subsequent warming, which is likely to move these change points earlier in the simulation than you would see in other scenarios.

Authors: We have added some text on this to the discussion (lines:384,392)

L394: NorESM2-LM has a particularly low (but feasible) Effective Climate Sensitivity or 2.54K

(https://gmd.copernicus.org/articles/13/6165/2020/), while EC-Earth3 has an ECS of 4.3K (https://gmd.copernicus.org/articles/13/3465/2020/). Does the difference in their sensitivity to carbon impact the overall conclusions? For instance, is it possible that the surface waters of EC-Earth3 will warm more than NorESM2-LM, and this may shift the locations of habitats in these models.

**Authors: It is possible, but it is far from the only difference between these models. Regional climate, climate variability, Arctic amplification and differing phytoplankton growth model's are just a number of other things that might play a role. With the simulations we have at our disposal it is not possible to attribute differences specifically to ECS differences. Note also that EC-Earth rather than NorESM is the outlier here; the likely range given by AR6 is 2.5-4, with a best estimate 3 degrees warming for a doubling of CO2. Our two models span this likely range fairly closely, but NORESM is much closer to the central estimate than EC-Earth.**

**Conclusions:**

L402: This is the first mention of the growing season. Please add a description of these around your figure 2 results in lines 220.

**Authors: We have removed this statement here as it is not related to our main result. We did however add a line on this in the results (line:234).**

L416-419: Please explain why this is important?

**Authors: We have added this on lines:463-465.**

**Figure captions:**

L655: Seasonal mean vertically integrated NPP-> "Vertically integrated seasonal mean NPP"

**Authors: This has been changed as suggested.**

L686: There are several issues with white space here. Including an extra space in (L1) and (L2), here and in the caption for figure 7.

**Authors: This has been corrected.**

L694: remove period from ".Figure 7".

**Authors: This has been corrected.**

L698: No period at the end.

**Authors: This has been corrected.**

Figure 2:

● This colour bar should be made with a pointed end at the maximum value (use extend = 'max' in matplotlib), to indicate that the highest values shown are beyond the end of the scale.

● It's a shame you don't include DJF, as it looks like both models really struggle to capture the behaviour then in figure 3.

**Authors: We have changed the colorbar so that it shows the full range. Unfortunately, data for DJF is not present in CAFE as the satellites do not see the study area in winter. We have also removed SON as was suggested by reviewer 1.**

Figures 1, 3 and 9:

● It would be nice if the line colours matched the colours in the map (and the region labels in figure 4). See my comments earlier.

**Authors: We have colored the province labels in accordance with Fig. 1. Fig. 3 has been separated into provinces instead as suggested by reviewer 1.**

Figure 6 & 7:

● While I understand why all figures share the same y axis, perhaps this figure would be better served by each region showing it's own bespoke range so that the change point is easier to see. If you

have to, you can move the region label above the axes.
- Please add ticks to the x axes of the top 6 panes.
- Move the legend outside the figure – ideally below the main figure.
- Can you also the solid and dashed lines to the legend.
- Replace the sidewards pointing triangle with a vertical pointing triangle.
- Is there any observational data for this that can be added? If you can't find anything, WOA has monthly MLD: https://www.ncei.noaa.gov/access/world-ocean-atlas-2018/bin/woa18.pl?parameter=M

**Authors:** We don't believe observational estimates of MLD would add anything useful to the investigation made here. Similarities would at best be climatological, as the internal variability is not in phase between our history and that modeled by the ESMs. Moreover, the ability of our ESMs to model MLDs, although an interesting question in its own right, is rather far from the topic of this article.

In the past, I've seen change point analysis that included a trend line either side of the change point. Is there any reason why this was not included here?

**Authors: We have given the subplots individual y-axes. We have added x-ticks, the legends have been moved outside the main figure, solid and dashed lines have been added to the legend and the triangles are now pointing upwards in accordance with the reviewers suggestions.**

**Regarding the trend line we find it much harder to argue for the plotting of a trend line on either side of the change point than against. Firstly, because the presence of a change point, an optimal segmentation of a time series, does not to our knowledge imply the presence of trends in the segments it separates. Secondly, because the kernel method identifies change points of many different kinds it would not be clear if one should look for a trend in mean, variance or something else.**

Figure 8

- You don't need the thick black contours here, just the colour scale might be more readable. As it stands, the figure really emphasizes the regions where the first change point is before the year 2000.
**Authors: We have removed the contours from Fig. 8.**

Figure 9:

- Can you remake this plot with the same style as figure 6 and 7?
- Can you highlight the times when you're within the 95% confidence bands, perhaps by making the line or dots thicker, or making them thinner when you're outside the confidence bands?
**Authors: We have added colored province labels to the plot and changed the colors. We have increased the thickness of the 95% confidence band.**

**Supplementary data**

There's no readme to describe the contents of each file.
**Authors:We have added Readmes**

**Supplementary Model**

The directory structure is not straightforward to understand. The structure described in the Readme doesn't match up with the directories in the zip. Please make the leading directory names more explicit.
**Authors: This is the officially released source code of NorESM2 that was used for CMIP6 simulations. In the README file there is a link to an extensive documentation web-page. We are sorry but it is not possible to change this release.**

---

## Author Response (AR3)

Response to revised "Net primary production annual maxima in the North Atlantic projected to shift in the 21st century."

Jenny Hieronymus, Magnus Hieronymus, Matthias Gröger, Jörg Schwinger, Raffaele Bernadello, Etienne Tourigny, Valentina Sicardi, Itzel Ruvalcaba Baroni, and Klaus Wyser

Submitted to Biogeoscience, BG-2023-54

Review by Dr. Lee de Mora, Plymouth Marine Laboratory, Plymouth, UK

Thanks for your hard work. The abstract, introduction are all significantly improved, with a much clearer flow and logic. The methods and results sections have been streamlined and the overall tone of the text makes it easier to follow. The revised figures look great and support your conclusions. I'm also glad that you are now using consistent colouring for the regions in your figures. It does really help! As does adding regular reminders of the region names in the text.

Also, in addition to the changes to the manuscript, thanks for your patience in answering my questions in the Author's response. This has helped me understand the work and clarified the overall paper.

I'm happy for this paper to accepted subject to a small number of very minor changes described below.

This is a great work and congratulations.

Lee de Mora

**Authors: Thank you again for the ambitious and thorough review that has greatly improved our work.**

**Minor comments:**

Please note that the line numbers here refer to the tracked changes document: https://editor.copernicus.org/index.php?_mdl=msover_md&_jrl=11&_lcm=oc104lcm105m&_acm=get_authors_tracked_changes_file&_ms=110119&id=2268736&salt=1650201345432405649

L163 – Please add a sentence justifying the use of SSP5-8.5 (as you have in your response to the previous review.)

**Authors: This has been done on lines 101-102.**

L243: Move the link to a reference.

**Authors: The link has been moved to Data Availability.**

Results section: It looks strange to me to refer to a decade as 2085s? Typically, the decade 2080-2089 would be the "2080s", not 2085s.

**Authors: This was done to simplify the notation of the 30yr periods: 1850-1879 (1865s), 1985-2014 (2000s) and 2070-2099 (2085s) as was suggested by reviewer 1. The definition to 2085s is stated on line 274.**

L412, L415: When you link to a software routine, it's best to use a monospace font like `Courier New` or something, or the command: \texttt{ks_2samp} in latex. ie `ks_2samp, crosscorr`

**Authors: This has been changed.**

L637: "That is, the large range of correlated lags indicate that the whole story is not simply that MLD acts as a control on NPP." – this is a bit strangely worded, can you re-phrase it please? Maybe something like: "The large range of correlated lags indicate that that NPP is likely to be controlled by other factors, in addition to MLD." Or something like that?

**Authors: We have rewritten.**

Finally, the conclusion sections needs a closing statement of some kind

**Authors: We have added a closing statement.**

Bibliography:

- DOI's sometimes have an http link and sometimes don't?
- L729: Extra spaces around "Sverdrup ' s"
- Sometimes author lists are separated by "and" sometimes by "&"
- Sometimes journals are abbreviated and sometimes they're fully named
    - ie: L 764: Geoscientific Model Development but L767: Geosci. Model Dev
- L898: Elsewhere Authors initials have periods: , ie: Smyth **T.J.**

**Authors: This has been corrected.**

As I said in my previous review, a citation manager should be able to automate this next time. It will save time in the long run. However, BG's typeset editor will likely notice (and hopefully correct) other issues here too.